# SINGLE-TIMESCALE ACTOR-CRITIC PROVABLY FINDS GLOBALLY OPTIMAL POLICY

**Zuyue Fu**
Northwestern University
zuyue.fu@u.northwestern.edu

**Zhuoran Yang**
Princeton University
zy6@princeton.edu

**Zhaoran Wang**
Northwestern University
zhaoranwang@gmail.com

## ABSTRACT

We study the global convergence and global optimality of actor-critic, one of the most popular families of reinforcement learning algorithms. While most existing works on actor-critic employ bi-level or two-timescale updates, we focus on the more practical single-timescale setting, where the actor and critic are updated simultaneously. Specifically, in each iteration, the critic update is obtained by applying the Bellman evaluation operator only once while the actor is updated in the policy gradient direction computed using the critic. Moreover, we consider two function approximation settings where both the actor and critic are represented by linear or deep neural networks. For both cases, we prove that the actor sequence converges to a globally optimal policy at a sublinear $O(K^{-1/2})$ rate, where $K$ is the number of iterations. To the best of our knowledge, we establish the rate of convergence and global optimality of single-timescale actor-critic with linear function approximation for the first time. Moreover, under the broader scope of policy optimization with nonlinear function approximation, we prove that actor-critic with deep neural network finds the globally optimal policy at a sublinear rate for the first time.

## 1 INTRODUCTION

In reinforcement learning (RL) (Sutton et al., 1998), the agent aims to make sequential decisions that maximize the expected total reward through interacting with the environment and learning from the experiences, where the environment is modeled as a Markov Decision Process (MDP) (Puterman, 2014). To learn a policy that achieves the highest possible total reward in expectation, the actor-critic method (Konda and Tsitsiklis, 2000) is among the most commonly used algorithms. In actor-critic, the actor refers to the policy and the critic corresponds to the value function that characterizes the performance of the actor. This method directly optimizes the expected total return over the policy class by iteratively improving the actor, where the update direction is determined by the critic. In particular, recently, actor-critic combined with deep neural networks (LeCun et al., 2015) achieves tremendous empirical successes in solving large-scale RL tasks, such as the game of Go (Silver et al., 2017), StarCraft (Vinyals et al., 2019), Dota (OpenAI, 2018), Rubik's cube (Agostinelli et al., 2019; Akkaya et al., 2019), and autonomous driving (Sallab et al., 2017). See Li (2017) for a detailed survey of the recent developments of deep reinforcement learning.

Despite these great empirical successes of actor-critic, there is still an evident chasm between theory and practice. Specifically, to establish convergence guarantees for actor-critic, most existing works either focus on the bi-level setting or the two-timescale setting, which are seldom adopted in practice. In particular, under the bi-level setting (Yang et al., 2019a; Wang et al., 2019; Agarwal et al., 2019; Fu et al., 2019; Liu et al., 2019; Abbasi-Yadkori et al., 2019a;b; Cai et al., 2019; Hao et al., 2020; Mei et al., 2020; Bhandari and Russo, 2020), the actor is updated only after the critic solves the policy evaluation sub-problem completely, which is equivalent to applying the Bellman evaluation operator to the previous critic for infinite times. Consequently, actor-critic under the bi-level setting

is a double-loop iterative algorithm where the inner loop is allocated for solving the policy evaluation sub-problem of the critic. In terms of theoretical analysis, such a double-loop structure decouples the analysis for the actor and critic. For the actor, the problem is essentially reduced to analyzing the convergence of a variant of the policy gradient method (Sutton et al., 2000; Kakade, 2002) where the error of the gradient estimate depends on the policy evaluation error of the critic. Besides, under the two-timescale setting (Borkar and Konda, 1997; Konda and Tsitsiklis, 2000; Xu et al., 2020; Wu et al., 2020; Hong et al., 2020), the actor and the critic are updated simultaneously, but with disparate stepsizes. More concretely, the stepsize of the actor is set to be much smaller than that of the critic, with the ratio between these stepsizes converging to zero. In an asymptotic sense, such a separation between stepsizes ensures that the critic completely solves its policy evaluation sub-problem asymptotically. In other words, such a two-timescale scheme results in a separation between actor and critic in an asymptotic sense, which leads to asymptotically unbiased policy gradient estimates. In sum, in terms of convergence analysis, the existing theory of actor-critic hinges on decoupling the analysis for critic and actor, which is ensured via focusing on the bi-level or two-timescale settings.

However, most practical implementations of actor-critic are under the single-timescale setting (Peters and Schaal, 2008a; Schulman et al., 2015; Mnih et al., 2016; Schulman et al., 2017; Haarnoja et al., 2018), where the actor and critic are simultaneously updated, and particularly, the actor is updated without the critic reaching an approximate solution to the policy evaluation sub-problem. Meanwhile, in comparison with the two-timescale setting, the actor is equipped with a much larger stepsize in the the single-timescale setting such that the asymptotic separation between the analysis of actor and critic is no longer valid.

Furthermore, when it comes to function approximation, most existing works only analyze the convergence of actor-critic with either linear function approximation (Xu et al., 2020; Wu et al., 2020; Hong et al., 2020), or shallow-neural-network parameterization (Wang et al., 2019; Liu et al., 2019). In contrast, practically used actor-critic methods such as asynchronous advantage actor-critic (Mnih et al., 2016) and soft actor-critic (Haarnoja et al., 2018) oftentimes represent both the actor and critic using deep neural networks.

Thus, the following question is left open:

> *Does single-timescale actor-critic provably find a globally optimal policy under the function approximation setting, especially when deep neural networks are employed?*

To answer such a question, we make the first attempt to investigate the convergence and global optimality of single-timescale actor-critic with linear and neural network function approximation. In particular, we focus on the family of energy-based policies and aim to find the optimal policy within this class. Here we represent both the energy function and the critic as linear or deep neural network functions. In our actor-critic algorithm, the actor update follows proximal policy optimization (PPO) (Schulman et al., 2017) and the critic update is obtained by applying the Bellman evaluation operator only once to the current critic iterate. As a result, the actor is updated before the critic solves the policy evaluation sub-problem. Such a coupled updating structure persists even when the number of iterations goes to infinity, which implies that the update direction of the actor is always biased compared with the policy gradient direction. This brings an additional challenge that is absent in the bi-level and the two-timescale settings, where the actor and critic are decoupled asymptotically.

To tackle such a challenge, our analysis captures the joint effect of actor and critic updates on the objective function, dubbed as the "double contraction" phenomenon, which plays a pivotal role for the success of single-timescale actor-critic. Specifically, thanks to the discount factor of the MDP, the Bellman evaluation operator is contractive, which implies that, after each update, the critic makes noticeable progress by moving towards the value function associated with the current actor. As a result, although we use a biased estimate of the policy gradient, thanks to the contraction brought by the discount factor, the accumulative effect of the biases is controlled. Such a phenomenon enables us to characterize the progress of each iteration of joint actor and critic update, and thus yields the convergence to the globally optimal policy. In particular, for both the linear and neural settings, we prove that, single-timescale actor-critic finds a $O(K^{-1/2})$-globally optimal policy after $K$ iterations. To the best of our knowledge, we seem to establish the first theoretical guarantee of global convergence and global optimality for actor-critic with function approximation in the single-timescale setting. Moreover, under the broader scope of policy optimization with nonlinear function

approximation, our work seems to prove convergence and optimality guarantees for actor-critic with deep neural network for the first time.

**Contribution.** Our contribution is two-fold. First, in the single-timescale setting with linear function approximation, we prove that, after $K$ iterations of actor and critic updates, actor-critic returns a policy that is at most $O(K^{-1/2})$ inferior to the globally optimal policy. Second, when both the actor and critic are represented by deep neural networks, we prove a similar $O(K^{-1/2})$ rate of convergence to the globally optimal policy when the architecture of the neural networks are properly chosen.

**Related Work.** Our work extends the line of works on the convergence of actor-critic under the function approximation setting. In particular, actor-critic is first introduced in Sutton et al. (2000); Konda and Tsitsiklis (2000). Later, Kakade (2002); Peters and Schaal (2008b) propose the natural actor-critic method which updates the policy via the natural gradient (Amari, 1998) direction. The convergence of (natural) actor-critic with linear function approximation are studied in Bhatnagar et al. (2008; 2009); Bhatnagar (2010); Castro and Meir (2010); Maei (2018). However, these works only characterize the asymptotic convergence of actor-critic and their proofs all resort to tools from stochastic approximation via ordinary differential equations (Borkar, 2008). As a result, these works only show that actor-critic with linear function approximation converges to the set of stable equilibria of a set of ordinary differential equations. Recently, Zhang et al. (2019) propose a variant of actor-critic where Monte-Carlo sampling is used to ensure the critic and the policy gradient estimates are unbiased. Although they incorporate nonlinear function approximation in the actor, they only establish finite-time convergence result to a stationary point of the expected total reward. Moreover, due to having an inner loop for solving the policy evaluation sub-problem, they focus on the bi-level setting. Moreover, under the two-timescale setting, Wu et al. (2020); Xu et al. (2020) show that actor-critic with linear function approximation finds an $\varepsilon$-stationary point with $\widetilde{O}(\varepsilon^{-5/2})$ samples, where $\varepsilon$ measures the squared norm of the policy gradient. All of these results establish the convergence of actor-critic, without characterizing the optimality of the policy obtained by actor-critic.

In terms of the global optimality of actor-critic, Fazel et al. (2018); Malik et al. (2018); Tu and Recht (2018); Yang et al. (2019a); Bu et al. (2019); Fu et al. (2019) show that policy gradient and bi-level actor-critic methods converge to the globally optimal policies under the linear-quadratic setting, where the state transitions follow a linear dynamical system and the reward function is quadratic. For general MDPs, Bhandari and Russo (2019) recently prove the global optimality of vanilla policy gradient under the assumption that the families of policies and value functions are both convex. In addition, our work is also related to Liu et al. (2019) and Wang et al. (2019), where they establish the global optimality of proximal policy optimization and (natural) actor-critic, respectively, where both the actor and critic are parameterized by two-layer neural networks. Our work is also related to Agarwal et al. (2019); Abbasi-Yadkori et al. (2019a;b); Cai et al. (2019); Hao et al. (2020); Mei et al. (2020); Bhandari and Russo (2020), which focus on characterizing the optimality of natural policy gradient in tabular and/or linear settings. However, these aforementioned works all focus on bi-level actor-critic, where the actor is updated only after the critic solves the policy evaluation sub-problem to an approximate optimum. Besides, these works consider linear or two-layer neural network function approximations whereas we focus on the setting with deep neural networks. Furthermore, under the two-timescale setting, Xu et al. (2020); Hong et al. (2020) prove that linear actor-critic requires a sample complexity of $\widetilde{O}(\varepsilon^{-4})$ for obtaining an $\varepsilon$-globally optimal policy. In comparison, our $O(K^{-1/2})$ convergence for single-timescale actor-critic can be translated into a similar $\widetilde{O}(\varepsilon^{-4})$ sample complexity directly. Moreover, when reusing the data, our result leads to an improved $\widetilde{O}(\varepsilon^{-2})$ sample complexity. In addition, our work is also related to Geist et al. (2019), which proposes a variant of policy iteration algorithm with Bregman divergence regularization. Without considering an explicit form of function approximation, their algorithm is shown to converge to the globally optimal policy at a similar $O(K^{-1/2})$ rate, where $K$ is the number of policy updates. In contrast, our method is single-timescale actor-critic with linear or deep neural network function approximation, which enjoys both global convergence and global optimality. Meanwhile, our proof is based on a finite-sample analysis, which involves dealing with the algorithmic errors that track the performance of actor and critic updates as well as the statistical error due to having finite data.

Our work is also related to the literature on deep neural networks. Previous works (Daniely, 2017; Jacot et al., 2018; Wu et al., 2018; Allen-Zhu et al., 2018a;b; Du et al., 2018; Zou et al., 2018; Chizat and Bach, 2018; Jacot et al., 2018; Li and Liang, 2018; Cao and Gu, 2019a;b; Arora et al., 2019; Lee et al., 2019; Gao et al., 2019) analyze the computational and statistical rates of supervised learning methods with overparameterized neural networks. In contrast, our work employs overparameterized deep neural networks in actor-critic for solving RL tasks, which is significantly more challenging than supervised learning due to the interplay between the actor and the critic.

**Notation.** We denote by $[n]$ the set $\{1, 2, \ldots, n\}$. For any measure $\nu$ and $1 \leq p \leq \infty$, we denote by $\|f\|_{\nu,p} = (\int_{\mathcal{X}} |f(x)|^p \mathrm{d}\nu)^{1/p}$ and $\|f\|_p = (\int_{\mathcal{X}} |f(x)|^p \mathrm{d}\mu)^{1/p}$, where $\mu$ is the Lebesgue measure.

## 2 BACKGROUND

In this section, we introduce the background on discounted Markov decision processes (MDPs) and actor-critic methods.

### 2.1 DISCOUNTED MDP

A discounted MDP is defined by a tuple $(\mathcal{S}, \mathcal{A}, P, \zeta, r, \gamma)$. Here $\mathcal{S}$ and $\mathcal{A}$ are the state and action spaces, respectively, $P \colon \mathcal{S} \times \mathcal{S} \times \mathcal{A} \to [0, 1]$ is the Markov transition kernel, $\zeta \colon \mathcal{S} \to [0, 1]$ is the initial state distribution, $r \colon \mathcal{S} \times \mathcal{A} \to \mathbb{R}$ is the deterministic reward function, and $\gamma \in [0, 1)$ is the discount factor. A policy $\pi(a \,|\, s)$ measures the probability of taking the action $a$ at the state $s$. We focus on a family of parameterized policies defined as follows,

$$\Pi = \{\pi_\theta(\cdot \,|\, s) \in \mathcal{P}(\mathcal{A}) \colon s \in \mathcal{S}\}, \tag{2.1}$$

where $\mathcal{P}(\mathcal{A})$ is the probability simplex on the action space $\mathcal{A}$ and $\theta$ is the parameter of the policy $\pi_\theta$. For any state-action pair $(s, a) \in \mathcal{S} \times \mathcal{A}$, we define the action-value function as follows,

$$Q^\pi(s, a) = (1 - \gamma) \cdot \mathbb{E}_\pi\Big[\sum_{t=0}^\infty \gamma^t \cdot r(s_t, a_t) \,\Big|\, s_0 = s, a_0 = a\Big], \tag{2.2}$$

where $s_{t+1} \sim P(\cdot \,|\, s_t, a_t)$ and $a_{t+1} \sim \pi(\cdot \,|\, s_{t+1})$ for any $t \geq 0$. We use $\mathbb{E}_\pi[\cdot]$ to denote that the actions follow the policy $\pi$, which further affect the transition of the states. We aim to find an optimal policy $\pi^*$ such that $Q^{\pi^*}(s, a) \geq Q^\pi(s, a)$ for any policy $\pi$ and state-action pair $(s, a) \in \mathcal{S} \times \mathcal{A}$. That is to say, such an optimal policy $\pi^*$ attains a higher expected total reward than any other policy $\pi$, regardless of the initial state-action pair $(s, a)$. For notational convenience, we denote by $Q^*(s, a) = Q^{\pi^*}(s, a)$ for any $(s, a) \in \mathcal{S} \times \mathcal{A}$ hereafter.

Meanwhile, we denote by $\nu_\pi(s)$ and $\rho_\pi(s, a) = \nu_\pi(s) \cdot \pi(a \,|\, s)$ the stationary state distribution and stationary state-action distribution of the policy $\pi$, respectively, for any $(s, a) \in \mathcal{S} \times \mathcal{A}$. Correspondingly, we denote by $\nu^*(s)$ and $\rho^*(s, a)$ the stationary state distribution and stationary state-action distribution of the optimal policy $\pi^*$, respectively, for any $(s, a) \in \mathcal{S} \times \mathcal{A}$. For ease of presentation, given any functions $g_1 \colon \mathcal{S} \to \mathbb{R}$ and $g_2 \colon \mathcal{S} \times \mathcal{A} \to \mathbb{R}$, we define two operators $\mathbb{P}$ and $\mathbb{P}^\pi$ as follows,

$$[\mathbb{P}g_1](s, a) = \mathbb{E}[g_1(s_1) \,|\, s_0 = s, a_0 = a], \quad [\mathbb{P}^\pi g_2](s, a) = \mathbb{E}_\pi[g_2(s_1, a_1) \,|\, s_0 = s, a_0 = a], \tag{2.3}$$

where $s_1 \sim P(\cdot \,|\, s_0, a_0)$ and $a_1 \sim \pi(\cdot \,|\, s_1)$. Intuitively, given the current state-action pair $(s_0, a_0)$, the operator $\mathbb{P}$ pushes the agent to its next state $s_1$ following the Markov transition kernel $P(\cdot \,|\, s_0, a_0)$, while the operator $\mathbb{P}^\pi$ pushes the agent to its next state-action pair $(s_1, a_1)$ following the Markov transition kernel $P(\cdot \,|\, s_0, a_0)$ and policy $\pi(\cdot \,|\, s_1)$. These operators also relate to the Bellman evaluation operator $\mathbb{T}^\pi$, which is defined for any function $g \colon \mathcal{S} \times \mathcal{A} \to \mathbb{R}$ as follows,

$$\mathbb{T}^\pi g = (1 - \gamma) \cdot r + \gamma \cdot \mathbb{P}^\pi g. \tag{2.4}$$

The Bellman evaluation operator $\mathbb{T}^\pi$ is used to characterize the actor-critic method in the following section. By the definition in (2.2), it is straightforward to verify that the action-value function $Q^\pi$ is the fixed point of the Bellman evaluation operator $\mathbb{T}^\pi$ defined in (2.4), that is, $Q^\pi = \mathbb{T}^\pi Q^\pi$ for any policy $\pi$. For notational convenience, we let $\mathbb{P}^\ell$ denote the $\ell$-fold composition $\mathbb{P}\mathbb{P} \cdots \mathbb{P}$, where there are $\ell$ operators $\mathbb{P}$ composed together. Such notation is also adopted for other linear operators such as $\mathbb{P}^\pi$ and $\mathbb{T}^\pi$.

## 2.2 ACTOR-CRITIC METHOD

To obtain an optimal policy $\pi^*$, the actor-critic method (Konda and Tsitsiklis, 2000) aims to maximize the expected total reward as a function of the policy, which is equivalent to solving the following maximization problem,

$$\max_{\pi \in \Pi} J(\pi) = \mathbb{E}_{s \sim \zeta, a \sim \pi(\cdot \mid s)} \big[ Q^\pi(s, a) \big], \tag{2.5}$$

where $\zeta$ is the initial state distribution, $Q^\pi$ is the action-value function defined in (2.2), and the family of parameterized polices $\Pi$ is defined in (2.1). The actor-critic method solves the maximization problem in (2.5) via first-order optimization using an estimator of the policy gradient $\nabla_\theta J(\pi)$. Here $\theta$ is the parameter of the policy $\pi$. In detail, by the policy gradient theorem (Sutton et al., 2000), we have

$$\nabla_\theta J(\pi) = \mathbb{E}_{(s,a) \sim \varrho_\pi} \big[ Q^\pi(s, a) \cdot \nabla_\theta \log \pi(a \mid s) \big]. \tag{2.6}$$

Here $\varrho_\pi$ is the state-action visitation measure of the policy $\pi$, which is defined as $\varrho_\pi(s, a) = (1 - \gamma) \cdot \sum_{t=0}^\infty \gamma^t \cdot \Pr[s_t = s, a_t = a]$. Based on the closed form of the policy gradient in (2.6), the actor-critic method consists of the following two parts: (i) the critic update, where a policy evaluation algorithm is invoked to estimate the action-value function $Q^\pi$, e.g., by applying the Bellman evaluation operator $\mathbb{T}^\pi$ to the current estimator of $Q^\pi$, and (ii) the actor update, where a policy improvement algorithm, e.g., the policy gradient method, is invoked using the updated estimator of $Q^\pi$.

In this paper, we consider the following variant of the actor-critic method,

$$\pi_{k+1} \leftarrow \underset{\pi \in \Pi}{\operatorname{argmax}} \, \mathbb{E}_{\nu_{\pi_k}} \big[ \langle Q_k(s, \cdot), \pi(\cdot \mid s) \rangle - \beta \cdot \mathrm{KL}\big( \pi(\cdot \mid s) \, \| \, \pi_k(\cdot \mid s) \big) \big],$$

$$Q_{k+1}(s, a) \leftarrow \mathbb{E}_{\pi_{k+1}} \big[ (1 - \gamma) \cdot r(s_0, a_0) + \gamma \cdot Q_k(s_1, a_1) \, \big| \, s_0 = s, a_0 = a \big], \tag{2.7}$$

for any $(s, a) \in \mathcal{S} \times \mathcal{A}$, where $s_1 \sim P(\cdot \mid s_0, a_0)$, $a_1 \sim \pi_{k+1}(\cdot \mid s_1)$, and we write $\mathbb{E}_{\nu_{\pi_k}}[\cdot] = \mathbb{E}_{s \sim \nu_{\pi_k}}[\cdot]$ for notational convenience. Here $\Pi$ is defined in (2.1) and $\mathrm{KL}(\pi(\cdot \mid s) \, \| \, \pi_k(\cdot \mid s))$ is the Kullback-Leibler (KL) divergence between $\pi(\cdot \mid s)$ and $\pi_k(\cdot \mid s)$, which is defined for any $s \in \mathcal{S}$ as follows, $\mathrm{KL}(\pi(\cdot \mid s) \, \| \, \pi_k(\cdot \mid s)) = \sum_{a \in \mathcal{A}} \log(\pi(a \mid s)/\pi_k(a \mid s)) \cdot \pi(a \mid s)$. In (2.7), the actor update uses the proximal policy optimization (PPO) method (Schulman et al., 2017), while the critic update applies the Bellman evaluation operator $\mathbb{T}^{\pi_{k+1}}$ defined in (2.4) to $Q_k$ only once, which is the current estimator of the action-value function. Furthermore, we remark that the updates in (2.7) provide a general framework in the following two aspects. First, the critic update can be extended to letting $Q_{k+1} \leftarrow (\mathbb{T}^{\pi_{k+1}})^\tau Q_k$ for any fixed $\tau \geq 1$, which corresponds to updating the value function via $\tau$-step rollouts following $\pi_{k+1}$. Here we only focus on the case with $\tau = 1$ for simplicity. Our theory can be easily modified for any fixed $\tau$. Moreover, the KL divergence used in the actor step can also be replaced by other Bregman divergences between probability distributions over $\mathcal{A}$. Second, the actor and critic updates in (2.7) is a general template that admits both on- and off-policy evaluation methods and various function approximators in the actor and critic. In the next section, we present an incarnation of (2.7) with on-policy sampling and linear and neural network function approximation.

Furthermore, for analyzing the actor-critic method, most existing works (Yang et al., 2019a; Wang et al., 2019; Agarwal et al., 2019; Fu et al., 2019; Liu et al., 2019) rely on (approximately) obtaining $Q^{\pi_{k+1}}$ at each iteration, which is equivalent to applying the Bellman evaluation operator $\mathbb{T}^{\pi_{k+1}}$ infinite times to $Q_k$. This is usually achieved by minimizing the mean-squared Bellman error $\|Q - \mathbb{T}^{\pi_{k+1}} Q\|_{\rho_{\pi_{k+1}},2}^2$ using stochastic semi-gradient descent, e.g., as in the temporal-difference method (Sutton, 1988), to update the critic for sufficiently many iterations. The unique global minimizer of the mean-squared Bellman error gives the action-value function $Q^{\pi_{k+1}}$, which is used in the actor update. Meanwhile, the two-timescale setting is also considered in existing works (Borkar and Konda, 1997; Konda and Tsitsiklis, 2000; Xu et al., 2019; 2020; Wu et al., 2020; Hong et al., 2020), which require the actor to be updated more slowly than the critic in an asymptotic sense. Such a requirement is usually satisfied by forcing the ratio between the stepsizes of the actor and critic updates to go to zero asymptotically.

In comparison with the setting with bi-level updates, we consider the single-timescale actor and critic updates in (2.7), where the critic involves only one step of update, that is, applying the Bellman evaluation operator $\mathbb{T}^\pi$ to $Q_k$ only once. Meanwhile, in comparison with the two-timescale

setting, where the actor and critic are updated simultaneously but with the ratio between their step-sizes asymptotically going to zero, the single-timescale setting is able to achieve a faster rate of convergence by allowing the actor to be updated with a larger stepsize, while updating the critic simultaneously. In particular, such a single-timescale setting better captures a broader range of practical algorithms (Peters and Schaal, 2008a; Schulman et al., 2015; Mnih et al., 2016; Schulman et al., 2017; Haarnoja et al., 2018), where the stepsize of the actor is not asymptotically zero. In §3, we discuss the implementation of the updates in (2.7) for different schemes of function approximation. In §4, we compare the rates of convergence between the two-timescale and single-timescale settings.

## 3  ALGORITHMS

We consider two settings, where the actor and critic are parameterized using linear functions and deep neural networks (which is deferred to §A of the appendix), respectively. We consider the energy-based policy $\pi_\theta(a \,|\, s) \propto \exp(\tau^{-1} f_\theta(s, a))$, where the energy function $f_\theta(s, a)$ is parameterized with the parameter $\theta$. Also, for the (estimated) action-value function, we consider the parameterization $Q_\omega(s, a)$ for any $(s, a) \in \mathcal{S} \times \mathcal{A}$, where $\omega$ is the parameter. For such parameterizations of the actor and critic, the updates in (2.7) have the following forms.

**Actor Update.** The following proposition gives the closed form of $\pi_{k+1}$ in (2.7).

**Proposition 3.1.** Let $\pi_{\theta_k}(a \,|\, s) \propto \exp(\tau_k^{-1} f_{\theta_k}(s, a))$ be an energy-based policy and $\widetilde{\pi}_{k+1} = \mathrm{argmax}_\pi \mathbb{E}_{\nu_k}\big[\langle Q_{\omega_k}(s, \cdot), \pi(\cdot \,|\, s)\rangle - \beta \cdot \mathrm{KL}\big(\pi(\cdot \,|\, s) \,\|\, \pi_{\theta_k}(\cdot \,|\, s)\big)\big]$. Then $\widetilde{\pi}_{k+1}$ has the following closed form: $\widetilde{\pi}_{k+1}(a \,|\, s) \propto \exp\big(\beta^{-1} Q_{\omega_k}(s, a) + \tau_k^{-1} f_{\theta_k}(s, a)\big)$, for any $(s, a) \in \mathcal{S} \times \mathcal{A}$, where $\nu_k = \nu_{\pi_{\theta_k}}$ is the stationary state distribution of $\pi_{\theta_k}$.

See §G.1 for a detailed proof of Proposition 3.1. Motivated by Proposition 3.1, to implement the actor update in (2.7), we update the actor parameter $\theta$ by solving the following minimization problem,

$$\theta_{k+1} \leftarrow \mathrm{argmin}_\theta \mathbb{E}_{\rho_k}\big[\big(f_\theta(s, a) - \tau_{k+1} \cdot \big(\beta^{-1} Q_{\omega_k}(s, a) + \tau_k^{-1} f_{\theta_k}(s, a)\big)\big)^2\big], \tag{3.1}$$

where $\rho_k = \rho_{\pi_{\theta_k}}$ is the stationary state-action distribution of $\pi_{\theta_k}$.

**Critic Update.** To implement the critic update in (2.7), we update the critic parameter $\omega$ by solving the following minimization problem,

$$\omega_{k+1} \leftarrow \mathrm{argmin}_\omega \mathbb{E}_{\rho_{k+1}}\big[\big([Q_\omega - (1-\gamma) \cdot r - \gamma \cdot \mathbb{P}^{\pi_{\theta_{k+1}}} Q_{\omega_k}](s, a)\big)^2\big], \tag{3.2}$$

where $\rho_{k+1} = \rho_{\pi_{\theta_{k+1}}}$ is the stationary state-action distribution of $\pi_{\theta_{k+1}}$ and the operator $\mathbb{P}^\pi$ is defined in (2.3).

### 3.1  LINEAR FUNCTION APPROXIMATION

In this section, we consider linear function approximation. More specifically, we parameterize the action-value function using $Q_\omega(s, a) = \omega^\top \varphi(s, a)$ and the energy function of the energy-based policy $\pi_\theta$ using $f_\theta(s, a) = \theta^\top \varphi(s, a)$. Here $\varphi(s, a) \in \mathbb{R}^d$ is the feature vector, where $d > 0$ is the dimension. Without loss of generality, we assume that $\|\varphi(s, a)\|_2 \le 1$ for any $(s, a) \in \mathcal{S} \times \mathcal{A}$, which can be achieved by normalization.

**Actor Update.** The minimization problem in (3.1) admits the following closed-form solution,

$$\theta_{k+1} = \tau_{k+1} \cdot (\beta^{-1} \omega_k + \tau_k^{-1} \theta_k), \tag{3.3}$$

which corresponds to a step of the natural policy gradient method (Kakade, 2002).

**Critic Update.** The minimization problem in (3.2) admits the following closed-form solution,

$$\widetilde{\omega}_{k+1} = \big(\mathbb{E}_{\rho_{k+1}}[\varphi(s, a)\varphi(s, a)^\top]\big)^{-1} \mathbb{E}_{\rho_{k+1}}\big[[(1-\gamma) \cdot r + \gamma \cdot \mathbb{P}^{\pi_{\theta_{k+1}}} Q_{\omega_k}](s, a) \cdot \varphi(s, a)\big]. \tag{3.4}$$

Since the closed-form solution $\widetilde{\omega}_{k+1}$ in (3.4) involves the expectation over the stationary state-action distribution $\rho_{k+1}$ of $\pi_{\theta_{k+1}}$, we use data to approximate such an expectation. More specifically, we sample $\{(s_{\ell,1}, a_{\ell,1})\}_{\ell \in [N]}$ and $\{(s_{\ell,2}, a_{\ell,2}, r_{\ell,2}, s'_{\ell,2}, a'_{\ell,2})\}_{\ell \in [N]}$ such that $(s_{\ell,1}, a_{\ell,1}) \sim \rho_{k+1}$,

$(s_{\ell,2}, a_{\ell,2}) \sim \rho_{k+1}$, $r_{\ell,2} = r(s_{\ell,2}, a_{\ell,2})$, $s'_{\ell,2} \sim P(\cdot \mid s_{\ell,2}, a_{\ell,2})$, and $a'_{\ell,2} \sim \pi_{\theta_{k+1}}(\cdot \mid s'_{\ell,2})$, where $N$ is the sample size. We approximate $\widetilde{\omega}_{k+1}$ using $\omega_{k+1}$, which is defined as follows,

$$\omega_{k+1} = \Gamma_R \Big\{ \Big( \sum_{\ell=1}^{N} \varphi(s_{\ell,1}, a_{\ell,1}) \varphi(s_{\ell,1}, a_{\ell,1})^\top \Big)^{-1} \tag{3.5}$$
$$\cdot \sum_{\ell=1}^{N} \big( (1-\gamma) \cdot r_{\ell,2} + \gamma \cdot Q_{\omega_k}(s'_{\ell,2}, a'_{\ell,2}) \big) \cdot \varphi(s_{\ell,2}, a_{\ell,2}) \Big\}.$$

Here $\Gamma_R$ is the projection operator, which projects the parameter onto the centered ball with radius $R$ in $\mathbb{R}^d$. Such a projection operator stabilizes the algorithm (Konda and Tsitsiklis, 2000; Bhatnagar et al., 2009). It is worth mentioning that one may also view the update in (3.5) as one step of the least-squares temporal difference method (Bradtke and Barto, 1996), which can be modified for the off-policy setting (Antos et al., 2007; Yu, 2010; Liu et al., 2018; Nachum et al., 2019; Xie et al., 2019; Zhang et al., 2020; Uehara and Jiang, 2019; Nachum and Dai, 2020). Such a modification allows the data points in (3.5) to be reused in the subsequent iterations, which further improves the sample complexity. Specifically, let $\rho_{\mathrm{bhv}} \in \mathcal{P}(\mathcal{S} \times \mathcal{A})$ be the stationary state-action distribution induced by a behavioral policy $\pi_{\mathrm{bhv}}$. We replace the actor and critic updates in (3.1) and (3.2) by

$$\theta_{k+1} \leftarrow \underset{\theta}{\arg\min}\, \mathbb{E}_{\rho_{\mathrm{bhv}}} \big[ \big( f_\theta(s,a) - \tau_{k+1} \cdot \big( \beta^{-1} Q_{\omega_k}(s,a) + \tau_k^{-1} f_{\theta_k}(s,a) \big) \big)^2 \big], \tag{3.6}$$

$$\omega_{k+1} \leftarrow \underset{\omega}{\arg\min}\, \mathbb{E}_{\rho_{\mathrm{bhv}}} \big[ \big( [Q_\omega - (1-\gamma) \cdot r - \gamma \cdot \mathbb{P}^{\pi_{\theta_{k+1}}} Q_{\omega_k}](s,a) \big)^2 \big], \tag{3.7}$$

respectively. With linear function approximation, the actor update in (3.6) is reduced to (3.3), while the critic update in (3.7) admits a closed form solution

$$\widetilde{\omega}_{k+1} = \big( \mathbb{E}_{\rho_{\mathrm{bhv}}}[\varphi(s,a)\varphi(s,a)^\top] \big)^{-1} \cdot \mathbb{E}_{\rho_{\mathrm{bhv}}} \big[ [(1-\gamma) \cdot r + \gamma \cdot \mathbb{P}^{\pi_{\theta_{k+1}}} Q_{\omega_k}](s,a) \cdot \varphi(s,a) \big],$$

which can be well approximated using state-action pairs drawn from $\rho_{\mathrm{bhv}}$. See §4 for a detailed discussion. Finally, by assembling the updates in (3.3) and (3.5), we present the linear actor-critic method in Algorithm 1, which is deferred to §B of the appendix.

## 4 THEORETICAL RESULTS

In this section, we upper bound the regret of the linear actor-critic method. We defer the analysis of the deep neural actor-critic method to §C of the appendix. Hereafter we assume that $|r(s,a)| \leq r_{\max}$ for any $(s,a) \in \mathcal{S} \times \mathcal{A}$, where $r_{\max}$ is a positive absolute constant. First, we impose the following assumptions. Recall that $\rho^*$ is the stationary state-action distribution of $\pi^*$, while $\rho_k$ is the stationary state-action distribution of $\pi_{\theta_k}$. Moreover, let $\rho \in \mathcal{P}(\mathcal{S} \times \mathcal{A})$ be a state-action distribution with respect to which we aim to characterize the performance of the actor-critic algorithm. Specifically, after $K + 1$ actor updates, we are interested in upper bounding the following regret

$$\mathbb{E}\Big[ \sum_{k=0}^{K} \big( \|Q^* - Q^{\pi_{\theta_{k+1}}}\|_{\rho,1} \big) \Big] = \mathbb{E}\Big[ \sum_{k=0}^{K} \big( Q^*(s,a) - Q^{\pi_{\theta_{k+1}}}(s,a) \big) \Big], \tag{4.1}$$

where the expectation is taken with respect to $\{\theta_k\}_{k \in [K+1]}$ and $(s,a) \sim \rho$. Here we allow $\rho$ to be any fixed distribution for generality, which might be different from $\rho^*$.

**Assumption 4.1** (Concentrability Coefficient). The following statements hold.

(i) There exists a positive absolute constant $\phi^*$ such that $\phi_k^* \leq \phi^*$ for any $k \geq 1$, where $\phi_k^* = \|\mathrm{d}\rho^*/\mathrm{d}\rho_k\|_{\rho_k, 2}$.

(ii) We assume that for any $k \geq 1$ and a sequence of policies $\{\pi_i\}_{i \geq 1}$, the $k$-step future-state-action distribution $\rho \mathbb{P}^{\pi_1} \cdots \mathbb{P}^{\pi_k}$ is absolutely continuous with respect to $\rho^*$, where $\rho$ is the same as the one in (4.1) Also, it holds for such $\rho$ that $C_{\rho,\rho^*} = (1-\gamma)^2 \sum_{k=1}^{\infty} k^2 \gamma^k \cdot c(k) < \infty$, where $c(k) = \sup_{\{\pi_i\}_{i \in [k]}} \|\mathrm{d}(\rho \mathbb{P}^{\pi_1} \cdots \mathbb{P}^{\pi_k})/\mathrm{d}\rho^*\|_{\rho^*, \infty}$.

In Assumption 4.1, $C_{\rho,\rho^*}$ is known as the discounted-average concentrability coefficient of the future-state-action distributions. Such an assumption indeed measures the stochastic stability properties of the MDP, and the class of MDPs with such properties is quite large. See Szepesvári and

Munos (2005); Munos and Szepesvári (2008); Antos et al. (2008a;b); Scherrer (2013); Scherrer et al. (2015); Farahmand et al. (2016); Yang et al. (2019b); Geist et al. (2019); Chen and Jiang (2019) for more examples and discussion.

**Assumption 4.2** (Zero Approximation Error). It holds for any $\omega, \theta \in \mathcal{B}(0, R)$ that $\inf_{\bar{\omega} \in \mathcal{B}(0,R)} \mathbb{E}_{\rho_{\pi_\theta}} \left[ \left( [\mathbb{T}^{\pi_\theta} Q_\omega - \bar{\omega}^\top \varphi](s, a) \right)^2 \right] = 0$, where $\mathbb{T}^{\pi_\theta}$ is defined in (2.4).

Assumption 4.2 imposes a structural assumption of the MDP under the linear setting. Specifically speaking, it assumes that the Bellman operator of each policy maps a linear value function to a linear function. Therefore, the value function associated with each policy (which is the fixed point of the corresponding Bellman operator) lies in the linear function class. Since the value functions are linear here, the energy-based policy class approximately covers the optimal policy as the temperature parameter $\tau$ goes to zero. In summary, our Assumption 4.2 ensures that the energy-based policy class approximately captures the optimal policy and thus there is no approximation error. When Assumption 4.2 does not hold, we only need to add an additional bias term to the regret upper bound in our theorem without much change in the proof.

**Assumption 4.3** (Well-Conditioned Feature). The minimum singular value of the matrix $\mathbb{E}_{\rho_k}[\varphi(s, a)\varphi(s, a)^\top]$ is uniformly lower bounded by a positive absolute constant $\sigma^*$ for any $k \geq 1$.

Assumption 4.3 ensures that the minimization problem in (3.2) admits a unique minimizer, which is used in the critic update. Similar assumptions are commonly imposed in the literature (Bhandari et al., 2018; Xu et al., 2019; Zou et al., 2019; Wu et al., 2020).

Under Assumptions 4.1, 4.2, and 4.3, we upper bound the regret of Algorithm 1 in the following theorem.

**Theorem 4.4.** We assume that Assumptions 4.1, 4.2, and 4.3 hold. Let $\rho$ be a state-action distribution satisfying (ii) of Assumption 4.1. Also, for any sufficiently large $K > 0$, let $\beta = K^{1/2}$, $N = \Omega(KC^2_{\rho,\rho^*} \cdot (\phi^*/\sigma^*)^2 \cdot \log^2 N)$, and the sequence of policy parameters $\{\theta_k\}_{k \in [K+1]}$ be generated by Algorithm 1. It holds that

$$\mathbb{E}\left[ \sum_{k=0}^{K} \left( Q^*(s, a) - Q^{\pi_{\theta_{k+1}}}(s, a) \right) \right] \leq \left( 2(1-\gamma)^{-3} \cdot \log |\mathcal{A}| + O(1) \right) \cdot K^{1/2}, \qquad (4.2)$$

where the expectation is taken with respect to $\{\theta_k\}_{k \in [K+1]}$ and $(s, a) \sim \rho$.

We sketch the proof in §D. See §E.1 for a detailed proof. Theorem 4.4 establishes an $O(K^{1/2})$ regret of Algorithm 1, where $K$ is the total number of iterations. Here $O(\cdot)$ omits terms involving $(1-\gamma)^{-1}$ and $\log |\mathcal{A}|$. To better understand Theorem 4.4, we consider the ideal setting, where we have access to the action-value function $Q^\pi$ of any policy $\pi$. In such an ideal setting, the critic update is unnecessary. However, the natural policy gradient method, which only uses the actor update, achieves the same $O(K^{1/2})$ regret (Liu et al., 2019; Agarwal et al., 2019; Cai et al., 2019). In other words, in terms of the iteration complexity, Theorem 4.4 shows that in the single-timescale setting, using only one step of the critic update along with one step of the actor update is as efficient as the natural policy gradient method in the ideal setting.

Furthermore, by the regret bound in (4.2), to obtain an $\varepsilon$-globally optimal policy, it suffices to set $K \asymp (1-\gamma)^{-6} \cdot \varepsilon^{-2} \cdot \log^2 |\mathcal{A}|$ in Algorithm 1 and output a randomized policy that is drawn from $\{\pi_{\theta_k}\}_{k=1}^{K+1}$ uniformly. Plugging such a $K$ into $N = \Omega(KC^2_{\rho,\rho^*}(\phi^*/\sigma^*)^2 \cdot \log^2 N)$, we obtain that $N = \widetilde{O}(\varepsilon^{-2})$, where $\widetilde{O}(\cdot)$ omits the logarithmic terms. Thus, to achieve an $\varepsilon$-globally optimal policy, the total sample complexity of Algorithm 1 is $\widetilde{O}(\varepsilon^{-4})$. This matches the sample complexity results established in Xu et al. (2020); Hong et al. (2020) for two-timescale actor-critic methods. Meanwhile, notice that here the critic updates are on-policy and we draw $N$ new data points in each critic update. As discussed in §3.1, under the off-policy setting, the critic updates given in (3.7) can be implemented using a fixed dataset sampled from $\rho_{\mathrm{bhv}}$, the stationary state-action distribution induced by the behavioral policy. Under this scenario, the total number of data points used by the algorithm is equal to $N$. Moreover, by imposing similar assumptions on $\rho_{\mathrm{bhv}}$ as in (i) of Assumption 4.1 and Assumption 4.3, we can establish a similar $O(K^{1/2})$ regret as in (4.2) for the off-policy setting. As a result, with data reuse, to obtain an $\varepsilon$-globally optimal policy, the sample complexity of Algorithm 1 is essentially $\widetilde{O}(\varepsilon^{-2})$, which demonstrates the advantage of our single-timescale

actor-critic method. Besides, only focusing on the convergence to an $\varepsilon$-stationary point, Wu et al. (2020); Xu et al. (2020) establish the sample complexity of $\widetilde{O}(\varepsilon^{-5/2})$ for two-timescale actor-critic, where $\varepsilon$ measures the squared Euclidean norm of the policy gradient. In contrast, by adopting the natural policy gradient (Kakade, 2002) in actor updates, we achieve convergence to the globally optimal policy. We remark that the idea of off-policy evaluation cannot be applied to typical two-timescale setting (Wu et al., 2020; Xu et al., 2020), where the critic is updated using TD learning (e.g. TD(0) and TD($\lambda$)), since it is shown that off-policy TD method may diverge even with linear function approximation (Baird et al., 1995; Sutton et al., 2008). To the best of our knowledge, we establish the rate of convergence and global optimality of the actor-critic method with function approximation in the single-timescale setting for the first time.

Furthermore, as we will show in Theorem C.5 of §B, when both the actor and the critic are represented using overparameterized deep neural networks, we establish a similar $O((1-\gamma)^{-3} \cdot \log |\mathcal{A}| \cdot K^{1/2})$ regret when the architecture of the actor and critic neural networks are properly chosen. To our best knowledge, this seems the first theoretical guarantee for the actor-critic method with deep neural network function approximation in terms of the rate of convergence and global optimality.

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

# A   DEEP NEURAL NETWORK APPROXIMATION

In this section, we consider deep neural network approximation. We first formally define deep neural networks. Then we introduce the actor-critic method under such a parameterization.

A deep neural network (DNN) $u_\theta(x)$ with the input $x \in \mathbb{R}^d$, depth $H$, and width $m$ is defined as

$$x^{(0)} = x, \quad x^{(h)} = \frac{1}{\sqrt{m}} \cdot \sigma(W_h^\top x^{(h-1)}), \text{ for } h \in [H], \quad u_\theta(x) = b^\top x^{(H)}. \qquad \text{(A.1)}$$

Here $\sigma \colon \mathbb{R}^m \to \mathbb{R}^m$ is the rectified linear unit (ReLU) activation function, which is define as $\sigma(y) = (\max\{0, y_1\}, \dots, \max\{0, y_m\})^\top$ for any $y = (y_1, \dots, y_m)^\top \in \mathbb{R}^m$. Also, we have $b \in \{-1, 1\}^m$, $W_1 \in \mathbb{R}^{d \times m}$, and $W_h \in \mathbb{R}^{m \times m}$ for $2 \le h \le H$. Meanwhile, we denote the parameter of the DNN $u_\theta$ as $\theta = (\text{vec}(W_1)^\top, \dots, \text{vec}(W_H)^\top)^\top \in \mathbb{R}^{m_{\text{all}}}$ with $m_{\text{all}} = md + (H - 1)m^2$. We call $\{W_h\}_{h \in [H]}$ the weight matrices of $\theta$. Without loss of generality, we normalize the input $x$ such that $\|x\|_2 = 1$.

We initialize the DNN such that each entry of $W_h$ follows the standard Gaussian distribution $\mathcal{N}(0, 1)$ for any $h \in [H]$, while each entry of $b$ follows the uniform distribution $\text{Unif}(\{-1, 1\})$. Without loss of generality, we fix $b$ during training and only optimize $\{W_h\}_{h \in [H]}$. We denote the initialization of the parameter $\theta$ as $\theta_0 = (\text{vec}(W_1^0)^\top, \dots, \text{vec}(W_H^0)^\top)^\top$. Meanwhile, we restrict $\theta$ within the ball $\mathcal{B}(\theta_0, R)$ during training, which is defined as follows,

$$\mathcal{B}(\theta_0, R) = \{\theta \in \mathbb{R}^{m_{\text{all}}} \colon \|W_h - W_h^0\|_{\text{F}} \le R, \text{ for } h \in [H]\}. \qquad \text{(A.2)}$$

Here $\{W_h\}_{h \in [H]}$ and $\{W_h^0\}_{h \in [H]}$ are the weight matrices of $\theta$ and $\theta_0$, respectively. By (A.2), we have $\|\theta - \theta_0\|_2 \le R\sqrt{H}$ for any $\theta \in \mathcal{B}(\theta_0, R)$. Now, we define the family of DNNs as

$$\mathcal{U}(m, H, R) = \{u_\theta \colon \theta \in \mathcal{B}(\theta_0, R)\}, \qquad \text{(A.3)}$$

where $u_\theta$ is a DNN with depth $H$ and width $m$.

We parameterize the action-value function using $Q_\omega(s, a) \in \mathcal{U}(m_{\text{c}}, H_{\text{c}}, R_{\text{c}})$ and the energy function of the energy-based policy $\pi_\theta$ using $f_\theta(s, a) \in \mathcal{U}(m_{\text{a}}, H_{\text{a}}, R_{\text{a}})$. Here $\mathcal{U}(m_{\text{c}}, H_{\text{c}}, R_{\text{c}})$ and $\mathcal{U}(m_{\text{a}}, H_{\text{a}}, R_{\text{a}})$ are the families of DNNs defined in (A.3). Hereafter we assume that the energy function $f_\theta$ and the action-value function $Q_\omega$ share the same architecture and initialization, i.e., $m_{\text{a}} = m_{\text{c}}$, $H_{\text{a}} = H_{\text{c}}$, $R_{\text{a}} = R_{\text{c}}$, and $\theta_0 = \omega_0$. Such shared architecture and initialization of the DNNs ensure that the parameterizations of the policy and the action-value function are approximately compatible. See Sutton et al. (2000); Konda and Tsitsiklis (2000); Kakade (2002); Peters and Schaal (2008a); Wang et al. (2019) for a detailed discussion.

**Actor Update.** To solve (3.1), we use projected stochastic gradient descent, whose $n$-th iteration has the following form,

$\theta(n + 1)$
$$\leftarrow \Gamma_{\mathcal{B}(\theta_0, R_{\text{a}})}\big(\theta(n) - \alpha \cdot \big(f_{\theta(n)}(s, a) - \tau_{k+1} \cdot \big(\beta^{-1} Q_{\omega_k}(s, a) + \tau_k^{-1} f_{\theta_k}(s, a)\big)\big) \cdot \nabla_\theta f_{\theta(n)}(s, a)\big).$$

Here $\Gamma_{\mathcal{B}(\theta_0, R_{\text{a}})}$ is the projection operator, which projects the parameter onto the ball $\mathcal{B}(\theta_0, R_{\text{a}})$ defined in (A.2). The state-action pair $(s, a)$ is sampled from the stationary state-action distribution $\rho_k$. We summarize the update in Algorithm 3, which is deferred to §B of the appendix.

**Critic Update.** To solve (3.2), we apply projected stochastic gradient descent. More specifically, at the $n$-th iteration of projected stochastic gradient descent, we sample a tuple $(s, a, r, s', a')$, where $(s, a) \sim \rho_{k+1}$, $r = r(s, a)$, $s' \sim P(\cdot \mid s, a)$, and $a' \sim \pi_{\theta_{k+1}}(\cdot \mid s')$. We define the residual at the $n$-th iteration as $\delta(n) = Q_{\omega(n)}(s, a) - (1 - \gamma) \cdot r - \gamma \cdot Q_{\omega_k}(s', a')$. Then the $n$-th iteration of projected stochastic gradient descent has the following form,

$$\omega(n + 1) \leftarrow \Gamma_{\mathcal{B}(\omega_0, R_{\text{c}})}\big(\omega(n) - \eta \cdot \delta(n) \cdot \nabla_\omega Q_{\omega(n)}(s, a)\big).$$

Here $\Gamma_{\mathcal{B}(\omega_0, R_{\text{c}})}$ is the projection operator, which projects the parameter onto the ball $\mathcal{B}(\omega_0, R_{\text{c}})$ defined in (A.2). We summarize the update in Algorithm 4, which is deferred to §B of the appendix.

By assembling Algorithms 3 and 4, we present the deep neural actor-critic method in Algorithm 2, which is deferred to §B of the appendix.

Finally, we remark that the off-policy actor and critic updates given in (3.6) and (3.7) can also incorporate deep neural network approximation with a slight modification, which enables data reuse in the algorithm.

## B    DETAILS OF ALGORITHMS

In this section, we summarize the algorithms in §3. We first introduce the actor-critic method with linear function approximation in Algorithm 1.

---

**Algorithm 1** Linear Actor-Critic Method

---

**Input:** Number of iterations $K$, sample size $N$, temperature parameter $\beta$.
**Initialization:** Set $\tau_0 \leftarrow \infty$, and randomly initialize the actor parameter $\theta_0$ and the critic parameter $\omega_0$.
**for** $k = 0, 1, 2, \ldots, K$ **do**
    **Actor Update:** Update $\theta_{k+1}$ via (3.3) with $\tau_{k+1}^{-1} = (k+1) \cdot \beta^{-1}$.
    **Critic Update:** Sample $\{(s_{\ell,1}, a_{\ell,1})\}_{\ell \in [N]}$ and $\{(s_{\ell,2}, a_{\ell,2}, r_{\ell,2}, s'_{\ell,2}, a'_{\ell,2})\}_{\ell \in [N]}$ as specified in §3.1. Update $\omega_{k+1}$ via (3.5).
**end for**
**Output:** $\{\pi_{\theta_k}\}_{k \in [K+1]}$, where $\pi_{\theta_k} \propto \exp(\tau_k^{-1} f_{\theta_k})$.

---

We introduce the actor-critic method with DNN approximation in Algorithm 2, which relies on Algorithms 3 and 4 for the actor and critic updates.

---

**Algorithm 2** Deep Neural Actor-Critic Method

---

**Input:** Number of iterations $K$, $N_{\rm a}$, $N_{\rm c}$, stepsizes $\alpha, \eta$, and temperature parameter $\beta$.
**Initialization:** Set $\tau_0 \leftarrow \infty$ and initialize DNNs $f_{\theta_0}$ and $Q_{\omega_0}$ as specified in §A.
**for** $k = 0, 1, 2, \ldots, K$ **do**
    **Actor Update:** Update $\theta_{k+1}$ via Algorithm 3 with input $\pi_{\theta_k}, \theta_0, Q_{\omega_k}, \alpha, \beta, \tau_{k+1} = (k+1)^{-1} \cdot \beta$, and $N_{\rm a}$.
    **Critic Update:** Update $\omega_{k+1}$ via Algorithm 4 with input $\pi_{\theta_{k+1}}, Q_{\omega_k}, \omega_0, \eta$, and $N_{\rm c}$.
**end for**
**Output:** $\{\pi_{\theta_k}\}_{k \in [K+1]}$, where $\pi_{\theta_k} \propto \exp(\tau_k^{-1} f_{\theta_k})$.

---

**Algorithm 3** Actor Update for Deep Neural Actor-Critic Method

---

**Input:** Policy $\pi_\theta \propto \exp(\tau^{-1} f_\theta)$, initial actor parameter $\theta_0$, action-value function $Q_\omega$, stepsize $\alpha$, temperature parameter $\beta$, temperature $\widetilde{\tau}$, and number of iterations $N_{\rm a}$.
**Initialization:** Set $\theta(0) \leftarrow \theta_0$.
**for** $n = 0, 1, 2, \ldots, N_{\rm a} - 1$ **do**
    Sample $(s, a)$ as specified in §A.
    Set $\theta(n+1) \leftarrow \Gamma_{\mathcal{B}(\theta_0, R_{\rm a})}(\theta(n) - \alpha \cdot (f_{\theta(n)}(s, a) - \widetilde{\tau} \cdot (\beta^{-1} Q_\omega(s, a) + \tau^{-1} f_\theta(s, a))) \cdot \nabla_\theta f_{\theta(n)}(s, a))$.
**end for**
**Output:** $\overline{\theta} = 1/N_{\rm a} \cdot \sum_{n=1}^{N_{\rm a}} \theta(n)$.

---

**Algorithm 4** Critic Update for Deep Neural Actor-Critic Method

---

**Input:** Policy $\pi_\theta$, action-value function $Q_\omega$, initial critic parameter $\omega_0$, stepsize $\eta$, and number of iterations $N_{\rm c}$.
**Initialization:** Set $\omega(0) \leftarrow \omega_0$.
**for** $n = 0, 1, 2, \ldots, N_{\rm c} - 1$ **do**
    Sample $(s, a, r, s', a')$ as specified in §A.
    Set $\delta(n) \leftarrow Q_{\omega(n)}(s, a) - (1 - \gamma) \cdot r - \gamma \cdot Q_\omega(s', a')$.
    Set $\omega(n+1) \leftarrow \Gamma_{\mathcal{B}(\omega_0, R_{\rm c})}(\omega(n) - \eta \cdot \delta(n) \cdot \nabla_\omega Q_{\omega(n)}(s, a))$.
**end for**
**Output:** $\overline{\omega} = 1/N_{\rm c} \cdot \sum_{n=1}^{N_{\rm c}} \omega(n)$.

---

## C  CONVERGENCE RESULTS OF ALGORITHM 2

In this section, we upper bound the regret of the deep neural actor-critic method. Hereafter we assume that $|r(s, a)| \leq r_{\max}$ for any $(s, a) \in \mathcal{S} \times \mathcal{A}$, where $r_{\max}$ is a positive absolute constant. First, we impose the following assumptions in parallel to Assumption 4.1. Recall that $\rho^*$ is the stationary state-action distribution of $\pi^*$, while $\rho_k$ is the stationary state-action distribution of $\pi_{\theta_k}$.

**Assumption C.1** (Concentrability Coefficient). The following statements hold.

(i) There exists a positive absolute constant $\phi^*$ such that $\phi_k^* \leq \phi^*$ for any $k \geq 1$, where $\phi_k^* = \|\mathrm{d}\rho^*/\mathrm{d}\rho_k\|_{\rho_k, 2}$.

(ii) For the state-action distribution $\rho$ used to define the regret in (4.1), we assume that for any $k \geq 1$ and a sequence of policies $\{\pi_i\}_{i \geq 1}$, the $k$-step future-state-action distribution $\rho \mathbb{P}^{\pi_1} \cdots \mathbb{P}^{\pi_k}$ is absolutely continuous with respect to $\rho^*$. Also, it holds that

$$C_{\rho, \rho^*} = (1 - \gamma)^2 \sum_{k=1}^{\infty} k^3 \gamma^k \cdot c(k) < \infty,$$

where $c(k) = \sup_{\{\pi_i\}_{i \in [k]}} \|\mathrm{d}(\rho \mathbb{P}^{\pi_1} \cdots \mathbb{P}^{\pi_k})/\mathrm{d}\rho^*\|_{\rho^*, \infty}$.

Meanwhile, we impose the following assumption in parallel to Assumption 4.2.

**Assumption C.2** (Zero Approximation Error). For any $Q_\omega \in \mathcal{U}(m_c, H_c, R_c)$ and policy $\pi$, it holds that $\mathbb{T}^\pi Q_\omega \in \mathcal{U}(m_c, H_c, R_c)$, where $\mathbb{T}^\pi$ is defined in (2.4).

Assumption C.2 states that $\mathcal{U}(m_c, H_c, R_c)$ is closed under the Bellman evaluation operator $\mathbb{T}^\pi$, which is commonly imposed in the literature (Munos and Szepesvári, 2008; Antos et al., 2008a; Farahmand et al., 2010; 2016; Tosatto et al., 2017; Yang et al., 2019b; Liu et al., 2019).

We upper bound the regret of the deep neural actor-critic method in Algorithm 2 in the sequel. To establish such an upper bound, we first establish the rates of convergence of Algorithms 3 and 4 as follows.

**Proposition C.3.** For any sufficiently large $N_a > 0$, let $m_a = \Omega(d^{3/2} R_a^{-1} H_a^{-3/2} \log(m_a^{1/2}/R_a)^{3/2})$, $H_a = O(N_a^{1/4})$, and $R_a = O(m_a^{1/2} H_a^{-6} (\log m_a)^{-3})$. We denote by $\overline{\theta}$ the output of Algorithm 3 with input $\pi_\theta \propto \exp(\tau^{-1} f_\theta)$, $\theta_0$, $Q_\omega$, $\alpha$, $\beta$, $\widetilde{\tau} = (\tau^{-1} + \beta^{-1})^{-1}$, and $N_a$. Also, let $\widetilde{f} = \widetilde{\tau} \cdot (\beta^{-1} Q_\omega + \tau^{-1} f_\theta)$. With probability at least $1 - \exp(-\Omega(R_a^{2/3} m_a^{2/3} H_a))$ over the random initialization $\theta_0$, we have

$$\mathbb{E}\big[\big(f_{\overline{\theta}}(s, a) - \widetilde{f}(s, a)\big)^2\big] = O(R_a^2 N_a^{-1/2} + R_a^{8/3} m_a^{-1/6} H_a^7 \log m_a).$$

Here the expectation is taken over the randomness of $\overline{\theta}$ conditioning on the initialization $\theta_0$ and $(s, a) \sim \rho_{\pi_\theta}$, where $\rho_{\pi_\theta}$ is the stationary state-action distribution of $\pi_\theta$.

*Proof.* See §G.2 for a detailed proof. □

**Proposition C.4.** For any sufficiently large $N_c > 0$, let $m_c = \Omega(d^{3/2} R_c^{-1} H_c^{-3/2} \log(m_c^{1/2}/R_c)^{3/2})$, $H_c = O(N_c^{1/4})$, and $R_c = O(m_c^{1/2} H_c^{-6} (\log m_c)^{-3})$. We denote by $\overline{\omega}$ the output of Algorithm 4 with input $\pi_\theta$, $Q_\omega$, $\omega_0$, $\eta$, and $N_c$. Also, let $\widetilde{Q} = (1 - \gamma) \cdot r + \gamma \cdot \mathbb{P}^{\pi_\theta} Q_\omega$. With probability at least $1 - \exp(-\Omega(R_c^{2/3} m_c^{2/3} H_c))$ over the random initialization $\omega_0$, we have

$$\mathbb{E}\big[\big(Q_{\overline{\omega}}(s, a) - \widetilde{Q}(s, a)\big)^2\big] = O(R_c^2 N_c^{-1/2} + R_c^{8/3} m_c^{-1/6} H_c^7 \log m_c).$$

Here the expectation is taken over the randomness of $\overline{\omega}$ conditioning on the initialization $\omega_0$ and $(s, a) \sim \rho_{\pi_\theta}$, where $\rho_{\pi_\theta}$ is the stationary state-action distribution of $\pi_\theta$.

*Proof.* See §G.3 for a detailed proof. □

Propositions C.3 and C.4 characterize the errors that arise from the actor and critic updates in Algorithm 2, respectively. In particular, if the widths $m_a$ and $m_c$ of the DNNs $f_\theta$ and $Q_\omega$ are sufficiently large, the errors characterized in Propositions C.3 and C.4 decay to zero at the rates of $O(N_a^{-1/2})$ and $O(N_c^{-1/2})$, respectively. Propositions C.3 and C.4 act as the key ingredients to upper bounding the regret of the deep neural actor-critic method.

Based on Propositions C.3 and C.4, we upper bound the regret of Algorithm 2 in the following theorem, which is in parallel to Theorem 4.4.

**Theorem C.5.** We assume that Assumptions C.1 and C.2 hold. Let $\rho$ be a state-action distribution satisfying (ii) of Assumption C.1. Also, for any sufficiently large $K > 0$, let $N_a = \Omega(K^2 C_{\rho,\rho^*}^4 (\phi^* + \psi^* + 1)^4 R_a^4)$, $N_c = \Omega(K^2 C_{\rho,\rho^*}^4 \phi^{*4} R_c^4)$, $H_a = H_c = O(N_c^{1/4})$, $R_a = R_c = O(m_c^{1/2} H_c^{-6} (\log m_c)^{-3})$, $m_a = m_c = \Omega(d^{3/2} K^6 C_{\rho,\rho^*}^{12} (\phi^* + \psi^* + 1)^{12} R_c^{16} H_c^{42} \log(m_c^{1/2}/R_c)^{3/2})$, $\beta = K^{1/2}$, and the sequence $\{\theta_k\}_{k \in [K]}$ be generated by Algorithm 2. With probability at least $1 - 1/K$ over the random initialization $\theta_0$ and $\omega_0$, it holds that

$$\mathbb{E}\Big[\sum_{k=0}^{K} Q^*(s,a) - Q^{\pi_{\theta_{k+1}}}(s,a)\Big] \leq \big(2(1-\gamma)^{-3} \log |\mathcal{A}| + O(1)\big) \cdot K^{1/2},$$

where the expectation is taken over the randomness of $(s,a) \sim \rho$ and $\{\theta_{k+1}\}_{k \in [K]}$ conditioning on the initialization $\theta_0$ and $\omega_0$.

*Proof.* See §E.2 for a detailed proof. □

When the architecture of the actor and critic neural networks are properly chosen, Theorem C.5 establishes an $O(K^{1/2})$ regret of Algorithm 2, where $K$ is the total number of iterations. Specifically speaking, to establish such a regret upper bound, we need the widths $m_a$ and $m_c$ of the DNNs $f_\theta$ and $Q_\omega$ to be sufficiently large. Meanwhile, to control the errors of actor update and critic update in Algorithm 2, we also run sufficiently large numbers of iterations in Algorithms 3 and 4.

In terms of the total sample complexity, to simplify our discussion, we omit constant and logarithmic terms here. To obtain an $\varepsilon$-globally optimal policy, it suffices to set $K \asymp \varepsilon^{-2}$ in Algorithm 2. By plugging such a $K$ into $N_a = \Omega(K^2 C_{\rho,\rho^*}^4 (\phi^* + \psi^* + 1)^4 R_a^4)$ and $N_c = \Omega(K^2 C_{\rho,\rho^*}^4 \phi^{*4} R_c^4)$ as required in Theorem C.5, we have $N_a = \widetilde{O}(\varepsilon^{-4})$ and $N_c = \widetilde{O}(\varepsilon^{-4})$. Thus, to achieve an $\varepsilon$-globally optimal policy, the total sample complexity of Algorithm 2 is $\widetilde{O}(\varepsilon^{-6})$. With the modification to off-policy setting as in §3.1, the total sample complexity of Algorithm 2 is $\widetilde{O}(\varepsilon^{-4})$. In comparison, Liu et al. (2019) requires a total sample complexity of $\widetilde{O}(\varepsilon^{-8})$ to achieve an $\varepsilon$-globally optimal policy, which is worse than our single-timescale algorithm. Meanwhile, since Liu et al. (2019) uses TD(0) in the critic update, which is shown to diverge under off-policy setting even with linear function approximation (Baird et al., 1995), the method of data reuse cannot be applied to Liu et al. (2019) to eliminate the total sample complexity.

To the best of our knowledge, we establish the rate of convergence and global optimality of the actor-critic method under single-timescale setting with DNN approximation for the first time.

## D PROOF SKETCH OF MAIN THEOREM 4.4

In this section, we sketch the proof of Theorem 4.4. Recall that $\rho$ is a state-action distribution satisfying (ii) of Assumption 4.1. We first upper bound $\sum_{k=0}^{K}(Q^*(s,a) - Q^{\pi_{\theta_{k+1}}}(s,a))$ for any $(s,a) \in \mathcal{S} \times \mathcal{A}$ in part 1. Then by further taking the expectation over $\rho$ in part 2, we conclude the proof of Theorem 4.4. See §E.1 for a detailed proof.

**Part 1.** In the sequel, we upper bound $\sum_{k=0}^{K}(Q^*(s,a) - Q^{\pi_{\theta_{k+1}}}(s,a))$ for any $(s,a) \in \mathcal{S} \times \mathcal{A}$. We first decompose $Q^* - Q^{\pi_{\theta_{k+1}}}$ into the following three terms,

$$\sum_{k=0}^{K} [Q^* - Q^{\pi_{\theta_{k+1}}}](s,a) = \sum_{k=0}^{K} \big[(I - \gamma \mathbb{P}^{\pi^*})^{-1}(A_{1,k} + A_{2,k} + A_{3,k})\big](s,a), \quad \text{(D.1)}$$

the proof of which is deferred to (E.1) and (E.2) in §E.1 of the appendix. Here the operator $\mathbb{P}^{\pi^*}$ is defined in (2.3), $(I - \gamma\mathbb{P}^{\pi^*})^{-1} = \sum_{i=0}^{\infty}(\gamma\mathbb{P}^{\pi^*})^i$, and $A_{1,k}$, $A_{2,k}$, and $A_{3,k}$ are defined as follows,

$$A_{1,k}(s,a) = [\gamma(\mathbb{P}^{\pi^*} - \mathbb{P}^{\pi_{\theta_{k+1}}})Q_{\omega_k}](s,a), \tag{D.2}$$

$$A_{2,k}(s,a) = \left[\gamma\mathbb{P}^{\pi^*}(Q^{\pi_{\theta_{k+1}}} - Q_{\omega_k})\right](s,a), \tag{D.3}$$

$$A_{3,k}(s,a) = [\mathbb{T}^{\pi_{\theta_{k+1}}}Q_{\omega_k} - Q^{\pi_{\theta_{k+1}}}](s,a). \tag{D.4}$$

To understand the intuition behind $A_{1,k}$, $A_{2,k}$, and $A_{3,k}$, we interpret them as follows.

**Interpretation of $A_{1,k}$.** As defined in (D.2), $A_{1,k}$ arises from the actor update and measures the convergence of the policy $\pi_{\theta_{k+1}}$ towards a globally optimal policy $\pi^*$, which implies the convergence of $\mathbb{P}^{\pi_{\theta_{k+1}}}$ towards $\mathbb{P}^{\pi^*}$.

**Interpretation of $A_{3,k}$.** Note that by (2.2) and (2.4), we have $Q^{\pi_{\theta_{k+1}}} = \mathbb{T}^{\pi_{\theta_{k+1}}}Q^{\pi_{\theta_{k+1}}}$ and $\mathbb{T}^{\pi_{\theta_{k+1}}}$ is a $\gamma$-contraction, which implies that applying the Bellman evaluation operator $\mathbb{T}^{\pi_{\theta_{k+1}}}$ to any $Q$, e.g., $Q_{\omega_k}$, infinite times yields $Q^{\pi_{\theta_{k+1}}}$. As defined in (D.4), $A_{3,k}$ measures the error of tracking the action-value function $Q^{\pi_{\theta_{k+1}}}$ of $\pi_{\theta_{k+1}}$ by applying the Bellman evaluation operator $\mathbb{T}^{\pi_{\theta_{k+1}}}$ to $Q_{\omega_k}$ only once, which arises from the critic update. Also, as $A_{3,k} = \mathbb{T}^{\pi_{\theta_{k+1}}}(Q_{\omega_k} - Q^{\pi_{\theta_{k+1}}})$, $A_{3,k}$ measures the difference between $Q^{\pi_{\theta_k}}$, which is approximated by $Q_{\omega_k}$ as discussed subsequently, and $Q^{\pi_{\theta_{k+1}}}$. Such a difference can also be viewed as the difference between $\pi_{\theta_k}$ and $\pi_{\theta_{k+1}}$, which arises from the actor update. Therefore, the convergence of $A_{3,k}$ to zero implies the contractions of not only the critic update but also the actor update, which illustrates the "double contraction" phenomenon. We establish the convergence of $A_{3,k}$ to zero in (D.10) subsequently.

**Interpretation of $A_{2,k}$.** Assuming that $A_{3,k-1}$ converges to zero, we have $\mathbb{T}^{\pi_{\theta_k}}Q_{\omega_{k-1}} \approx Q^{\pi_{\theta_k}}$. Moreover, assuming that the number of data points $N$ is sufficiently large and ignoring the projection in (3.5), we have $\mathbb{T}^{\pi_{\theta_k}}Q_{\omega_{k-1}} = Q_{\widetilde{\omega}_k} \approx Q_{\omega_k}$ as $\widetilde{\omega}_k$ defined in (3.4) is an estimator of $\omega_k$. Hence, we have $Q^{\pi_{\theta_k}} \approx Q_{\omega_k}$. Such an approximation error is characterized by $\epsilon_k^c$ defined in (D.5) subsequently. Hence, $A_{2,k}$ measures the difference between $\pi_{\theta_k}$ and $\pi_{\theta_{k+1}}$ through the difference between $Q^{\pi_{\theta_k}} \approx Q_{\omega_k}$ and $Q^{\pi_{\theta_{k+1}}}$, which relies on the convergence of $A_{3,k-1}$ to zero.

In the sequel, we upper bound $A_{1,k}$, $A_{2,k}$, and $A_{3,k}$, respectively. To establish such upper bounds, we define the following quantities,

$$\epsilon_{k+1}^c(s,a) = [\mathbb{T}^{\pi_{\theta_{k+1}}}Q_{\omega_k} - Q_{\omega_{k+1}}](s,a), \tag{D.5}$$

$$e_{k+1}(s,a) = [Q_{\omega_k} - \mathbb{T}^{\pi_{\theta_{k+1}}}Q_{\omega_k}](s,a), \tag{D.6}$$

$$\vartheta_k(s) = \mathrm{KL}\big(\pi^*(\cdot \,|\, s) \,\|\, \pi_{\theta_k}(\cdot \,|\, s)\big) - \mathrm{KL}\big(\pi^*(\cdot \,|\, s) \,\|\, \pi_{\theta_{k+1}}(\cdot \,|\, s)\big). \tag{D.7}$$

To understand the intuition behind $\epsilon_{k+1}^c$, $e_{k+1}$, and $\vartheta_k$, we interpret them as follows.

**Interpretation of $\epsilon_{k+1}^c$.** Recall that $\widetilde{\omega}_{k+1}$ is defined in (3.4), which parameterizes $\mathbb{T}^{\pi_{\theta_{k+1}}}Q_{\omega_k}$ (ignoring the projection in (3.5)). Here $\epsilon_{k+1}^c$ arises from approximating $\widetilde{\omega}_{k+1}$ using $\omega_{k+1}$ as an estimator, which is constructed based on $\omega_k$ and the $N$ data points. In particular, $\epsilon_{k+1}^c$ decreases to zero as $N \to \infty$, which is used in characterizing $A_{2,k}$ defined in (D.3).

**Interpretation of $e_{k+1}$.** Assuming that $A_{3,k-1}$ defined in (D.4) and $\epsilon_k^c$ defined in (D.5) converge to zero, which implies $\mathbb{T}^{\pi_{\theta_k}}Q_{\omega_{k-1}} \approx Q^{\pi_{\theta_k}}$ and $\mathbb{T}^{\pi_{\theta_k}}Q_{\omega_{k-1}} \approx Q_{\omega_k}$, respectively, we have $Q_{\omega_k} \approx Q^{\pi_{\theta_k}}$. Therefore, as defined in (D.6), $e_{k+1} = Q_{\omega_k} - \mathbb{T}^{\pi_{\theta_{k+1}}}Q_{\omega_k} \approx Q^{\pi_{\theta_k}} - \mathbb{T}^{\pi_{\theta_{k+1}}}Q^{\pi_{\theta_k}} = (\mathbb{T}^{\pi_{\theta_k}} - \mathbb{T}^{\pi_{\theta_{k+1}}})Q^{\pi_{\theta_k}}$ measures the difference between $\pi_{\theta_k}$ and $\pi_{\theta_{k+1}}$, which implies the difference between $\mathbb{T}^{\pi_{\theta_k}}$ and $\mathbb{T}^{\pi_{\theta_{k+1}}}$. We remark that $e_{k+1}$ fully characterizes $A_{3,k}$ defined in (D.4) as shown in (D.8) subsequently.

**Interpretation of $\vartheta_k$.** As defined in (D.7), $\vartheta_k$ measures the difference between $\pi_{\theta_k}$ and $\pi_{\theta_{k+1}}$ in terms of their differences with $\pi^*$, which are measured by the corresponding KL-divergences. In particular, $\vartheta_k$ is used in characterizing $A_{1,k}$ and $A_{2,k}$ defined in (D.2) and (D.3), respectively.

We remark that $\epsilon_{k+1}^c$ measures the statistical error in the critic update, while $\vartheta_k$ measures the optimization error in the actor update. As discussed above, the convergence of $A_{3,k}$ to zero implies the

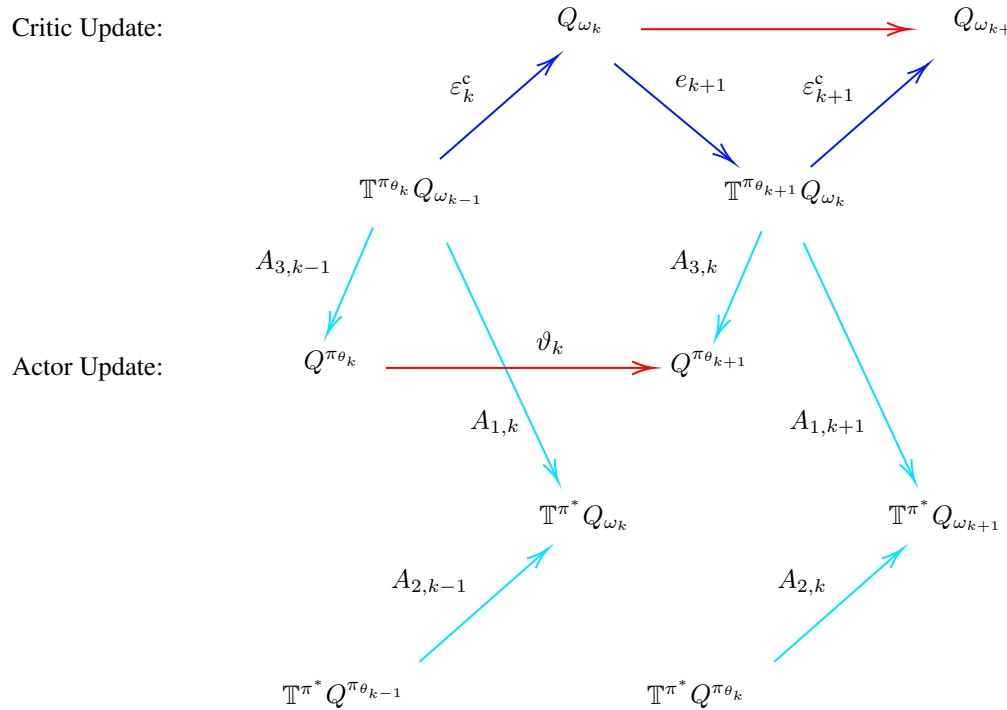

Figure 1: Illustration of the relationship among $A_{1,k}$, $A_{2,k}$, $A_{3,k}$, $\epsilon_{k+1}^{\mathrm{c}}$, $e_{k+1}$, and $\vartheta_k$. Here $\{\theta_k, \omega_k\}$ and $\{\theta_{k+1}, \omega_{k+1}\}$ are two consecutive iterates of actor-critic. The red arrow from $Q_{\omega_k}$ to $Q_{\omega_{k+1}}$ represents the critic update and the red arrow from $Q^{\pi_{\theta_k}}$ to $Q^{\pi_{\theta_{k+1}}}$ represents the action-value functions associated with the two policies in any actor update. Here $\vartheta_k$ given in (D.7) quantifies the difference between $\pi_{\theta_k}$ and $\pi_{\theta_{k+1}}$ in terms of their KL distances to $\pi^*$. In addition, the cyan arrows represent quantities $A_{1,k}$, $A_{2,k}$, and $A_{3,k}$ introduced in (D.2)–(D.4), which are intermediate terms used for analyzing the error $Q^* - Q^{\pi_{k+1}}$. Finally, the blue arrows represent $\varepsilon_{k+1}^{\mathrm{c}}$ and $e_{k+1}$ defined in (D.5) and (D.6), respectively. Here $\varepsilon_{k+1}^{\mathrm{c}}$ corresponds to the statistical error due to having finite data whereas $e_{k+1}$ essentially quantifies the difference between $\pi_{\theta_k}$ and $\pi_{\theta_{k+1}}$.

contraction of both the actor update and the critic update, which illustrates the "double contraction" phenomenon. Meanwhile, since $e_{k+1}$ fully characterizes $A_{3,k}$ as shown in (D.8) subsequently, $e_{k+1}$ plays a key role in the "double contraction" phenomenon. In particular, the convergence of $e_{k+1}$ to zero is established in (D.9) subsequently. See Figure 1 for an illustration of these quantities.

With the quantities defined in (D.5), (D.6), and (D.7), we upper bound $A_{1,k}$, $A_{2,k}$, and $A_{3,k}$ as follows,

$$A_{1,k}(s,a) \leq \gamma\beta \cdot [\mathbb{P}\vartheta_k](s,a),$$

$$A_{2,k}(s,a) \leq \big[(\gamma\mathbb{P}^{\pi^*})^{k+1}(Q^* - Q_{\omega_0})\big](s,a) + \gamma\beta \cdot \sum_{i=0}^{k-1}\big[(\gamma\mathbb{P}^{\pi^*})^{k-i}\mathbb{P}\vartheta_i\big](s,a)$$

$$+ \sum_{i=0}^{k-1}\big[(\gamma\mathbb{P}^{\pi^*})^{k-i}\epsilon_{i+1}^{\mathrm{c}}\big](s,a),$$

$$A_{3,k}(s,a) = \big[\gamma\mathbb{P}^{\pi_{\theta_{k+1}}}(I - \gamma\mathbb{P}^{\pi_{\theta_{k+1}}})^{-1}e_{k+1}\big](s,a), \tag{D.8}$$

the proof of which is deferred to Lemmas E.1, E.2, and E.3 in §E.1 of the appendix, respectively. Meanwhile, by recursively expanding (D.5) and (D.6), we have

$$e_{k+1}(s,a) \leq \Big[\gamma^k\Big(\prod_{s=1}^{k}\mathbb{P}^{\pi_{\theta_s}}\Big)e_1 + \sum_{i=1}^{k}\gamma^{k-i}\Big(\prod_{s=i+1}^{k}\mathbb{P}^{\pi_{\theta_s}}\Big)(I - \gamma\mathbb{P}^{\pi_{\theta_i}})\epsilon_i^{\mathrm{c}}\Big](s,a), \tag{D.9}$$

the proof of which is deferred to Lemma E.4 in §E.1 of the appendix. By plugging (D.9) into (D.8), we have

$$
A_{3,k}(s,a) \leq \left[ \gamma \mathbb{P}^{\pi_{\theta_{k+1}}} (I - \gamma \mathbb{P}^{\pi_{\theta_{k+1}}})^{-1} \left( \gamma^k \Big( \prod_{s=1}^{k} \mathbb{P}^{\pi_{\theta_s}} \Big) e_1 \right. \right. \tag{D.10}
$$
$$
\left. \left. + \sum_{i=1}^{k} \gamma^{k-i} \Big( \prod_{s=i+1}^{k} \mathbb{P}^{\pi_{\theta_s}} \Big) (I - \gamma \mathbb{P}^{\pi_{\theta_i}}) \epsilon_i^{\mathrm{c}} \right) \right] (s,a).
$$

To better understand (D.10) and how it relates to the convergence of $A_{3,k}$, $A_{2,k}$, and $A_{1,k}$ to zero, we discuss in the following two steps.

**Step (i).** We assume $\epsilon_i^{\mathrm{c}} = 0$, which corresponds to the number of data points $N \to \infty$. Then (D.10) yields $A_{3,k} = O(\gamma^k)$, which implies that $A_{3,k}$ defined in (D.4) converges to zero driven by the discount factor $\gamma$. As discussed above, the convergence of $A_{3,k}$ to zero also implies the contraction between $\pi_{\theta_k}$ and $\pi_{\theta_{k+1}}$ of the actor update and the contraction between $Q_{\omega_k}$ and $Q^{\pi_{\theta_k}}$ of the critic update, which illustrates the "double contraction" phenomenon.

**Step (ii).** The convergence of $A_{3,k}$ to zero further ensures that $A_{2,k}$ converges to zero. To see this, we further assume $A_{3,k} = 0$, which together with the assumption that $\epsilon_{k+1}^{\mathrm{c}} = 0$ implies $Q^{\pi_{\theta_{k+1}}} = \mathbb{T}^{\pi_{\theta_{k+1}}} Q_{\omega_k} = Q_{\omega_{k+1}}$ by their definitions in (D.4) and (D.5), respectively. Then by telescoping the sum of $A_{2,k}$ defined in (D.3), which cancels out $Q_{\omega_{k+1}}$ and $Q^{\pi_{\theta_{k+1}}}$, we obtain the convergence of $A_{2,k}$ to zero. Meanwhile, telescoping the sum of $A_{1,k}$ defined in (D.2) and the sum of its upper bound in (D.8) implies that $A_{1,k}$ converges to zero.

Now, by plugging (D.8) and (D.10) into (D.1), we establish an upper bound of $\sum_{k=0}^{K}(Q^*(s,a) - Q^{\pi_{\theta_{k+1}}}(s,a))$ for any $(s,a) \in \mathcal{S} \times \mathcal{A}$, which is deferred to (E.12) in §E.1 of the appendix. Hence, we conclude the proof in part 1. See part 1 of §E.1 for details.

**Part 2.** Recall that $\rho$ is a state-action distribution satisfying (ii) of Assumption 4.1. In the sequel, we take the expectation over $\rho$ in (E.12) and upper bound each term. We first introduce the following lemma, which upper bounds $\epsilon_{k+1}^{\mathrm{c}}$ defined in (D.5).

**Lemma D.1.** Under Assumptions 4.2 and 4.3, it holds for any $k \geq 1$ that

$$
\mathbb{E}\big[\epsilon_{k+1}^{\mathrm{c}}(s,a)^2\big] = \mathbb{E}\big[\big(Q_{\omega_{k+1}}(s,a) - [\mathbb{T}^{\pi_{\theta_k}} Q_{\omega_k}](s,a)\big)^2\big] \leq \frac{16(r_{\max} + R)^2}{N \sigma^{*4}} \cdot \log(N + d)^2,
$$

where the expectation is taken with respect to randomness of $\omega_{k+1}$ and $(s,a) \sim \rho_{k+1}$.

*Proof.* See §H.1 for a detailed proof. $\qquad \square$

On the right-hand side of (E.12) in §E.1 of the appendix, for the terms not involving $\epsilon_{k+1}^{\mathrm{c}}$, i.e., $M_1$, $M_2$, and $M_3$ in (E.13), we take the expectation over $\rho$ and establish their upper bounds in the $\ell_\infty$-norm over $(s,a)$ in Lemma E.5. On the other hand, for the terms involving $\epsilon_{k+1}^{\mathrm{c}}$, i.e., $M_4$ and $M_5$ in (E.14), we take the expectation over $\rho$ and then change the measure from $\rho$ to $\rho_{k+1}$. By Assumption 4.1 and Lemma D.1, which relies on $\rho_{k+1}$, we establish the upper bounds in Lemma E.6. See part 2 of §E.1 for details.

Combining Lemmas E.5 and E.6 yields Theorem 4.4. See §E.1 for a detailed proof.

# E  PROOFS OF THEOREMS

## E.1  PROOF OF THEOREM 4.4

Recall that $\rho$ is a state-action distribution satisfying (ii) of Assumption 4.1. We first upper bound $\sum_{k=0}^{K}(Q^*(s,a) - Q^{\pi_{\theta_{k+1}}}(s,a))$ for any $(s,a) \in \mathcal{S} \times \mathcal{A}$ in part 1. Then by further taking the expectation over $\rho$ and invoking Lemma D.1 in part 2, we conclude the proof of Theorem 4.4.

**Part 1.** In the sequel, we upper bound $\sum_{k=0}^{K}(Q^*(s,a) - Q^{\pi_{\theta_{k+1}}}(s,a))$ for any $(s,a) \in \mathcal{S} \times \mathcal{A}$. By the definition of $Q^*$ in (2.2), it holds for any $(s,a) \in \mathcal{S} \times \mathcal{A}$ that

$$
[Q^* - Q^{\pi_{\theta_{k+1}}}](s,a)
$$

$$
= \sum_{\ell=0}^{\infty} \left[ (1-\gamma) \cdot (\gamma \mathbb{P}^{\pi^*})^{\ell} r \right](s,a) - Q^{\pi_{\theta_{k+1}}}(s,a)
$$

$$
= \sum_{\ell=0}^{\infty} \left[ (1-\gamma) \cdot (\gamma \mathbb{P}^{\pi^*})^{\ell} r + (\gamma \mathbb{P}^{\pi^*})^{\ell+1} Q^{\pi_{\theta_{k+1}}} - (\gamma \mathbb{P}^{\pi^*})^{\ell+1} Q^{\pi_{\theta_{k+1}}} \right](s,a) - Q^{\pi_{\theta_{k+1}}}(s,a)
$$

$$
= \sum_{\ell=0}^{\infty} \left[ (1-\gamma) \cdot (\gamma \mathbb{P}^{\pi^*})^{\ell} r + (\gamma \mathbb{P}^{\pi^*})^{\ell+1} Q^{\pi_{\theta_{k+1}}} - (\gamma \mathbb{P}^{\pi^*})^{\ell} Q^{\pi_{\theta_{k+1}}} \right](s,a)
$$

$$
= \sum_{\ell=0}^{\infty} \left[ (\gamma \mathbb{P}^{\pi^*})^{\ell} \big( (1-\gamma) \cdot r + \gamma \cdot \mathbb{P}^{\pi^*} Q^{\pi_{\theta_{k+1}}} - Q^{\pi_{\theta_{k+1}}} \big) \right](s,a), \tag{E.1}
$$

where $\mathbb{P}^{\pi^*}$ is defined in (2.3). We upper bound $[(1-\gamma) \cdot r + \gamma \cdot \mathbb{P}^{\pi^*} Q^{\pi_{\theta_{k+1}}} - Q^{\pi_{\theta_{k+1}}}](s,a)$ on the RHS of (E.1) in the sequel. By calculation, we have

$$
\left[ (1-\gamma) \cdot r + \gamma \cdot \mathbb{P}^{\pi^*} Q^{\pi_{\theta_{k+1}}} - Q^{\pi_{\theta_{k+1}}} \right](s,a)
$$

$$
= \left[ \big( (1-\gamma) \cdot r + \gamma \cdot \mathbb{P}^{\pi^*} Q^{\pi_{\theta_{k+1}}} \big) - \big( (1-\gamma) \cdot r + \gamma \cdot \mathbb{P}^{\pi^*} Q_{\omega_k} \big) \right](s,a)
$$

$$
+ \left[ \big( (1-\gamma) \cdot r + \gamma \cdot \mathbb{P}^{\pi^*} Q_{\omega_k} \big) - \big( (1-\gamma) \cdot r + \gamma \cdot \mathbb{P}^{\pi_{\theta_{k+1}}} Q_{\omega_k} \big) \right](s,a)
$$

$$
+ \left[ \big( (1-\gamma) \cdot r + \gamma \cdot \mathbb{P}^{\pi_{\theta_{k+1}}} Q_{\omega_k} \big) - Q^{\pi_{\theta_{k+1}}} \right](s,a)
$$

$$
= A_{1,k}(s,a) + A_{2,k}(s,a) + A_{3,k}(s,a), \tag{E.2}
$$

where $A_{1,k}$, $A_{2,k}$, and $A_{3,k}$ are defined as follows,

$$
A_{1,k}(s,a) = \left[ \gamma (\mathbb{P}^{\pi^*} - \mathbb{P}^{\pi_{\theta_{k+1}}}) Q_{\omega_k} \right](s,a),
$$

$$
A_{2,k}(s,a) = \left[ \gamma \mathbb{P}^{\pi^*} (Q^{\pi_{\theta_{k+1}}} - Q_{\omega_k}) \right](s,a),
$$

$$
A_{3,k}(s,a) = [\mathbb{T}^{\pi_{\theta_{k+1}}} Q_{\omega_k} - Q^{\pi_{\theta_{k+1}}}](s,a). \tag{E.3}
$$

Here $\mathbb{T}^{\pi_{\theta_{k+1}}}$ is defined in (2.4). By the following three lemmas, we upper bound $A_{1,k}$, $A_{2,k}$, and $A_{3,k}$ on the RHS of (E.2), respectively.

**Lemma E.1.** It holds for any $(s,a) \in \mathcal{S} \times \mathcal{A}$ that

$$
A_{1,k}(s,a) = \left[ \gamma (\mathbb{P}^{\pi^*} - \mathbb{P}^{\pi_{\theta_{k+1}}}) Q_{\omega_k} \right](s,a) \leq \left[ \gamma \beta \cdot \mathbb{P}(\vartheta_k + \epsilon^{\mathrm{a}}_{k+1}) \right](s,a),
$$

where $\vartheta_k$ and $\epsilon^{\mathrm{a}}_{k+1}$ are defined as follows,

$$
\vartheta_k(s) = \mathrm{KL}\big(\pi^*(\cdot \mid s) \,\|\, \pi_{\theta_k}(\cdot \mid s)\big) - \mathrm{KL}\big(\pi^*(\cdot \mid s) \,\|\, \pi_{\theta_{k+1}}(\cdot \mid s)\big), \tag{E.4}
$$

$$
\epsilon^{\mathrm{a}}_{k+1}(s) = \big\langle \log\big(\pi_{\theta_{k+1}}(\cdot \mid s)/\pi_{\theta_k}(\cdot \mid s)\big) - \beta^{-1} \cdot Q_{\omega_k}(s,\cdot), \pi^*(\cdot \mid s) - \pi_{\theta_{k+1}}(\cdot \mid s) \big\rangle. \tag{E.5}
$$

*Proof.* See §H.2 for a detailed proof. $\qquad \square$

We remark that $\epsilon^{\mathrm{a}}_{k+1} = 0$ for any $k$ in the linear actor-critic method. Meanwhile, such a term is included in Lemma E.1 only aiming to generalize to the deep neural actor-critic method.

**Lemma E.2.** It holds for any $(s,a) \in \mathcal{S} \times \mathcal{A}$ that

$$
A_{2,k}(s,a) \leq \left[ (\gamma \mathbb{P}^{\pi^*})^{k+1} (Q^* - Q_{\omega_0}) \right](s,a) + \gamma \beta \cdot \sum_{i=0}^{k-1} \left[ (\gamma \mathbb{P}^{\pi^*})^{k-i} \mathbb{P}(\vartheta_i + \epsilon^{\mathrm{a}}_{i+1}) \right](s,a)
$$

$$
+ \sum_{i=0}^{k-1} \left[ (\gamma \mathbb{P}^{\pi^*})^{k-i} \epsilon^{\mathrm{c}}_{i+1} \right](s,a),
$$

where $\vartheta_i$ is defined in (E.4) of Lemma E.1, $\epsilon^{\mathrm{a}}_{i+1}$ is defined in (E.5) of Lemma E.1, and $\epsilon^{\mathrm{c}}_{i+1}$ is defined as follows,

$$
\epsilon^{\mathrm{c}}_{i+1}(s,a) = [\mathbb{T}^{\pi_{\theta_{i+1}}} Q_{\omega_i} - Q_{\omega_{i+1}}](s,a). \tag{E.6}
$$

*Proof.* See §H.3 for a detailed proof. □

We remark that $\epsilon_{k+1}^{\mathrm{a}} = 0$ for any $k$ in the linear actor-critic method. Meanwhile, such a term is included in Lemma E.2 only aiming to generalize to the deep neural actor-critic method.

**Lemma E.3.** It holds for any $(s, a) \in \mathcal{S} \times \mathcal{A}$ that

$$A_{3,k}(s, a) = \left[\gamma \mathbb{P}^{\pi_{\theta_{k+1}}} (I - \gamma \mathbb{P}^{\pi_{\theta_{k+1}}})^{-1} e_{k+1}\right](s, a),$$

where $e_{k+1}$ is defined as follows,

$$e_{k+1}(s, a) = [Q_{\omega_k} - \mathbb{T}^{\pi_{\theta_{k+1}}} Q_{\omega_k}](s, a). \tag{E.7}$$

*Proof.* See §H.4 for a detailed proof. □

We upper bound $e_{k+1}$ in (E.7) of Lemma E.3 using Lemma E.4 as follows.

**Lemma E.4.** It holds for any $(s, a) \in \mathcal{S} \times \mathcal{A}$ that

$$e_{k+1}(s, a) \le \left[\gamma^k \Big(\prod_{s=1}^{k} \mathbb{P}^{\pi_{\theta_s}}\Big) e_1 + \sum_{i=1}^{k} \gamma^{k-i} \Big(\prod_{s=i+1}^{k} \mathbb{P}^{\pi_{\theta_s}}\Big) \big(\gamma \beta \mathbb{P} \epsilon_{i+1}^{\mathrm{b}} + (I - \gamma \mathbb{P}^{\pi_{\theta_i}}) \epsilon_i^{\mathrm{c}}\big)\right](s, a).$$

where $\epsilon_i^{\mathrm{c}}(s, a)$ is defined in (E.6) of Lemma E.2 and $\epsilon_{i+1}^{\mathrm{b}}(s)$ is defined as follows,

$$\epsilon_{i+1}^{\mathrm{b}}(s) = \big\langle \log\big(\pi_{\theta_{i+1}}(\cdot \,|\, s)/\pi_{\theta_i}(\cdot \,|\, s)\big) - \beta^{-1} \cdot Q_{\omega_i}(s, \cdot), \pi_{\theta_i}(\cdot \,|\, s) - \pi_{\theta_{i+1}}(\cdot \,|\, s)\big\rangle. \tag{E.8}$$

*Proof.* See §H.5 for a detailed proof. □

We remark that $\epsilon_{i+1}^{\mathrm{b}} = 0$ for any $i$ in the linear actor-critic method. Meanwhile, such a term is included in Lemma E.4 only aiming to generalize to the deep neural actor-critic method.

Combining Lemmas E.3 and E.4, we obtain the following upper bound of $A_{3,k}$,

$$A_{3,k}(s, a) = \left[\gamma \mathbb{P}^{\pi_{\theta_{k+1}}} (I - \gamma \mathbb{P}^{\pi_{\theta_{k+1}}})^{-1} e_{k+1}\right](s, a)$$

$$\le \left[\gamma \mathbb{P}^{\pi_{\theta_{k+1}}} (I - \gamma \mathbb{P}^{\pi_{\theta_{k+1}}})^{-1} \left(\gamma^k \Big(\prod_{s=1}^{k} \mathbb{P}^{\pi_{\theta_s}}\Big) e_1\right.\right. \tag{E.9}$$

$$\left.\left. + \sum_{i=1}^{k} \gamma^{k-i} \Big(\prod_{s=i+1}^{k} \mathbb{P}^{\pi_{\theta_s}}\Big) \big(\beta \gamma \mathbb{P} \epsilon_{i+1}^{\mathrm{b}} + (I - \gamma \mathbb{P}^{\pi_{\theta_i}}) \epsilon_i^{\mathrm{c}}\big)\right)\right](s, a).$$

Combining (E.1), (E.2), Lemma E.1 and Lemma E.2, it holds for any $(s, a) \in \mathcal{S} \times \mathcal{A}$ that

$$\sum_{k=0}^{K} [Q^* - Q^{\pi_{\theta_{k+1}}}](s, a)$$

$$\le \sum_{k=0}^{K} \left[(I - \gamma \mathbb{P}^{\pi^*})^{-1} \Big((\gamma \mathbb{P}^{\pi^*})^{k+1} (Q^* - Q_{\omega_0}) + \sum_{i=0}^{k} (\gamma \mathbb{P}^{\pi^*})^{k-i} \gamma \beta \mathbb{P}(\vartheta_i + \epsilon_{i+1}^{\mathrm{a}})\right.$$

$$\left. + \sum_{i=0}^{k-1} (\gamma \mathbb{P}^{\pi^*})^{k-i} \epsilon_{i+1}^{\mathrm{c}} + A_{3,k}\Big)\right](s, a)$$

$$= \left[(I - \gamma \mathbb{P}^{\pi^*})^{-1} \Big(\sum_{k=0}^{K} (\gamma \mathbb{P}^{\pi^*})^{k+1} (Q^* - Q_{\omega_0}) + \sum_{k=0}^{K} \sum_{i=0}^{k} (\gamma \mathbb{P}^{\pi^*})^{k-i} \gamma \beta \mathbb{P} \epsilon_{i+1}^{\mathrm{a}}\right. \tag{E.10}$$

$$\left. + \sum_{k=0}^{K} \sum_{i=0}^{k-1} (\gamma \mathbb{P}^{\pi^*})^{k-i} \epsilon_{i+1}^{\mathrm{c}} + \sum_{k=0}^{K} A_{3,k} + \sum_{k=0}^{K} \sum_{i=0}^{k} (\gamma \mathbb{P}^{\pi^*})^{k-i} \gamma \beta \mathbb{P} \vartheta_i\Big)\right](s, a),$$

where $\vartheta_i$, $\epsilon_{i+1}^{\mathrm{a}}$, $\epsilon_{i+1}^{\mathrm{c}}$, and $e_{k+1}$ are defined in (E.4) of Lemma E.1, (E.5) of Lemma E.1, (E.6) of Lemma E.2, and (E.7) of Lemma E.3, respectively. We upper bound the last term as follows,

$$
\begin{aligned}
\left[\sum_{k=0}^{K}\sum_{i=0}^{k}(\gamma\mathbb{P}^{\pi^*})^{k-i}\gamma\beta\mathbb{P}\vartheta_i\right](s,a) &= \left[\sum_{k=0}^{K}\sum_{i=0}^{k}\gamma\beta(\gamma\mathbb{P}^{\pi^*})^i\mathbb{P}\vartheta_{k-i}\right](s,a) \\
&= \left[\sum_{i=0}^{K}\gamma\beta(\gamma\mathbb{P}^{\pi^*})^i\mathbb{P}\sum_{k=i}^{K}\vartheta_{k-i}\right](s,a) \\
&= \left[\sum_{i=0}^{K}\gamma\beta(\gamma\mathbb{P}^{\pi^*})^i\mathbb{P}\sum_{k=i}^{K}\Big(\mathrm{KL}\big(\pi^*\,\|\,\pi_{\theta_{k-i}}\big) - \mathrm{KL}\big(\pi^*\,\|\,\pi_{\theta_{k-i+1}}\big)\Big)\right](s,a) \\
&= \left[\sum_{i=0}^{K}\gamma\beta(\gamma\mathbb{P}^{\pi^*})^i\mathbb{P}\big(\mathrm{KL}(\pi^*\,\|\,\pi_{\theta_0}) - \mathrm{KL}(\pi^*\,\|\,\pi_{\theta_{K-i+1}})\big)\right](s,a) \\
&\le \left[\sum_{i=0}^{K}\gamma\beta(\gamma\mathbb{P}^{\pi^*})^i\mathbb{P}\mathrm{KL}(\pi^*\,\|\,\pi_{\theta_0})\right](s,a), \tag{E.11}
\end{aligned}
$$

where we use the definition of $\vartheta_{k-i}$ in (E.4) of Lemma E.1 and the non-negativity of the KL divergence in the second equality and the last inequality, respectively. By plugging (E.9) and (E.11) into (E.10), we have

$$
\begin{aligned}
\sum_{k=0}^{K}&[Q^* - Q^{\pi_{\theta_{k+1}}}](s,a) \\
&\le \left[(I-\gamma\mathbb{P}^{\pi^*})^{-1}\left(\sum_{k=0}^{K}(\gamma\mathbb{P}^{\pi^*})^{k+1}(Q^*-Q_{\omega_0}) + \sum_{k=0}^{K}\sum_{i=0}^{k}(\gamma\mathbb{P}^{\pi^*})^{k-i}\gamma\beta\mathbb{P}\epsilon_{i+1}^{\mathrm{a}}\right.\right. \tag{E.12} \\
&\quad + \sum_{k=0}^{K}\sum_{i=0}^{k-1}(\gamma\mathbb{P}^{\pi^*})^{k-i}\epsilon_{i+1}^{\mathrm{c}} + \sum_{k=0}^{K}\gamma^{k+1}\mathbb{P}^{\pi_{\theta_{k+1}}}\big(I-\gamma\mathbb{P}^{\pi_{\theta_{k+1}}}\big)^{-1}\Big(\prod_{s=1}^{k}\mathbb{P}^{\pi_{\theta_s}}\Big)e_1 \\
&\quad + \left.\left.\sum_{k=0}^{K}\mathbb{P}^{\pi_{\theta_{k+1}}}\big(I-\gamma\mathbb{P}^{\pi_{\theta_{k+1}}}\big)^{-1}\sum_{\ell=1}^{k}\gamma^{k-\ell+1}\Big(\prod_{s=\ell+1}^{k}\mathbb{P}^{\pi_{\theta_s}}\Big)\big(\gamma\beta\mathbb{P}\epsilon_{\ell+1}^{\mathrm{b}} + (I-\gamma\mathbb{P}^{\pi_{\theta_\ell}})\epsilon_\ell^{\mathrm{c}}\big)\right)\right](s,a). \\
&\quad + \sum_{i=0}^{K}(\gamma\mathbb{P}^{\pi^*})^i\gamma\beta\mathbb{P}\mathrm{KL}(\pi^*\,\|\,\pi_{\theta_0})
\end{aligned}
$$

We remark that $\epsilon_{i+1}^{\mathrm{a}} = \epsilon_{i+1}^{\mathrm{b}} = 0$ for any $i$ in the linear actor-critic method. Meanwhile, such terms is included in (E.12) only aiming to generalize to the deep neural actor-critic method. This concludes the proof in part 1.

**Part 2.** Recall that $\rho$ is a state-action distribution satisfying (ii) of Assumption 4.1. In the sequel, we take the expectation over $\rho$ in (E.12) and upper bound each term. Recall that $\epsilon_{i+1}^{\mathrm{a}} = \epsilon_{i+1}^{\mathrm{b}} = 0$ for any $i$ in the linear actor-critic method. Hence, we only need to consider terms in (E.12) that do not involve $\epsilon_{i+1}^{\mathrm{a}}$ or $\epsilon_{i+1}^{\mathrm{b}}$. We first upper bound terms on the RHS of (E.12) that do not involve $\epsilon_{i+1}^{\mathrm{c}}$. More specifically, for any measure $\rho$ satisfying satisfying (ii) of Assumption 4.1, we upper bound the following three terms,

$$
M_1 = \mathbb{E}_\rho\left[(I-\gamma\mathbb{P}^{\pi^*})^{-1}\sum_{k=0}^{K}(\gamma\mathbb{P}^{\pi^*})^{k+1}(Q^*-Q_{\omega_0})\right],
$$

$$
M_2 = \mathbb{E}_\rho\left[(I-\gamma\mathbb{P}^{\pi^*})^{-1}\sum_{k=0}^{K}\gamma^{k+1}\mathbb{P}^{\pi_{\theta_{k+1}}}\big(I-\gamma\mathbb{P}^{\pi_{\theta_{k+1}}}\big)^{-1}\Big(\prod_{s=1}^{k}\mathbb{P}^{\pi_{\theta_s}}\Big)e_1\right],
$$

$$
M_3 = \mathbb{E}_\rho\left[(I-\gamma\mathbb{P}^{\pi^*})^{-1}\sum_{i=0}^{K}(\gamma\mathbb{P}^{\pi^*})^i\gamma\beta\mathbb{P}\mathrm{KL}(\pi^*\,\|\,\pi_{\theta_0})\right]. \tag{E.13}
$$

We upper bound $M_1$, $M_2$, and $M_3$ in the following lemma.

**Lemma E.5.** It holds that

$$|M_1| \le 4(1-\gamma)^{-2} \cdot (r_{\max} + R), \qquad |M_2| \le (1-\gamma)^{-3} \cdot (2R + r_{\max}),$$
$$|M_3| \le (1-\gamma)^{-2} \cdot \log|\mathcal{A}| \cdot K^{1/2},$$

where $M_1$, $M_2$, and $M_3$ are defined in (E.13).

*Proof.* See §H.6 for a detailed proof. $\qquad\square$

Now, we upper bound terms on the RHS of (E.12) that involve $\epsilon_{i+1}^{\mathrm{c}}$. More specifically, for any measure $\rho$ satisfying (ii) of Assumption 4.1, we upper bound the following two terms,

$$M_4 = \mathbb{E}_\rho\Big[(I - \gamma\mathbb{P}^{\pi^*})^{-1}\sum_{k=0}^{K}\sum_{i=0}^{k}(\gamma\mathbb{P}^{\pi^*})^{k-i}\epsilon_{i+1}^{\mathrm{c}}\Big], \tag{E.14}$$

$$M_5 = \mathbb{E}_\rho\Big[(I - \gamma\mathbb{P}^{\pi^*})^{-1}\sum_{k=0}^{K}\mathbb{P}^{\pi_{\theta_{k+1}}}(I - \gamma\mathbb{P}^{\pi_{\theta_{k+1}}})^{-1}\sum_{\ell=1}^{k}\gamma^{k-\ell+1}\Big(\prod_{s=\ell+1}^{k}\mathbb{P}^{\pi_{\theta_s}}\Big)(I - \gamma\mathbb{P}^{\pi_{\theta_\ell}})\epsilon_\ell^{\mathrm{c}}\Big].$$

We upper bound $M_4$ and $M_5$ in the following lemma.

**Lemma E.6.** It holds that

$$|M_4| \le 3KC_{\rho,\rho^*} \cdot \varepsilon_Q, \qquad |M_5| \le KC_{\rho,\rho^*} \cdot \varepsilon_Q.$$

where $M_4$ and $M_5$ are defined in (E.14).

*Proof.* See §H.7 for a detailed proof. $\qquad\square$

Now, by plugging Lemmas E.5 and E.6 into (E.12), we have

$$\mathbb{E}_\rho\Big[\sum_{k=0}^{K}Q^*(s,a) - Q^{\pi_{\theta_{k+1}}}(s,a)\Big]$$
$$\le 2(1-\gamma)^{-3} \cdot \log|\mathcal{A}| \cdot K^{1/2} + 4KC_{\rho,\rho^*} \cdot \varepsilon_Q + O(1). \tag{E.15}$$

Meanwhile, by changing measure from $\rho^*$ to $\rho_{k+1}$, it holds for any $k$ that

$$\mathbb{E}_{\rho^*}[|\epsilon_{k+1}^{\mathrm{c}}|] \le \sqrt{\mathbb{E}_{\rho_{k+1}}\big[(\epsilon_{k+1}^{\mathrm{c}}(s,a))^2\big]} \cdot \phi_{k+1}^*, \tag{E.16}$$

where $\phi_{k+1}^*$ is defined in Assumption 4.1. Also, by Lemma D.1, it holds that

$$\sqrt{\mathbb{E}_{\rho_{k+1}}\big[(\epsilon_{k+1}^{\mathrm{c}}(s,a))^2\big]} = O\big(1/(\sqrt{N}\sigma^*) \cdot \log N\big). \tag{E.17}$$

Now, by plugging (E.17) into (E.16), combining the definition of $\varepsilon_Q = \max_k \mathbb{E}_{\rho^*}[|\epsilon_{k+1}^{\mathrm{c}}|]$, we have

$$\varepsilon_Q = O\big(\phi^*/(\sqrt{N}\sigma^*) \cdot \log N\big). \tag{E.18}$$

Combining (E.15), (E.18), and the choices of parameters stated in the theorem that

$$N = \Omega\big(KC_{\rho,\rho^*}^2(\phi^*/\sigma^*)^2 \cdot \log^2 N\big),$$

we have

$$\mathbb{E}_\rho\Big[\sum_{k=0}^{K}Q^*(s,a) - Q^{\pi_{\theta_{k+1}}}(s,a)\Big] \le \big(2(1-\gamma)^{-3}\log|\mathcal{A}| + O(1)\big) \cdot K^{1/2},$$

which concludes the proof of Theorem 4.4.

### E.2 Proof of Theorem C.5

We follow the proof of Theorem 4.4 in §E.1. Following similar arguments when deriving (E.12) in §E.1, we have

$$
\sum_{k=0}^{K} [Q^* - Q^{\pi_{\theta_{k+1}}}](s,a)
$$

$$
\leq \Bigg[ (I - \gamma \mathbb{P}^{\pi^*})^{-1} \cdot \Bigg( \sum_{k=0}^{K} (\gamma \mathbb{P}^{\pi^*})^{k+1} (Q^* - Q_{\omega_0}) + \sum_{k=0}^{K} \sum_{i=0}^{k} (\gamma \mathbb{P}^{\pi^*})^{k-i} \cdot \gamma \beta \mathbb{P} \epsilon_{i+1}^{\mathrm{a}} \tag{E.19}
$$

$$
+ \sum_{k=0}^{K} \sum_{i=0}^{k-1} (\gamma \mathbb{P}^{\pi^*})^{k-i} \epsilon_{i+1}^{\mathrm{c}} + \sum_{i=0}^{K} (\gamma \mathbb{P}^{\pi^*})^{i} \cdot \gamma \beta \mathbb{P} \cdot \mathrm{KL}(\pi^* \,\|\, \pi_{\theta_0})
$$

$$
+ \sum_{k=0}^{K} \gamma^{k+1} \mathbb{P}^{\pi_{\theta_{k+1}}} (I - \gamma \mathbb{P}^{\pi_{\theta_{k+1}}})^{-1} \Big( \prod_{s=1}^{k} \mathbb{P}^{\pi_{\theta_s}} \Big) e_1
$$

$$
+ \sum_{k=0}^{K} \mathbb{P}^{\pi_{\theta_{k+1}}} (I - \gamma \mathbb{P}^{\pi_{\theta_{k+1}}})^{-1} \sum_{\ell=1}^{k} \gamma^{k-\ell+1} \Big( \prod_{s=\ell+1}^{k} \mathbb{P}^{\pi_{\theta_s}} \Big) \Big( \beta \gamma \mathbb{P} \epsilon_{\ell+1}^{\mathrm{b}} - (I - \gamma \mathbb{P}^{\pi_{\theta_\ell}}) \epsilon_\ell^{\mathrm{c}} \Big) \Bigg) \Bigg](s,a),
$$

for any $(s,a) \in \mathcal{S} \times \mathcal{A}$. Here $\epsilon_{i+1}^{\mathrm{a}}$, $\epsilon_{\ell+1}^{\mathrm{b}}$, $\epsilon_{i+1}^{\mathrm{c}}$, and $e_1$ are defined in (E.5), (E.8), (E.6), and (E.7), respectively.

Now, it remains to upper bound each term on the RHS of (E.19). We introduce the following error propagation lemma.

**Lemma E.7.** Suppose that

$$
\mathbb{E}_{\rho_k} \Big[ \big( f_{\theta_{k+1}}(s,a) - \tau_{k+1} \cdot (\beta^{-1} Q_{\omega_k}(s,a) - \tau_k^{-1} f_{\theta_k}(s,a)) \big)^2 \Big]^{1/2} \leq \varepsilon_{k+1,f}. \tag{E.20}
$$

Then, we have

$$
\mathbb{E}_{\nu^*} \big[ |\epsilon_{k+1}^{\mathrm{a}}(s)| \big] \leq \sqrt{2} \tau_{k+1}^{-1} \cdot \varepsilon_{k+1,f} \cdot (\phi_k^* + \psi_k^*), \quad \mathbb{E}_{\nu^*} \big[ |\epsilon_{k+1}^{\mathrm{b}}(s)| \big] \leq \sqrt{2} \tau_{k+1}^{-1} \cdot \varepsilon_{k+1,f} \cdot (1 + \psi_k^*),
$$

where $\epsilon_{k+1}^{\mathrm{a}}$ and $\epsilon_{k+1}^{\mathrm{b}}$ are defined in (E.5) and (E.8), respectively, $\phi_k^*$ and $\psi_k^*$ are defined in Assumption C.1.

*Proof.* See §H.8 for a detailed proof. □

Following from Lemma F.4, with probability at least $1 - O(H_{\mathrm{c}}) \exp(-\Omega(H_{\mathrm{c}}^{-1} m_{\mathrm{c}}))$, we have $|Q_{\omega_0}| \leq 2$. Also, from the fact that $|r(s,a)| \leq r_{\max}$, we know that $|Q^*| \leq r_{\max}$. Therefore, for any measure $\rho$, we have

$$
\Big| \mathbb{E}_{\rho} \Big[ (I - \gamma \mathbb{P}^{\pi^*})^{-1} \sum_{k=0}^{K} (\gamma \mathbb{P}^{\pi^*})^{k+1} (Q^* - Q_{\omega_0}) \Big] \Big|
$$

$$
\leq \mathbb{E}_{\rho} \Big[ (I - \gamma \mathbb{P}^{\pi^*})^{-1} \sum_{k=0}^{K} (\gamma \mathbb{P}^{\pi^*})^{k+1} |Q^* - Q_{\omega_0}| \Big]
$$

$$
\leq r_{\max} (1 - \gamma)^{-1} \sum_{k=0}^{K} \gamma^{k+1} \leq r_{\max} (1 - \gamma)^{-2}. \tag{E.21}
$$

Also, by changing the index of summation, we have

$$
\left| \mathbb{E}_\rho \Big[ (I - \gamma \mathbb{P}^{\pi^*})^{-1} \sum_{k=0}^{K} \sum_{i=0}^{k} (\gamma \mathbb{P}^{\pi^*})^{k-i} \gamma \beta \mathbb{P} \epsilon_{i+1}^{\mathrm{a}} \Big] \right|
$$

$$
= \left| \mathbb{E}_\rho \Big[ \sum_{k=0}^{K} \sum_{i=0}^{k} \sum_{j=0}^{\infty} (\gamma \mathbb{P}^{\pi^*})^{k-i+j} \gamma \beta \mathbb{P} \epsilon_{i+1}^{\mathrm{a}} \Big] \right|
$$

$$
= \left| \mathbb{E}_\rho \Big[ \sum_{k=0}^{K} \sum_{i=0}^{k} \sum_{t=k-i}^{\infty} (\gamma \mathbb{P}^{\pi^*})^{t} \gamma \beta \mathbb{P} \epsilon_{i+1}^{\mathrm{a}} \Big] \right|
$$

$$
\leq \sum_{k=0}^{K} \sum_{i=0}^{k} \sum_{t=k-i}^{\infty} \left| \mathbb{E}_\rho \big[ (\gamma \mathbb{P}^{\pi^*})^{t} \gamma \beta \mathbb{P} \epsilon_{i+1}^{\mathrm{a}} \big] \right|, \tag{E.22}
$$

where we expand $(I - \gamma \mathbb{P}^{\pi^*})^{-1}$ into an infinite sum in the first equality. Further, by changing the measure of the expectation on the RHS of (E.22), we have

$$
\sum_{k=0}^{K} \sum_{i=0}^{k} \sum_{t=k-i}^{\infty} \left| \mathbb{E}_\rho \big[ (\gamma \mathbb{P}^{\pi^*})^{t} \gamma \beta \mathbb{P} \epsilon_{i+1}^{\mathrm{a}} \big] \right| \leq \sum_{k=0}^{K} \sum_{i=0}^{k} \sum_{t=k-i}^{\infty} \beta \gamma^{t+1} c(t) \cdot \mathbb{E}_{\nu^*} \big[ |\epsilon_{i+1}^{\mathrm{A}}| \big], \tag{E.23}
$$

where $c(t)$ is defined in Assumption C.1. Further, by Lemma E.7 and interchanging the summation on the RHS of (E.23), we have

$$
\left| \mathbb{E}_\rho \Big[ (I - \gamma \mathbb{P}^{\pi^*})^{-1} \sum_{k=0}^{K} \sum_{i=0}^{k} (\gamma \mathbb{P}^{\pi^*})^{k-i} \gamma \beta \mathbb{P} \epsilon_{i+1}^{\mathrm{a}} \Big] \right|
$$

$$
\leq 2 \sum_{k=0}^{K} \sum_{t=0}^{\infty} \sum_{i=\max\{0, k-t\}}^{k} \beta \gamma^{t+1} c(t) \cdot \tau_{i+1}^{-1} \varepsilon_f (\phi_i^* + \psi_i^*)
$$

$$
\leq \sum_{k=0}^{K} \sum_{t=0}^{\infty} 4 k t \gamma^{t+1} c(t) \cdot \varepsilon_f (\phi^* + \psi^*)
$$

$$
\leq \gamma \sum_{k=0}^{K} 4 C_{\rho,\rho^*} \cdot \varepsilon_f (\phi^* + \psi^*) \leq 2 \gamma K C_{\rho,\rho^*} (\phi^* + \psi^*) \cdot \varepsilon_f, \tag{E.24}
$$

where $\varepsilon_f = \max_i \mathbb{E}_{\rho_i} [(f_{\theta_{i+1}}(s,a) - \tau_{i+1} \cdot (\beta^{-1} Q_{\omega_i}(s,a) - \tau_i^{-1} f_{\theta_i}(s,a)))^2]^{1/2}$, and $C_{\rho,\rho^*}$ is defined in Assumption C.1. Here in the second inequality, we use the fact that $\tau_{i+1}^{-1} = (i+1) \cdot \beta^{-1}$, and $\phi_i^* \leq \phi^*$ and $\psi_i^* \leq \psi^*$ by Assumption C.1.

By similar arguments in the derivation of (E.24), we have

$$
\left| \mathbb{E}_\rho \Big[ (I - \gamma \mathbb{P}^{\pi^*})^{-1} \sum_{k=0}^{K} \sum_{i=0}^{k-1} (\gamma \mathbb{P}^{\pi^*})^{k-i} \epsilon_{i+1}^{\mathrm{c}} \Big] \right| \leq 2(K+1) C_{\rho,\rho^*} \phi^* \cdot \varepsilon_Q, \tag{E.25}
$$

$$
\left| \mathbb{E}_\rho \Big[ (I - \gamma \mathbb{P}^{\pi^*})^{-1} \sum_{i=0}^{K} (\gamma \mathbb{P}^{\pi^*})^{i} \gamma \beta \mathbb{P} \mathbf{KL}(\pi^* \,\|\, \pi_{\theta_0}) \Big] \right| \leq \log |\mathcal{A}| \cdot K^{1/2} (1-\gamma)^{-2},
$$

$$
\mathbb{E}_\rho \Big[ (I - \gamma \mathbb{P}^{\pi^*})^{-1} \sum_{k=0}^{K} \gamma^{k+1} \mathbb{P}^{\pi_{\theta_{k+1}}} (I - \gamma \mathbb{P}^{\pi_{\theta_{k+1}}})^{-1} \Big( \prod_{s=1}^{k} \mathbb{P}^{\pi_{\theta_s}} \Big) e_1 \Big] \leq (2 + r_{\max}) \cdot (1-\gamma)^{-3},
$$

where $\varepsilon_Q = \max_i \mathbb{E}_{\rho^*} [|\epsilon_{i+1}^{\mathrm{c}}|]$. And we use the fact that $\beta = K^{1/2}$.

Now, it remains to upper bound the last term on the RHS of (E.19). We first consider the terms involving $\epsilon_{\ell+1}^{\mathrm{b}}$. We have

$$
\mathbb{E}_\rho \left[ (I - \gamma \mathbb{P}^{\pi^*})^{-1} \sum_{k=0}^{K} \mathbb{P}^{\pi_{\theta_{k+1}}} (I - \gamma \mathbb{P}^{\pi_{\theta_{k+1}}})^{-1} \sum_{\ell=1}^{k} \gamma^{k-\ell+1} \Big( \prod_{s=\ell+1}^{k} \mathbb{P}^{\pi_{\theta_s}} \Big) \beta \gamma \mathbb{P} \epsilon_{\ell+1}^{\mathrm{b}} \right]
$$

$$
= \sum_{j=0}^{\infty} \sum_{i=0}^{\infty} \sum_{k=0}^{K} \sum_{\ell=1}^{k} \mathbb{E}_\rho \left[ (\gamma \mathbb{P}^{\pi^*})^j (\gamma \mathbb{P}^{\pi_{\theta_{k+1}}})^{i+1} \gamma^{k-\ell} \Big( \prod_{s=\ell+1}^{k} \mathbb{P}^{\pi_{\theta_s}} \Big) \beta \gamma \mathbb{P} \epsilon_{\ell+1}^{\mathrm{b}} \right]
$$

$$
\leq \beta \gamma \sum_{k=0}^{K} \sum_{\ell=1}^{k} \sum_{j=0}^{\infty} \sum_{i=0}^{\infty} \gamma^{i+j+k-\ell+1} \cdot \mathbb{E}_{\rho^*}[|\mathbb{P} \epsilon_{\ell+1}^{\mathrm{b}}|] \cdot c(i+j+k-\ell+1)
$$

$$
\leq 2\gamma \sum_{k=0}^{K} \sum_{\ell=1}^{k} \sum_{j=0}^{\infty} \sum_{i=0}^{\infty} \gamma^{i+j+k-\ell+1} \cdot (\ell+1)\varepsilon_f \cdot (1 + \psi_\ell^*) \cdot c(i+j+k-\ell+1), \quad \text{(E.26)}
$$

where we expand $(I - \gamma \mathbb{P}^{\pi^*})^{-1}$ and $(I - \gamma \mathbb{P}^{\pi_{\theta_{k+1}}})^{-1}$ to infinite sums in the first equality, change the measure of the expectation in the first inequality, and use Lemma E.7 in the last inequality. Now, by changing the index of the summation, we have

$$
\gamma \sum_{k=0}^{K} \sum_{\ell=1}^{k} \sum_{j=0}^{\infty} \sum_{i=0}^{\infty} \gamma^{i+j+k-\ell+1} \cdot (\ell+1)\varepsilon_f \cdot (1 + \psi_\ell^*) \cdot c(i+j+k-\ell+1)
$$

$$
= \gamma \sum_{k=0}^{K} \sum_{\ell=1}^{k} \sum_{j=0}^{\infty} \sum_{t=j+k-\ell+1}^{\infty} \gamma^t \cdot (\ell+1)\varepsilon_f \cdot (1 + \psi_\ell^*) \cdot c(t)
$$

$$
\leq \gamma \sum_{k=0}^{K} \sum_{j=0}^{\infty} \sum_{t=j+1}^{\infty} \sum_{\ell=\max\{0,j+k-t+1\}}^{k} \gamma^t \cdot (\ell+1)\varepsilon_f \cdot (1 + \psi^*) \cdot c(t), \quad \text{(E.27)}
$$

where we use the fact that $\psi_\ell^* \leq \psi^*$ from Assumption C.1 in the last inequality. By further manipulating the order of summations of the RHS of (E.27), we have

$$
\gamma \sum_{k=0}^{K} \sum_{j=0}^{\infty} \sum_{t=j+1}^{\infty} \sum_{\ell=\max\{0,j+k-t+1\}}^{k} \gamma^t \cdot (\ell+1)\varepsilon_f (1 + \psi^*) \cdot c(t)
$$

$$
\leq \gamma \sum_{k=0}^{K} \sum_{j=0}^{\infty} \Big( \sum_{t=j+1}^{j+k+1} (t-j)(2k+j-k+1) \cdot \gamma^t c(t) + \sum_{t=j+k+2}^{\infty} k^2 \cdot \gamma^t c(t) \Big) \cdot \varepsilon_f (1 + \psi^*)
$$

$$
= \gamma \sum_{k=0}^{K} \Big( \sum_{t=1}^{\infty} \sum_{j=\max\{0,t-k-1\}}^{t-1} (t-j)(2k+j-k+1) \cdot \gamma^t c(t)
$$

$$
+ \sum_{t=k+2}^{\infty} \sum_{j=1}^{t-k-2} k^2 \cdot \gamma^t c(t) \Big) \cdot \varepsilon_f (1 + \psi^*)
$$

$$
\leq 20\gamma \sum_{k=0}^{K} \Big( \sum_{t=1}^{\infty} k^2 \cdot t \gamma^t c(t) + \sum_{t=1}^{\infty} k^2 \cdot t \gamma^t c(t) \Big) \cdot \varepsilon_f (1 + \psi^*)
$$

$$
\leq 20\gamma K \cdot C_{\rho,\rho^*} \cdot \varepsilon_f (1 + \psi^*), \quad \text{(E.28)}
$$

where we use the definition of $C_{\rho,\rho^*}$ from Assumption C.1 in the last inequality. Now, combining (E.26), (E.27), and (E.28), we have

$$
\mathbb{E}_\rho \left[ (I - \gamma \mathbb{P}^{\pi^*})^{-1} \sum_{k=0}^{K} \mathbb{P}^{\pi_{\theta_{k+1}}} (I - \gamma \mathbb{P}^{\pi_{\theta_{k+1}}})^{-1} \sum_{\ell=1}^{k} \gamma^{k-\ell+1} \Big( \prod_{s=\ell+1}^{k} \mathbb{P}^{\pi_{\theta_s}} \Big) \beta \gamma \mathbb{P} \epsilon_{\ell+1}^{\mathrm{b}} \right]
$$

$$
\leq 20\gamma K \cdot C_{\rho,\rho^*} \cdot \varepsilon_f \cdot (1 + \psi^*). \quad \text{(E.29)}
$$

Following from similar arguments when deriving (E.29), we have

$$\mathbb{E}_\rho\Big[(I-\gamma\mathbb{P}^{\pi^*})^{-1}\sum_{k=0}^{K}\mathbb{P}^{\pi_{\theta_{k+1}}}(I-\gamma\mathbb{P}^{\pi_{\theta_{k+1}}})^{-1}\sum_{\ell=1}^{k}\gamma^{k-\ell+1}\Big(\prod_{s=\ell+1}^{k}\mathbb{P}^{\pi_{\theta_s}}\Big)(I-\gamma\mathbb{P}^{\pi_{\theta_\ell}})\epsilon_\ell^c\Big]$$

$$\leq 20K\cdot C_{\rho,\rho^*}\phi^*\cdot\varepsilon_Q, \tag{E.30}$$

Now, by plugging (E.21), (E.24), (E.25), (E.29), and (E.30) into (E.19), with probability at least $1-O(H_c)\exp(-\Omega(H_c^{-1}m_c))$, we have

$$\mathbb{E}_\rho\Big[\sum_{k=0}^{K}Q^*(s,a)-Q^{\pi_{\theta_{k+1}}}(s,a)\Big] \tag{E.31}$$

$$\leq 2\log|\mathcal{A}|\cdot K^{1/2}(1-\gamma)^{-3}+60KC_{\rho,\rho^*}(\phi^*+\psi^*+1)\cdot\varepsilon_f+50KC_{\rho,\rho^*}\phi^*\cdot\varepsilon_Q.$$

Meanwhile, following from Propositions C.3 and C.4, it holds with probability at least $1-1/K$ that

$$\varepsilon_f=O\big(R_a N_a^{-1/4}+R_a^{4/3}m_a^{-1/12}H_a^{7/2}(\log m_a)^{1/2}\big),$$
$$\varepsilon_Q=O\big(R_c N_c^{-1/4}+R_c^{4/3}m_c^{-1/12}H_c^{7/2}(\log m_c)^{1/2}\big). \tag{E.32}$$

Combining (E.31), (E.32), and the choices of parameters stated in the theorem, it holds with probability at least $1-1/K$ that

$$\mathbb{E}_\rho\Big[\sum_{k=0}^{K}Q^*(s,a)-Q^{\pi_{\theta_{k+1}}}(s,a)\Big]\leq\big(2(1-\gamma)^{-3}\log|\mathcal{A}|+O(1)\big)\cdot K^{1/2},$$

which concludes the proof of Theorem C.5.

## F SUPPORTING RESULTS

In this section, we provide some supporting results in the proof of Theorems 4.4 and C.5. We introduce Lemma F.1, which applies to both Algorithms 1 and 2. To introduce Lemma F.1, for any policy $\pi$ and action-value function $Q$, we define $\widetilde{\pi}(a\,|\,s)\propto\exp(\beta^{-1}Q(s,a))\cdot\pi(a\,|\,s)$.

**Lemma F.1.** For any $s\in\mathcal{S}$ and $\pi^\dagger$, we have

$$\beta^{-1}\cdot\langle Q(s,\cdot),\pi^\dagger(\cdot\,|\,s)-\widetilde{\pi}(\cdot\,|\,s)\rangle\leq \mathrm{KL}\big(\pi^\dagger(\cdot\,|\,s)\,\|\,\pi(\cdot\,|\,s)\big)-\mathrm{KL}\big(\pi^\dagger(\cdot\,|\,s)\,\|\,\widetilde{\pi}(\cdot\,|\,s)\big)$$
$$+\big\langle\log\big(\widetilde{\pi}(\cdot\,|\,s)/\pi(\cdot\,|\,s)\big)-\beta^{-1}\cdot Q(s,\cdot),\pi^\dagger(\cdot\,|\,s)-\widetilde{\pi}(\cdot\,|\,s)\big\rangle.$$

*Proof.* By calculation, it suffices to show that

$$\big\langle\log(\widetilde{\pi}(\cdot\,|\,s)/\pi(\cdot\,|\,s)),\pi^\dagger(\cdot\,|\,s)-\widetilde{\pi}(\cdot\,|\,s)\big\rangle$$
$$\leq\mathrm{KL}(\pi^\dagger(\cdot\,|\,s)\,\|\,\pi(\cdot\,|\,s))-\mathrm{KL}(\pi^\dagger(\cdot\,|\,s)\,\|\,\widetilde{\pi}(\cdot\,|\,s)).$$

By the definition of the KL divergence, it holds for any $s\in\mathcal{S}$ that

$$\mathrm{KL}(\pi^\dagger(\cdot\,|\,s)\,\|\,\pi(\cdot\,|\,s))-\mathrm{KL}(\pi^\dagger(\cdot\,|\,s)\,\|\,\widetilde{\pi}(\cdot\,|\,s))$$
$$=\big\langle\log(\widetilde{\pi}(\cdot\,|\,s)/\pi(\cdot\,|\,s)),\pi^\dagger(\cdot\,|\,s)\big\rangle. \tag{F.1}$$

Meanwhile, for the term on the RHS of (F.1), we have

$$\big\langle\log(\widetilde{\pi}(\cdot\,|\,s)/\pi_{\theta_k}(\cdot\,|\,s)),\pi^\dagger(\cdot\,|\,s)\big\rangle$$
$$=\big\langle\log(\widetilde{\pi}(\cdot\,|\,s)/\pi(\cdot\,|\,s)),\pi^\dagger(\cdot\,|\,s)-\widetilde{\pi}(\cdot\,|\,s)\big\rangle$$
$$+\big\langle\log(\widetilde{\pi}(\cdot\,|\,s)/\pi(\cdot\,|\,s)),\widetilde{\pi}(\cdot\,|\,s)\big\rangle$$
$$=\big\langle\log(\widetilde{\pi}(\cdot\,|\,s)/\pi(\cdot\,|\,s)),\pi^\dagger(\cdot\,|\,s)-\widetilde{\pi}(\cdot\,|\,s)\big\rangle+\mathrm{KL}(\widetilde{\pi}(\cdot\,|\,s)\,\|\,\pi(\cdot\,|\,s))$$
$$\geq\big\langle\log(\widetilde{\pi}(\cdot\,|\,s)/\pi(\cdot\,|\,s)),\pi^\dagger(\cdot\,|\,s)-\widetilde{\pi}(\cdot\,|\,s)\big\rangle. \tag{F.2}$$

Combining (F.1) and (F.2), we obtain that

$$\big\langle\log(\widetilde{\pi}(\cdot\,|\,s)/\pi(\cdot\,|\,s)),\pi^\dagger(\cdot\,|\,s)-\widetilde{\pi}(\cdot\,|\,s)\big\rangle$$
$$\leq\mathrm{KL}(\pi^\dagger(\cdot\,|\,s)\,\|\,\pi(\cdot\,|\,s))-\mathrm{KL}(\pi^\dagger(\cdot\,|\,s)\,\|\,\widetilde{\pi}(\cdot\,|\,s)),$$

which concludes the proof of Lemma F.1. $\square$

### F.1 LOCAL LINEARIZATION OF DNNS

In the proofs of Propositions C.3 and C.4 in §G.2 and §G.3, respectively, we utilize the linearization of DNNs. We introduce some related auxiliary results here. First, we define the linearization $\bar{u}_\theta$ of the DNN $u_\theta \in \mathcal{U}(w, H, R)$ as follows,

$$\bar{u}_\theta(\cdot) = u_{\theta_0}(\cdot) + (\theta - \theta_0)^\top \nabla_{\theta_0} u_\theta(\cdot),$$

where $\theta_0$ is the initialization of $u_\theta$. The following lemmas characterize the linearization error.

**Lemma F.2.** Suppose that $H = O(m^{1/12} R^{-1/6} (\log m)^{-1/2})$ and $m = \Omega(d^{3/2} R^{-1} H^{-3/2} \cdot \log(m^{1/2}/R)^{3/2})$. Then with probability at least $1 - \exp(-\Omega(R^{2/3} m^{2/3} H))$ over the random initialization $\theta_0$, it holds for any $\theta \in \mathcal{B}(\theta_0, R)$ and any $(s, a) \in \mathcal{S} \times \mathcal{A}$ that

$$\|\nabla_\theta u_\theta(s, a) - \nabla_\theta u_{\theta_0}(s, a)\|_2 = O(R^{1/3} m^{-1/6} H^{5/2} (\log m)^{1/2})$$

and

$$\|\nabla_\theta u_\theta(s, a)\|_2 = O(H).$$

*Proof.* See the proof of Lemma A.5 in Gao et al. (2019) for a detailed proof. □

**Lemma F.3.** Suppose that $H = O(m^{1/12} R^{-1/6} (\log m)^{-1/2})$ and $m = \Omega(d^{3/2} R^{-1} H^{-3/2} \cdot \log(m^{1/2}/R)^{3/2})$. Then with probability at least $1 - \exp(-\Omega(R^{2/3} m^{2/3} H))$ over the random initialization $\theta_0$, it holds for any $\theta \in \mathcal{B}(\theta_0, R)$ and any $(s, a) \in \mathcal{S} \times \mathcal{A}$ that

$$|u_\theta(s, a) - \bar{u}_\theta(s, a)| = O(R^{4/3} m^{-1/6} H^{5/2} (\log m)^{1/2}).$$

*Proof.* Recall that

$$\bar{u}_\theta(s, a) = u_{\theta_0}(s, a) + (\theta - \theta_0)^\top \nabla_\theta u_{\theta_0}(s, a).$$

By mean value theorem, there exists $t \in [0, 1]$, which depends on $\theta$ and $(s, a)$, such that

$$u_\theta(s, a) - \bar{u}_\theta(s, a) = (\theta - \theta_0)^\top \big( \nabla_\theta u_{\theta_0 + t(\theta - \theta_0)}(s, a) - \nabla_\theta u_{\theta_0}(s, a) \big).$$

Further by Lemma F.2, we have

$$|u_\theta(s, a) - \bar{u}_\theta(s, a)| \leq \|\theta - \theta_0\|_2 \cdot \big\| \nabla_\theta u_{\theta_0 + t \cdot (\theta - \theta_0)}(s, a) - \nabla_\theta u_{\theta_0}(s, a) \big\|_2$$
$$= O(R^{4/3} m^{-1/6} H^{5/2} (\log m)^{1/2}),$$

where we use Cauchy-Schwarz inequality in the first inequality. This concludes the proof of Lemma F.3. □

We denote by $x^{(h)}$ the output of the $h$-th layer of the DNN $u_\theta \in \mathcal{U}(m, H, R)$, and $x^{(h),0}$ the output of the $h$-th layer of the DNN $u_{\theta_0} \in \mathcal{U}(m, H, R)$. The following lemma upper bounds the distance between $x^{(h)}$ and $x^{(h),0}$.

**Lemma F.4.** With probability at least $1 - \exp(-\Omega(R^{2/3} m^{2/3} H))$ over the random initialization $\theta_0$, for any $\theta \in \mathcal{B}(\theta_0, R)$ and any $h \in [H]$, we have

$$\|x^{(h)} - x^{(h),0}\|_2 = O(R H^{5/2} m^{-1/2} (\log m)^{1/2}).$$

Also, with probability at least $1 - O(H) \exp(-\Omega(H^{-1} m))$ over the random initialization $\theta_0$, for any $\theta \in \mathcal{B}(\theta_0, R)$ and any $h \in [H]$, it holds that

$$2/3 \leq \|x^{(h)}\|_2 \leq 4/3.$$

*Proof.* The first inequality follows from Lemma A.5 in Gao et al. (2019), and the second inequality follows from Lemma 7.1 in Allen-Zhu et al. (2018b). □

## G    PROOFS OF PROPOSITIONS

### G.1    PROOF OF PROPOSITION 3.1

The proof follows the proof of Proposition 3.1 in Liu et al. (2019). First, we write the update $\widetilde{\pi}_{k+1} \leftarrow \operatorname{argmax}_\pi \mathbb{E}_{\nu_k}[\langle Q_{\omega_k}(s, \cdot), \pi(\cdot\,|\,s)\rangle - \beta \cdot \mathrm{KL}(\pi(\cdot\,|\,s)\,\|\,\pi_{\theta_k}(\cdot\,|\,s))]$ as a constrained optimization problem in the following way,

$$\max_\pi \; \mathbb{E}_{\nu_k}\big[\langle \pi(\cdot\,|\,s), Q_{\omega_k}(s, \cdot)\rangle - \beta \cdot \mathrm{KL}(\pi(\cdot\,|\,s)\,\|\,\pi_{\theta_k}(\cdot\,|\,s))\big]$$

$$\text{s.t.} \sum_{a\in\mathcal{A}} \pi(a\,|\,s) = 1, \qquad \text{for any } s \in \mathcal{S}.$$

We consider the Lagrangian of the above program,

$$\int_{s\in\mathcal{S}} \Big(\langle\pi(\cdot\,|\,s), Q_{\omega_k}(s, \cdot)\rangle - \beta \cdot \mathrm{KL}\big(\pi(\cdot\,|\,s)\,\|\,\pi_{\theta_k}(\cdot\,|\,s)\big)\Big)\mathrm{d}\nu_k(s) + \int_{s\in\mathcal{S}} \Big(\sum_{a\in\mathcal{A}} \pi(a\,|\,s) - 1\Big)\mathrm{d}\lambda(s),$$

where $\lambda(\cdot)$ is the dual parameter, which is a function on $\mathcal{S}$. Now, by plugging in

$$\pi_{\theta_k}(a\,|\,s) = \frac{\exp(\tau_k^{-1} f_{\theta_k}(s, a))}{\sum_{a'\in\mathcal{A}} \exp(\tau_k^{-1} f_{\theta_k}(s, a'))},$$

we have the following optimality condition,

$$Q_{\omega_k}(s, a) + \beta\tau_k^{-1} f_{\theta_k}(s, a) - \beta \cdot \Big(\log\Big(\sum_{a'\in\mathcal{A}} \exp(\tau_k^{-1} f_{\theta_k}(s, a'))\Big) + \log\pi(a\,|\,s) + 1\Big) + \frac{\lambda(s)}{\nu_k(s)} = 0,$$

for any $(s, a) \in \mathcal{S} \times \mathcal{A}$. Note that $\log(\sum_{a'\in\mathcal{A}} \exp(\tau_k^{-1} f_{\theta_k}(s, a')))$ is only a function of $s$. Thus, we have

$$\widehat{\pi}_{k+1}(a\,|\,s) \propto \exp(\beta^{-1} Q_{\omega_k}(s, a) + \tau_k^{-1} f_{\theta_k}(s, a))$$

for any $(s, a) \in \mathcal{S} \times \mathcal{A}$, which concludes the proof of Proposition 3.1.

### G.2    PROOF OF PROPOSITION C.3

We define the local linearization of $f_\theta$ as follows,

$$\bar{f}_\theta = f_{\theta_0} + (\theta - \theta_0)^\top \nabla_{\theta_0} f_\theta. \tag{G.1}$$

Meanwhile, we denote by

$$
\begin{aligned}
g_n &= \big(f_{\theta(n)} - \widetilde{\tau}\cdot(\beta^{-1}Q_\omega + \tau^{-1}f_\theta)\big)\cdot\nabla_\theta f_{\theta(n)}, & g_n^e &= \mathbb{E}_{\rho_{\pi_\theta}}[g_n], \\
\bar{g}_n &= \big(\bar{f}_{\theta(n)} - \widetilde{\tau}\cdot(\beta^{-1}Q_\omega + \tau^{-1}f_\theta)\big)\cdot\nabla_\theta f_{\theta_0}, & \bar{g}_n^e &= \mathbb{E}_{\rho_{\pi_\theta}}[\bar{g}_n], \\
g_* &= \big(f_{\theta_*} - \widetilde{\tau}\cdot(\beta^{-1}Q_\omega + \tau^{-1}f_\theta)\big)\cdot\nabla_\theta f_{\theta_*}, & g_*^e &= \mathbb{E}_{\rho_{\pi_\theta}}[g_*], \\
\bar{g}_* &= \big(\bar{f}_{\theta_*} - \widetilde{\tau}\cdot(\beta^{-1}Q_\omega + \tau^{-1}f_\theta)\big)\cdot\nabla_\theta f_{\theta_0}, & \bar{g}_*^e &= \mathbb{E}_{\rho_{\pi_\theta}}[\bar{g}_*], \tag{G.2}
\end{aligned}
$$

where $\theta_*$ satisfies that

$$\theta_* = \Gamma_{\mathcal{B}(\theta_0, R_\mathrm{a})}(\theta_* - \alpha\cdot\bar{g}_*^e). \tag{G.3}$$

By Algorithm 3, we know that

$$\theta(n+1) = \Gamma_{\mathcal{B}(\theta_0, R_\mathrm{a})}(\theta(n) - \alpha\cdot g_n). \tag{G.4}$$

By (G.3) and (G.4), we have

$$
\begin{aligned}
&\mathbb{E}_{\rho_{\pi_\theta}}\big[\|\theta(n+1) - \theta_*\|_2^2\,|\,\theta(n)\big] \\
&\quad = \mathbb{E}_{\rho_{\pi_\theta}}\big[\|\Gamma_{\mathcal{B}(\theta_0, R_\mathrm{a})}(\theta(n) - \alpha\cdot g_n) - \Gamma_{\mathcal{B}(\theta_0, R_\mathrm{a})}(\theta_* - \alpha\cdot\bar{g}_*^e)\|_2^2\,|\,\theta(n)\big] \\
&\quad \leq \mathbb{E}_{\rho_{\pi_\theta}}\big[\|(\theta(n) - \alpha\cdot g_n) - (\theta_* - \alpha\cdot\bar{g}_*^e)\|_2^2\,|\,\theta(n)\big] \\
&\quad = \|\theta(n) - \theta_*\|_2^2 + 2\alpha\cdot\underbrace{\langle\theta_* - \theta(n), g_n^e - \bar{g}_*^e\rangle}_{\text{(i)}} + \alpha^2\cdot\underbrace{\mathbb{E}_{\rho_{\pi_\theta}}\big[\|g_n - \bar{g}_*^e\|_2^2\,|\,\theta(n)\big]}_{\text{(ii)}}, \tag{G.5}
\end{aligned}
$$

where we use the fact that $\Gamma_{\mathcal{B}(\theta_0, R_{\mathrm{a}})}$ is a contraction mapping in the first inequality. We upper bound term (i) and term (ii) on the RHS of (G.5) in the sequel.

**Upper Bound of Term (i).** By Cauchy–Schwarz inequality, it holds that

$$
\begin{aligned}
\langle \theta_* - \theta(n), g_n^e - \bar{g}_*^e \rangle &= \langle \theta_* - \theta(n), g_n^e - \bar{g}_n^e \rangle + \langle \theta_* - \theta(n), \bar{g}_n^e - \bar{g}_*^e \rangle \\
&\leq \|\theta_* - \theta(n)\|_2 \cdot \|g_n^e - \bar{g}_n^e\|_2 + \langle \theta_* - \theta(n), \bar{g}_n^e - \bar{g}_*^e \rangle \\
&\leq 2R_{\mathrm{a}} \cdot \|g_n^e - \bar{g}_n^e\|_2 + \langle \theta_* - \theta(n), \bar{g}_n^e - \bar{g}_*^e \rangle,
\end{aligned} \tag{G.6}
$$

where we use the fact that $\theta(n), \theta_* \in \mathcal{B}(\theta_0, R_{\mathrm{a}})$ in the last inequality. Further, by the definitions in (G.2), it holds that

$$
\begin{aligned}
\langle \theta_* - \theta(n), \bar{g}_n^e - \bar{g}_*^e \rangle &= \mathbb{E}_{\rho_{\pi_\theta}} \big[ (\bar{f}_{\theta(n)} - \bar{f}_{\theta_*}) \cdot \langle \theta_* - \theta(n), \nabla_\theta f_{\theta_0} \rangle \big] \\
&= \mathbb{E}_{\rho_{\pi_\theta}} \big[ (\bar{f}_{\theta(n)} - \bar{f}_{\theta_*}) \cdot (\bar{f}_{\theta_*} - \bar{f}_{\theta(n)}) \big] \\
&= -\mathbb{E}_{\rho_{\pi_\theta}} \big[ (\bar{f}_{\theta(n)} - \bar{f}_{\theta_*})^2 \big],
\end{aligned} \tag{G.7}
$$

where we use (G.1) in the second equality. Combining (G.6) and (G.7), we obtain the following upper bound of term (i),

$$
\langle \theta_* - \theta(n), g_n^e - \bar{g}_*^e \rangle \leq 2R_{\mathrm{a}} \cdot \|g_n^e - \bar{g}_n^e\|_2 - \mathbb{E}_{\rho_{\pi_\theta}} \big[ (\bar{f}_{\theta(n)} - \bar{f}_{\theta_*})^2 \big]. \tag{G.8}
$$

**Upper Bound of Term (ii).** We now upper bound term (ii) on the RHS of (G.5). It holds by Cauchy-Schwarz inequality that

$$
\begin{aligned}
\mathbb{E}_{\rho_{\pi_\theta}} \big[ \|g_n - \bar{g}_*^e\|_2^2 \,|\, \theta(n) \big] &\leq 2\mathbb{E}_{\rho_{\pi_\theta}} \big[ \|g_n - g_n^e\|_2^2 \,|\, \theta(n) \big] + 2\|g_n^e - \bar{g}_*^e\|_2^2 \\
&\leq 2 \underbrace{\mathbb{E}_{\rho_{\pi_\theta}} \big[ \|g_n - g_n^e\|_2^2 \,|\, \theta(n) \big]}_{\text{(ii).a}} + 4 \underbrace{\|g_n^e - \bar{g}_n^e\|_2^2}_{\text{(ii).b}} + 4 \underbrace{\|\bar{g}_n^e - \bar{g}_*^e\|_2^2}_{\text{(ii).c}}.
\end{aligned} \tag{G.9}
$$

We upper bound term (ii).a, term (ii).b, and term (ii).c in the sequel.

**Upper Bound of Term (ii).a.** Note that

$$
\mathbb{E}_{\rho_{\pi_\theta}} \big[ \|g_n - g_n^e\|_2^2 \,|\, \theta(n) \big] = \mathbb{E}_{\rho_{\pi_\theta}} \big[ \|g_n\|_2^2 - \|g_n^e\|_2^2 \,|\, \theta(n) \big] \leq \mathbb{E}_{\rho_{\pi_\theta}} \big[ \|g_n\|_2^2 \,|\, \theta(n) \big]. \tag{G.10}
$$

Meanwhile, by the definition of $g_n$ in (G.2), it holds that

$$
\|g_n\|_2^2 = \big( f_{\theta(n)} - \widetilde{\tau} \cdot (\beta^{-1} Q_\omega + \tau^{-1} f_\theta) \big)^2 \cdot \|\nabla_\theta f_{\theta(n)}\|_2^2. \tag{G.11}
$$

We first upper bound $f_\theta$ as follows,

$$
f_\theta^2 = x^{(H_{\mathrm{a}})\top} b b^\top x^{(H_{\mathrm{a}})} = x^{(H_{\mathrm{a}})\top} x^{(H_{\mathrm{a}})} = \|x^{(H_{\mathrm{a}})}\|_2^2,
$$

where $x^{(H_{\mathrm{a}})}$ is the output of the $H_{\mathrm{a}}$-th layer of the DNN $f_\theta$. Further combining Lemma F.4, it holds with probability at least $1 - O(H_{\mathrm{a}}) \exp(-\Omega(H_{\mathrm{a}}^{-1} m_{\mathrm{a}}))$ that

$$
|f_\theta| \leq 2. \tag{G.12}
$$

Following from similar arguments, with probability at least $1 - O(H_{\mathrm{a}}) \exp(-\Omega(H_{\mathrm{a}}^{-1} m_{\mathrm{a}}))$, we have

$$
|Q_\omega| \leq 2, \qquad |f_{\theta(n)}| \leq 2. \tag{G.13}
$$

Combining Lemma F.2, (G.10), (G.11), (G.12), and (G.13), it holds with probability at least $1 - \exp(-\Omega(R_{\mathrm{a}}^{2/3} m_{\mathrm{a}}^{2/3} H_{\mathrm{a}}))$ that

$$
\mathbb{E}_{\rho_{\pi_\theta}} \big[ \|g_n - g_n^e\|_2^2 \,|\, \theta(n) \big] = O(H_{\mathrm{a}}^2), \tag{G.14}
$$

which establishes an upper bound of term (ii).a.

**Upper Bound of Term (ii).b.** It holds that

$$
\begin{aligned}
\|g_n^e - \bar{g}_n^e\|_2 &= \big\| \mathbb{E}_{\rho_{\pi_\theta}} \big[ \big( f_{\theta(n)} - \widetilde{\tau} \cdot (\beta^{-1} Q_\omega + \tau^{-1} f_\theta) \big) \cdot \nabla_\theta f_{\theta(n)} \\
&\qquad - \big( \bar{f}_{\theta(n)} - \widetilde{\tau} \cdot (\beta^{-1} Q_\omega + \tau^{-1} f_\theta) \big) \cdot \nabla_\theta f_{\theta_0} \big] \big\|_2 \\
&\leq \mathbb{E}_{\rho_{\pi_\theta}} \big[ \|f_{\theta(n)} \nabla_\theta f_{\theta(n)} - \bar{f}_{\theta(n)} \nabla_\theta f_{\theta_0}\|_2 \big] \\
&\qquad + \widetilde{\tau} \cdot \mathbb{E}_{\rho_{\pi_\theta}} \big[ \|(\beta^{-1} Q_\omega + \tau^{-1} f_\theta) \cdot (\nabla_\theta f_{\theta_0} - \nabla_\theta f_{\theta(n)})\|_2 \big] \\
&\leq \mathbb{E}_{\rho_{\pi_\theta}} \big[ \|f_{\theta(n)} \nabla_\theta f_{\theta_0} - \bar{f}_{\theta(n)} \nabla_\theta f_{\theta_0}\|_2 \big] + \mathbb{E}_{\rho_{\pi_\theta}} \big[ \|f_{\theta(n)} \nabla_\theta f_{\theta(n)} - f_{\theta(n)} \nabla_\theta f_{\theta_0}\|_2 \big] \\
&\qquad + \mathbb{E}_{\rho_{\pi_\theta}} \big[ \|\widetilde{\tau} \cdot (\beta^{-1} Q_\omega + \tau^{-1} f_\theta) \cdot (\nabla_\theta f_{\theta_0} - \nabla_\theta f_{\theta(n)})\|_2 \big].
\end{aligned} \tag{G.15}
$$

We upper bound the three terms on the RHS of (G.15) in the sequel, respectively.

For the term $\|f_{\theta(n)}\nabla_\theta f_{\theta_0} - \bar{f}_{\theta(n)}\nabla_\theta f_{\theta_0}\|_2$ on the RHS of (G.15), following from Lemmas F.2 and F.3, it holds with probability at least $1 - \exp(-\Omega(R_a^{2/3}m_a^{2/3}H_a))$ that

$$\|f_{\theta(n)}\nabla_\theta f_{\theta_0} - \bar{f}_{\theta(n)}\nabla_\theta f_{\theta_0}\|_2 = O\big(R_a^{4/3}m_a^{-1/6}H_a^{7/2}(\log m_a)^{1/2}\big). \tag{G.16}$$

For the term $\|f_{\theta(n)}\nabla_\theta f_{\theta(n)} - f_{\theta(n)}\nabla_\theta f_{\theta_0}\|_2$ on the RHS of (G.15), following from (G.13) and Lemma F.2, with probability at least $1 - \exp(-\Omega(R_a^{2/3}m_a^{2/3}H_a))$, we have

$$\|f_{\theta(n)}\nabla_\theta f_{\theta(n)} - f_{\theta(n)}\nabla_\theta f_{\theta_0}\|_2 = O\big(R_a^{1/3}m_a^{-1/6}H_a^{5/2}(\log m_a)^{1/2}\big). \tag{G.17}$$

For the term $\|\widetilde{\tau}\cdot(\beta^{-1}Q_\omega + \tau^{-1}f_\theta)\cdot(\nabla_\theta f_{\theta_0} - \nabla_\theta f_{\theta(n)})\|_2$ on the RHS of (G.15), we first upper bound $\widetilde{\tau}\cdot(\beta^{-1}Q_\omega + \tau^{-1}f_\theta)$ as follows,

$$|\widetilde{\tau}\cdot(\beta^{-1}Q_\omega + \tau^{-1}f_\theta)| \le 2,$$

where we use (G.12), (G.13), and the fact that $\widetilde{\tau}^{-1} = \beta^{-1} + \tau^{-1}$. Further combining Lemma F.2, it holds with probability at least $1 - \exp(-\Omega(R_a^{2/3}m_a^{2/3}H_a))$ that

$$\|\widetilde{\tau}\cdot(\beta^{-1}Q_\omega + \tau^{-1}f_\theta)\cdot(\nabla_\theta f_{\theta_0} - \nabla_\theta f_{\theta(n)})\|_2 = O\big(R_a^{1/3}m_a^{-1/6}H_a^{5/2}(\log m_a)^{1/2}\big). \tag{G.18}$$

Now, combining (G.15), (G.16), (G.17), and (G.18), it holds with probability at least $1 - \exp(-\Omega(R_a^{2/3}m_a^{2/3}H_a))$ that

$$\|g_n^e - \bar{g}_n^e\|_2^2 = O\big(R_a^{8/3}m_a^{-1/3}H_a^7\log m_a\big), \tag{G.19}$$

which establishes an upper bound of term (ii).b.

**Upper Bound of Term (ii).c.** It holds that

$$\|\bar{g}_n^e - \bar{g}_*^e\|_2^2 = \big\|\mathbb{E}_{\rho_{\pi_\theta}}[(\bar{f}_{\theta(n)} - \bar{f}_{\theta_*})\nabla_\theta f_{\theta_0}]\big\|_2^2 \le \mathbb{E}_{\rho_{\pi_\theta}}\big[(\bar{f}_{\theta(n)} - \bar{f}_{\theta_*})^2\cdot\|\nabla_\theta f_{\theta_0}\|_2^2\big].$$

Further combining Lemma F.2, it holds with probability at least $1 - \exp(-\Omega(R_a^{2/3}m_a^{2/3}H_a))$ that

$$\|\bar{g}_n^e - \bar{g}_*^e\|_2^2 \le O(H_a^2)\cdot\mathbb{E}_{\rho_{\pi_\theta}}\big[(\bar{f}_{\theta(n)} - \bar{f}_{\theta_*})^2\big], \tag{G.20}$$

which establishes an upper bound of term (ii).c.

Now, combining (G.9), (G.14), (G.19), and (G.20), we have

$$\mathbb{E}_{\rho_{\pi_\theta}}\big[\|g_n - \bar{g}_*^e\|_2^2\,|\,\theta(n)\big] \le O\big(R_a^{8/3}m_a^{-1/3}H_a^7\log m_a\big) + O(H_a^2)\cdot\mathbb{E}_{\rho_{\pi_\theta}}\big[(\bar{f}_{\theta(n)} - \bar{f}_{\theta_*})^2\big], \tag{G.21}$$

which is an upper bound of term (ii) on the RHS of (G.5).

By plugging the upper bound of term (i) in (G.8) and the upper bound of term (ii) in (G.21) into (G.5), combining (G.19), with probability at least $1 - \exp(-\Omega(R_a^{2/3}m_a^{2/3}H_a))$, we have

$$\mathbb{E}_{\rho_{\pi_\theta}}\big[\|\theta(n+1) - \theta_*\|_2^2\,|\,\theta(n)\big]$$
$$\le \|\theta(n) - \theta_*\|_2^2 + 2\alpha\cdot\Big(O\big(R_a^{7/3}m_a^{-1/6}H_a^{7/2}(\log m_a)^{1/2}\big) - \mathbb{E}_{\rho_{\pi_\theta}}\big[(\bar{f}_{\theta(n)} - \bar{f}_{\theta_*})^2\big]\Big) \tag{G.22}$$
$$+ \alpha^2\cdot\Big(O\big(R_a^{8/3}m_a^{-1/3}H_a^7\log m_a\big) + O(H_a^2)\cdot\mathbb{E}_{\rho_{\pi_\theta}}\big[(\bar{f}_{\theta(n)} - \bar{f}_{\theta_*})^2\big]\Big).$$

Rearranging terms in (G.22), it holds with probability at least $1 - \exp(-\Omega(R_a^{2/3}m_a^{2/3}H_a))$ that

$$(2\alpha - \alpha^2\cdot O(H_a^2))\cdot\mathbb{E}_{\rho_{\pi_\theta}}\big[(\bar{f}_{\theta(n)} - \bar{f}_{\theta_*})^2\big]$$
$$\le \|\theta(n) - \theta_*\|_2^2 - \mathbb{E}_{\rho_{\pi_\theta}}\big[\|\theta(n+1) - \theta_*\|_2^2\,|\,\theta(n)\big] + \alpha\cdot O\big(R_a^{8/3}m_a^{-1/6}H_a^7\log m_a\big). \tag{G.23}$$

By telescoping the sum and using Jensen's inequality in (G.23), we have

$$\mathbb{E}_{\rho_{\pi_\theta}}\big[(\bar{f}_{\bar{\theta}} - \bar{f}_{\theta_*})^2\big] \leq \frac{1}{N_{\mathrm{a}}} \cdot \sum_{n=0}^{N_{\mathrm{a}}-1} \mathbb{E}_{\rho_{\pi_\theta}}\big[(\bar{f}_{\theta(n)} - \bar{f}_{\theta_*})^2\big]$$

$$\leq 1/N_{\mathrm{a}} \cdot \big(2\alpha - \alpha^2 \cdot O(H_{\mathrm{a}}^2)\big)^{-1} \cdot \big(\|\theta_0 - \theta_*\|_2^2 + \alpha N_{\mathrm{a}} \cdot O(R_{\mathrm{a}}^{8/3} m_{\mathrm{a}}^{-1/6} H_{\mathrm{a}}^7 \log m_{\mathrm{a}})\big)$$

$$\leq N_{\mathrm{a}}^{-1/2} \cdot \|\theta_0 - \theta_*\|_2^2 + O(R_{\mathrm{a}}^{8/3} m_{\mathrm{a}}^{-1/6} H_{\mathrm{a}}^7 \log m_{\mathrm{a}}),$$

where the last line comes from the choices that $\alpha = N_{\mathrm{a}}^{-1/2}$ and $H_{\mathrm{a}} = O(N_{\mathrm{a}}^{1/4})$. Further combining Lemma F.3 and using triangle inequality, we have

$$\mathbb{E}_{\rho_{\pi_\theta}}\big[(f_{\bar{\theta}} - \bar{f}_{\theta_*})^2\big] = O(R_{\mathrm{a}}^2 N_{\mathrm{a}}^{-1/2} + R_{\mathrm{a}}^{8/3} m_{\mathrm{a}}^{-1/6} H_{\mathrm{a}}^7 \log m_{\mathrm{a}}). \tag{G.24}$$

By the definition of $\theta_*$ in (G.3), we know that

$$\langle \bar{g}_*^e, \theta - \theta_* \rangle \geq 0, \qquad \text{for any } \theta \in \mathcal{B}(\theta_0, R_{\mathrm{a}}). \tag{G.25}$$

By plugging the definition of $\bar{g}_*^e$ into (G.25), we have

$$\mathbb{E}_{\rho_{\pi_\theta}}\big[\langle \bar{f}_{\theta_*} - \widetilde{\tau} \cdot (\beta^{-1}Q_\omega + \tau^{-1}f_\theta), \bar{f}_{\theta^\dagger} - \bar{f}_{\theta_*}\rangle\big] \geq 0, \qquad \text{for any } \theta^\dagger \in \mathcal{B}(\theta_0, R_{\mathrm{a}}),$$

which is equivalent to

$$\theta_* = \operatorname*{argmin}_{\theta^\dagger \in \mathcal{B}(\theta_0, R_{\mathrm{a}})} \mathbb{E}_{\rho_{\pi_\theta}}\big[(\bar{f}_{\theta^\dagger} - \widetilde{\tau} \cdot (\beta^{-1}Q_\omega + \tau^{-1}f_\theta))^2\big]. \tag{G.26}$$

Meanwhile, by the fact that $\theta_0 = \omega_0$, we have

$$\widetilde{\tau} \cdot (\beta^{-1}\bar{Q}_\omega + \tau^{-1}\bar{f}_\theta) = \widetilde{\tau} \cdot \big(\beta^{-1} \cdot (Q_{\omega_0} + (\omega - \omega_0)^\top \nabla_\omega Q_{\omega_0}) + \tau^{-1} \cdot (f_{\theta_0} + (\theta - \theta_0)^\top \nabla_\theta f_{\theta_0})\big)$$

$$= f_{\theta_0} + \big(\widetilde{\tau} \cdot (\beta^{-1}\omega + \tau^{-1}\theta) - \theta_0\big)^\top \nabla_\theta f_{\theta_0},$$

where the second line comes from $\widetilde{\tau}^{-1} = \beta^{-1} + \tau^{-1}$. Note that $\theta \in \mathcal{B}(\theta_0, R_{\mathrm{a}})$, $\omega \in \mathcal{B}(\omega_0, R_{\mathrm{c}})$, $\theta_0 = \omega_0$, and $R_{\mathrm{a}} = R_{\mathrm{c}}$, we know that $\widetilde{\tau} \cdot (\beta^{-1}\omega + \tau^{-1}\theta) \in \mathcal{B}(\theta_0, R_{\mathrm{a}})$. Therefore, with probability at least $1 - \exp(-\Omega(R_{\mathrm{a}}^{2/3} m_{\mathrm{a}}^{2/3} H_{\mathrm{a}}))$ we have

$$\mathbb{E}_{\rho_{\pi_\theta}}\big[(\bar{f}_{\theta_*} - \widetilde{\tau} \cdot (\beta^{-1}Q_\omega + \tau^{-1}f_\theta))^2\big]$$

$$\leq \mathbb{E}_{\rho_{\pi_\theta}}\big[(\widetilde{\tau} \cdot (\beta^{-1}\bar{Q}_\omega + \tau^{-1}\bar{f}_\theta) - \widetilde{\tau} \cdot (\beta^{-1}Q_\omega + \tau^{-1}f_\theta))^2\big]$$

$$\leq \widetilde{\tau}^2 \cdot \beta^{-2} \cdot \mathbb{E}_{\rho_{\pi_\theta}}[(\bar{Q}_\omega - Q_\omega)^2] + \widetilde{\tau}^2 \cdot \tau^{-2} \cdot \mathbb{E}_{\rho_{\pi_\theta}}[(\bar{f}_\theta - f_\theta)^2]$$

$$= O(R_{\mathrm{a}}^{8/3} m_{\mathrm{a}}^{-1/3} H_{\mathrm{a}}^5 \log m_{\mathrm{a}}), \tag{G.27}$$

where the first inequality comes from (G.26), and the last inequality comes from Lemma F.3 and the fact that $R_{\mathrm{c}} = R_{\mathrm{a}}$, $m_{\mathrm{c}} = m_{\mathrm{a}}$, and $H_{\mathrm{c}} = H_{\mathrm{a}}$. Combining (G.24) and (G.27), by triangle inequality, we have

$$\mathbb{E}_{\rho_{\pi_\theta}}\big[(f_{\bar{\theta}}(s, a) - \widetilde{\tau} \cdot (\beta^{-1}Q_\omega(s, a) + \tau^{-1}f_\theta(s, a)))^2\big] = O(R_{\mathrm{a}}^2 N_{\mathrm{a}}^{-1/2} + R_{\mathrm{a}}^{8/3} m_{\mathrm{a}}^{-1/6} H_{\mathrm{a}}^7 \log m_{\mathrm{a}}),$$

which finishes the proof of Proposition C.3.

## G.3    PROOF OF PROPOSITION C.4

The proof is similar to that of Proposition C.3 in §G.2. For the completeness of the paper, we present it here. We define the local linearization of $Q_\omega$ as follows,

$$\bar{Q}_\omega = Q_{\omega_0} + (\omega - \omega_0)^\top \nabla_{\omega_0} Q_\omega. \tag{G.28}$$

We denote by

$$g_n = \big(Q_{\omega(n)}(s_0, a_0) - \gamma \cdot Q_\omega(s_1, a_1) - (1 - \gamma) \cdot r_0\big) \cdot \nabla_\omega Q_{\omega(n)}(s_0, a_0), \qquad g_n^e = \mathbb{E}_{\pi_\theta}[g_n],$$

$$\bar{g}_n = \big(\bar{Q}_{\omega(n)}(s_0, a_0) - \gamma \cdot Q_\omega(s_1, a_1) - (1 - \gamma) \cdot r_0\big) \cdot \nabla_\omega Q_{\omega_0}(s_0, a_0), \qquad \bar{g}_n^e = \mathbb{E}_{\pi_\theta}[\bar{g}_n],$$

$$g_* = \big(Q_{\omega_*}(s_0, a_0) - \gamma \cdot Q_\omega(s_1, a_1) - (1 - \gamma) \cdot r_0\big) \cdot \nabla_\omega Q_{\omega_*}(s_0, a_0), \qquad g_*^e = \mathbb{E}_{\pi_\theta}[g_*],$$

$$\bar{g}_* = \big(\bar{Q}_{\omega_*}(s_0, a_0) - \gamma \cdot Q_\omega(s_1, a_1) - (1 - \gamma) \cdot r_0\big) \cdot \nabla_\omega Q_{\omega_0}(s_0, a_0), \qquad \bar{g}_*^e = \mathbb{E}_{\pi_\theta}[\bar{g}_*],$$

$$\tag{G.29}$$

where $\omega_*$ satisfies that

$$\omega_* = \Gamma_{\mathcal{B}(\omega_0, R_c)}(\omega_* - \alpha \cdot \bar{g}_*^e). \tag{G.30}$$

Here the expectation $\mathbb{E}_{\pi_\theta}[\cdot]$ is taken following $(s_0, a_0) \sim \rho_{\pi_\theta}(\cdot)$, $s_1 \sim P(\cdot \mid s_0, a_0)$, $a_1 \sim \pi_\theta(\cdot \mid s_1)$, and $r_0 = r(s_0, a_0)$. By Algorithm 4, we know that

$$\omega(n+1) = \Gamma_{\mathcal{B}(\omega_0, R_c)}(\omega(n) - \eta \cdot g_n).$$

Note that

$$\mathbb{E}_{\pi_\theta}\big[\|\omega(n+1) - \omega_*\|_2^2 \,|\, \omega(n)\big]$$
$$= \mathbb{E}_{\pi_\theta}\big[\|\Gamma_{\mathcal{B}(\omega_0, R_c)}(\omega(n) - \eta \cdot g_n) - \Gamma_{\mathcal{B}(\omega_0, R_c)}(\omega_* - \eta \cdot \bar{g}_*^e)\|_2^2 \,|\, \omega(n)\big]$$
$$\leq \mathbb{E}_{\pi_\theta}\big[\|(\omega(n) - \eta \cdot g_n) - (\omega_* - \eta \cdot \bar{g}_*^e)\|_2^2 \,|\, \omega(n)\big]$$
$$= \|\omega(n) - \omega_*\|_2^2 + 2\eta \cdot \underbrace{\langle \omega_* - \omega(n), g_n^e - \bar{g}_*^e \rangle}_{\text{(iii)}} + \eta^2 \cdot \underbrace{\mathbb{E}_{\pi_\theta}\big[\|g_n - \bar{g}_*^e\|_2^2 \,|\, \omega(n)\big]}_{\text{(iv)}}. \tag{G.31}$$

We upper bound term (iii) and term (iv) on the RHS of (G.31) in the sequel.

**Upper Bound of Term (iii).** By Hölder's inequality, it holds that

$$\langle \omega_* - \omega(n), g_n^e - \bar{g}_*^e \rangle$$
$$= \langle \omega_* - \omega(n), g_n^e - \bar{g}_n^e \rangle + \langle \omega_* - \omega(n), \bar{g}_n^e - \bar{g}_*^e \rangle$$
$$\leq \|\omega_* - \omega(n)\|_2 \cdot \|g_n^e - \bar{g}_n^e\|_2 + \langle \omega_* - \omega(n), \bar{g}_n^e - \bar{g}_*^e \rangle$$
$$\leq 2R_c \cdot \|g_n^e - \bar{g}_n^e\|_2 + \langle \omega_* - \omega(n), \bar{g}_n^e - \bar{g}_*^e \rangle, \tag{G.32}$$

where we use the fact that $\omega(n), \omega_* \in \mathcal{B}(\omega_0, R_c)$ in the last line. Further, by the definitions in (G.29), it holds that

$$\langle \omega_* - \omega(n), \bar{g}_n^e - \bar{g}_*^e \rangle$$
$$= \mathbb{E}_{\pi_\theta}\big[(\bar{Q}_{\omega(n)}(s_0, a_0) - \bar{Q}_{\omega_*}(s_0, a_0)) \cdot \langle \omega_* - \omega(n), \nabla_\omega Q_{\omega_0}(s_0, a_0) \rangle\big]$$
$$= \mathbb{E}_{\pi_\theta}\big[(\bar{Q}_{\omega(n)}(s_0, a_0) - \bar{Q}_{\omega_*}(s_0, a_0)) \cdot (\bar{Q}_{\omega_*}(s_0, a_0) - \bar{Q}_{\omega(n)}(s_0, a_0))\big]$$
$$= -\mathbb{E}_{\pi_\theta}\big[(\bar{Q}_{\omega(n)}(s_0, a_0) - \bar{Q}_{\omega_*}(s_0, a_0))^2\big] = -\mathbb{E}_{\rho_{\pi_\theta}}\big[(\bar{Q}_{\omega(n)} - \bar{Q}_{\omega_*})^2\big], \tag{G.33}$$

where the second equality comes from (G.28), and the last equality comes from the fact that the expectation is only taken to the state-action pair $(s_0, a_0)$. Combining (G.32) and (G.33), we obtain the following upper bound of term (i),

$$\langle \omega_* - \omega(n), g_n^e - \bar{g}_*^e \rangle \leq 2R_c \cdot \|g_n^e - \bar{g}_n^e\|_2 - \mathbb{E}_{\rho_{\pi_\theta}}\big[(\bar{Q}_{\omega(n)} - \bar{Q}_{\omega_*})^2\big]. \tag{G.34}$$

**Upper Bound of Term (iv).** We now upper bound term (iv) on the RHS of (G.31). It holds by Cauchy-Schwarz inequality that

$$\mathbb{E}_{\pi_\theta}\big[\|g_n - \bar{g}_*^e\|_2^2 \,|\, \omega(n)\big]$$
$$\leq 2\mathbb{E}_{\pi_\theta}\big[\|g_n - g_n^e\|_2^2 \,|\, \omega(n)\big] + 2\|g_n^e - \bar{g}_*^e\|_2^2$$
$$\leq 2\underbrace{\mathbb{E}_{\pi_\theta}\big[\|g_n - g_n^e\|_2^2 \,|\, \omega(n)\big]}_{\text{(iv).a}} + 4\underbrace{\|g_n^e - \bar{g}_n^e\|_2^2}_{\text{(iv).b}} + 4\underbrace{\|\bar{g}_n^e - \bar{g}_*^e\|_2^2}_{\text{(iv).c}}. \tag{G.35}$$

We upper bound term (iv).a, term (iv).b, and term (iv).c in the sequel.

**Upper Bound of Term (iv).a.** We now upper bound term (iv).a on the RHS of (G.35). By expanding the square, we have

$$\mathbb{E}_{\pi_\theta}\big[\|g_n - g_n^e\|_2^2 \,|\, \omega(n)\big] = \mathbb{E}_{\pi_\theta}\big[\|g_n\|_2^2 - \|g_n^e\|_2^2 \,|\, \omega(n)\big] \leq \mathbb{E}_{\pi_\theta}\big[\|g_n\|_2^2 \,|\, \omega(n)\big]. \tag{G.36}$$

Meanwhile, by the definition of $g_n$ in (G.29), it holds that

$$\|g_n\|_2^2 = \big(Q_{\omega(n)}(s_0, a_0) - \gamma \cdot Q_\omega(s_1, a_1) - (1-\gamma) \cdot r_0\big)^2 \cdot \|\nabla_\omega Q_{\omega(n)}(s_0, a_0)\|_2^2. \tag{G.37}$$

We first upper bound $Q_\omega$ as follows,

$$Q_\omega^2 = x^{(H_c)\top} bb^\top x^{(H_c)} = x^{(H_c)\top} x^{(H_c)} = \|x^{(H_c)}\|_2^2,$$

where $x^{(H_c)}$ is the output of the $H_c$-th layer of the DNN $Q_\omega$. Further combining Lemma F.4, it holds that

$$|Q_\omega| \le 2. \tag{G.38}$$

Similarly, we have

$$|Q_{\omega(n)}| \le 2. \tag{G.39}$$

Combining Lemma F.2, (G.36), (G.37), (G.38), and (G.39), we have

$$\mathbb{E}_{\pi_\theta}\big[\|g_n - g_n^e\|_2^2 \,|\, \omega(n)\big] = O(H_c^2). \tag{G.40}$$

**Upper Bound of Term (iv).b.** We now upper bound term (iv).b on the RHS of (G.35). It holds that

$$\begin{aligned}
&\|g_n^e - \bar{g}_n^e\|_2 \\
&= \big\|\mathbb{E}_{\pi_\theta}\big[\big(Q_{\omega(n)}(s_0,a_0) - \gamma \cdot Q_\omega(s_1,a_1) - (1-\gamma) \cdot r_0\big) \cdot \nabla_\omega Q_{\omega(n)}(s_0,a_0) \\
&\qquad\quad - \big(\bar{Q}_{\omega(n)}(s_0,a_0) - \gamma \cdot Q_\omega(s_1,a_1) - (1-\gamma) \cdot r_0\big) \cdot \nabla_\omega Q_{\omega_0}(s_0,a_0)\big]\big\|_2 \\
&\le \mathbb{E}_{\pi_\theta}\big[\big\|\big(\gamma \cdot Q_\omega(s_1,a_1) + (1-\gamma) \cdot r_t\big) \cdot (\nabla_\omega Q_{\omega_0}(s_0,a_0) - \nabla_\omega Q_{\omega(n)}(s_0,a_0))\big\|_2\big] \\
&\quad + \mathbb{E}_{\rho_{\pi_\theta}}\big[\|Q_{\omega(n)}\nabla_\omega Q_{\omega(n)} - \bar{Q}_{\omega(n)}\nabla_\omega Q_{\omega_0}\|_2\big] \\
&\le \mathbb{E}_{\pi_\theta}\big[\big\|\big(\gamma \cdot Q_\omega(s_1,a_1) + (1-\gamma) \cdot r_0\big) \cdot (\nabla_\omega Q_{\omega_0}(s_0,a_0) - \nabla_\omega Q_{\omega(n)}(s_0,a_0))\big\|_2\big] \quad \text{(G.41)} \\
&\quad + \mathbb{E}_{\rho_{\pi_\theta}}\big[\|(Q_{\omega(n)} - \bar{Q}_{\omega(n)}) \cdot \nabla_\omega Q_{\omega_0}\|_2\big] + \mathbb{E}_{\rho_{\pi_\theta}}\big[\|Q_{\omega(n)} \cdot (\nabla_\omega Q_{\omega(n)} - \nabla_\omega Q_{\omega_0})\|_2\big].
\end{aligned}$$

We now upper bound the three terms on the RHS of (G.41) in the sequel, respectively.

For the term $\mathbb{E}_{\rho_{\pi_\theta}}[\|(Q_{\omega(n)} - \bar{Q}_{\omega(n)}) \cdot \nabla_\omega Q_{\omega_0}\|_2]$ on the RHS of (G.41), following from Lemmas F.2 and F.3, it holds with probability at least $1 - \exp(-\Omega(R_c^{2/3}m_c^{2/3}H_c))$ that

$$\mathbb{E}_{\rho_{\pi_\theta}}\big[\|(Q_{\omega(n)} - \bar{Q}_{\omega(n)}) \cdot \nabla_\omega Q_{\omega_0}\|_2\big] = O\big(R_c^{4/3}m_c^{-1/6}H_c^{7/2}(\log m_c)^{1/2}\big). \tag{G.42}$$

For the term $\mathbb{E}_{\rho_{\pi_\theta}}[\|Q_{\omega(n)} \cdot (\nabla_\omega Q_{\omega(n)} - \nabla_\omega Q_{\omega_0})\|_2]$ on the RHS of (G.41), following from (G.39) and Lemma F.2, with probability at least $1 - \exp(-\Omega(R_c^{2/3}m_c^{2/3}H_c))$, we have

$$\mathbb{E}_{\rho_{\pi_\theta}}\big[\|Q_{\omega(n)} \cdot (\nabla_\omega Q_{\omega(n)} - \nabla_\omega Q_{\omega_0})\|_2\big] = O\big(R_c^{1/3}m_c^{-1/6}H_c^{5/2}(\log m_c)^{1/2}\big). \tag{G.43}$$

For the term $\mathbb{E}_{\pi_\theta}[\|(\gamma \cdot Q_\omega(s_1,a_1) + (1-\gamma) \cdot r_0) \cdot (\nabla_\omega Q_{\omega_0}(s_0,a_0) - \nabla_\omega Q_{\omega(n)}(s_0,a_0))\|_2]$ on the RHS of (G.41), we first upper bound $|\gamma \cdot Q_\omega(s_1,a_1) + (1-\gamma) \cdot r_0|$ as follows,

$$|\gamma \cdot Q_\omega(s_1,a_1) + (1-\gamma) \cdot r_0| \le 2 + r_{\max},$$

where we use (G.38) and the fact that $|r(s,a)| \le r_{\max}$ for any $(s,a) \in \mathcal{S} \times \mathcal{A}$. Further combining Lemma F.2, with probability at least $1 - \exp(-\Omega(R_c^{2/3}m_c^{2/3}H_c))$, we have

$$\begin{aligned}
&\mathbb{E}_{\pi_\theta}\big[\big\|\big(\gamma \cdot Q_\omega(s_1,a_1) + (1-\gamma) \cdot r_0\big) \cdot (\nabla_\omega Q_{\omega_0}(s_0,a_0) - \nabla_\omega Q_{\omega(n)}(s_0,a_0))\big\|_2\big] \\
&\quad = O\big(R_c^{1/3}m_c^{-1/6}H_c^{5/2}(\log m_c)^{1/2}\big). \tag{G.44}
\end{aligned}$$

Now, combining (G.41), (G.42), (G.43), and (G.44), it holds with probability at least $1 - \exp(-\Omega(R_c^{2/3}m_c^{2/3}H_c))$ that

$$\|g_n^e - \bar{g}_n^e\|_2^2 = O(R_c^{8/3}m_c^{-1/3}H_c^7 \log m_c). \tag{G.45}$$

**Upper Bound of Term (iv).c.** We now upper bound term (iv).c on the RHS of (G.35). It holds that

$$\|\bar{g}_n^e - \bar{g}_*^e\|_2^2 = \big\|\mathbb{E}_{\rho_{\pi_\theta}}[(\bar{Q}_{\omega(n)} - \bar{Q}_{\omega_*})\nabla_\omega Q_{\omega_0}]\big\|_2^2 \le \mathbb{E}_{\rho_{\pi_\theta}}\big[(\bar{Q}_{\omega(n)} - \bar{Q}_{\omega_*})^2 \cdot \|\nabla_\omega Q_{\omega_0}\|_2^2\big].$$

Further combining Lemma F.2, it holds that

$$\mathbb{E}_{\pi_\theta}\big[\|\bar{g}_n^e - \bar{g}_*^e\|_2^2 \,|\, \omega(n)\big] \le O(H_c^2) \cdot \mathbb{E}_{\rho_{\pi_\theta}}\big[(\bar{Q}_{\omega(n)} - \bar{Q}_{\omega_*})^2\big]. \tag{G.46}$$

Combining (G.35), (G.40), (G.45), and (G.46), we obtain the following upper bound for term (iv) on the RHS of (G.31),

$$\mathbb{E}_{\pi_\theta}\big[\|g_n - \bar{g}_*^e\|_2^2 \,|\, \omega(n)\big] \le O(R_c^{8/3} m_c^{-1/3} H_c^7 \log m_c) + O(H_c^2) \cdot \mathbb{E}_{\rho_{\pi_\theta}}\big[(\bar{Q}_{\omega(n)} - \bar{Q}_{\omega_*})^2\big]. \tag{G.47}$$

We continue upper bounding (G.31). By plugging (G.34) and (G.47) into (G.31), it holds with probability at least $1 - \exp(-\Omega(R_c^{2/3} m_c^{2/3} H_c))$ that

$$\mathbb{E}_{\pi_\theta}\big[\|\omega(n+1) - \omega_*\|_2^2 \,|\, \omega(n)\big]$$
$$\le \|\omega(n) - \omega_*\|_2^2 + 2\eta \cdot \Big(O\big(R_c^{7/3} m_c^{-1/6} H_c^{7/2} (\log m_c)^{1/2}\big) - \mathbb{E}_{\rho_{\pi_\theta}}\big[(\bar{Q}_{\omega(n)} - \bar{Q}_{\omega_*})^2\big]\Big)$$
$$+ \eta^2 \cdot \Big(O\big(R_c^{8/3} m_c^{-1/3} H_c^7 \log m_c\big) + O(H_c^2) \cdot \mathbb{E}_{\rho_{\pi_\theta}}\big[(\bar{Q}_{\omega(n)} - \bar{Q}_{\omega_*})^2\big]\Big). \tag{G.48}$$

Rearranging terms in (G.48), it holds with probability at least $1 - \exp(-\Omega(R_c^{2/3} m_c^{2/3} H_c))$ that

$$(2\eta - \eta^2 \cdot O(H_c^2)) \cdot \mathbb{E}_{\rho_{\pi_\theta}}\big[(\bar{Q}_{\omega(n)} - \bar{Q}_{\omega_*})^2\big]$$
$$\le \|\omega(n) - \omega_*\|_2^2 - \mathbb{E}_{\rho_{\pi_\theta}}[\|\omega(n+1) - \omega_*\|_2^2 \,|\, \omega(n)] + \eta \cdot O(R_c^{8/3} m_c^{-1/3} H_c^7 \log m_c). \tag{G.49}$$

By telescoping the sum and using Jensen's inequality in (G.49), we have

$$\mathbb{E}_{\rho_{\pi_\theta}}\big[(\bar{Q}_{\bar{\omega}} - \bar{Q}_{\omega_*})^2\big] \le \frac{1}{N_c} \cdot \sum_{n=0}^{N_c-1} \mathbb{E}_{\rho_{\pi_\theta}}\big[(\bar{Q}_{\omega(n)} - \bar{Q}_{\omega_*})^2\big]$$
$$\le 1/N_c \cdot \big(2\eta - \eta^2 \cdot O(H_c^2)\big)^{-1} \cdot \big(\|\omega_0 - \omega_*\|_2^2 + \eta N_c \cdot O(R_c^{8/3} m_c^{-1/6} H_c^7 \log m_c)\big)$$
$$\le N_c^{-1/2} \cdot \|\theta_0 - \theta_*\|_2^2 + O(R_c^{8/3} m_c^{-1/6} H_c^7 \log m_c),$$

where the last line comes from the choices that $\eta = N_c^{-1/2}$ and $H_c = O(N_c^{1/4})$. Further combining Lemma F.3 and using triangle inequality, we have

$$\mathbb{E}_{\rho_{\pi_\theta}}\big[(Q_{\bar{\omega}} - \bar{Q}_{\omega_*})^2\big] = O(R_c^2 N_c^{-1/2} + R_c^{8/3} m_c^{-1/6} H_c^7 \log m_c). \tag{G.50}$$

To establish the upper bound of $\mathbb{E}_{\rho_{\pi_\theta}}[(\bar{Q}_{\omega_*} - \widetilde{Q})^2]$, we upper bound $\mathbb{E}_{\rho_{\pi_\theta}}[(\bar{Q}_{\omega_*} - \widetilde{Q})^2]$ in the sequel. By the definition of $\omega_*$ in (G.30), following a similar argument to derive (G.26), we have

$$\omega_* = \operatorname*{argmin}_{\omega^\dagger \in \mathcal{B}(\omega_0, R_c)} \mathbb{E}_{\rho_{\pi_\theta}}\big[(\bar{Q}_{\omega^\dagger}(s_0, a_0) - \widetilde{Q}(s_0, a_0))^2\big]. \tag{G.51}$$

From the fact that $\widetilde{Q} \in \mathcal{U}(m_c, H_c, R_c)$ by Assumption C.2, we know that $\widetilde{Q} = Q_{\widetilde{\omega}}$ for some $\widetilde{\omega} \in \mathcal{B}(\omega_0, R_c)$. Therefore, by (G.51), with probability at least $1 - \exp(-\Omega(R_c^{2/3} m_c^{2/3} H_c))$, we have

$$\mathbb{E}_{\rho_{\pi_\theta}}\big[(\bar{Q}_{\omega_*} - \widetilde{Q})^2\big] \le \mathbb{E}_{\rho_{\pi_\theta}}\big[(\bar{Q}_{\widetilde{\omega}} - \widetilde{Q})^2\big] = O(R_c^{8/3} m_c^{-1/3} H_c^5 \log m_c), \tag{G.52}$$

where we use Lemma F.3 in the last inequality. Now, combining (G.50) and (G.52), by triangle inequality, with probability at least $1 - \exp(-\Omega(R_c^{2/3} m_c^{2/3} H_c))$, we have

$$\mathbb{E}_{\rho_{\pi_\theta}}\big[(Q_{\bar{\omega}} - \widetilde{Q})^2\big] \le 2\mathbb{E}_{\rho_{\pi_\theta}}\big[(Q_{\bar{\omega}} - \bar{Q}_{\omega_*})^2\big] + 2\mathbb{E}_{\rho_{\pi_\theta}}\big[(\bar{Q}_{\omega_*} - \widetilde{Q})^2\big]$$
$$= O(R_c^2 N_c^{-1/2} + R_c^{8/3} m_c^{-1/6} H_c^7 \log m_c),$$

which concludes the proof of Proposition C.4.

# H   Proofs of Lemmas

## H.1   Proof of Lemma D.1

W denote by $\widetilde{Q} = \mathbb{T}^{\pi_{\theta_k}} Q_{\omega_k}$. In the sequel, we upper bound $\mathbb{E}_{\rho_{k+1}}[(Q_{\omega_{k+1}} - Q_{\bar{\omega}_{k+1}})^2]$, where $\bar{\omega}_{k+1} = \Gamma_R(\widetilde{\omega}_{k+1})$ and $\widetilde{\omega}_{k+1}$ is defined in (3.4). Note that by the fact that $\|\varphi(s,a)\|_2 \le 1$ uniformly, it suffices to upper bound $\|\omega_{k+1} - \widetilde{\omega}_{k+1}\|_2$. By the definitions of $\omega_{k+1}$ and $\widetilde{\omega}_{k+1}$ in (3.5) and (3.4), respectively, we have

$$\|\omega_{k+1} - \bar{\omega}_{k+1}\|_2 \le \|\widehat{\Phi}\widehat{v} - \Phi v\|_2 \le \|\Phi\|_2 \cdot \|\widehat{v} - v\|_2 + \|\widehat{\Phi} - \Phi\|_2 \cdot \|\widehat{v}\|_2. \tag{H.1}$$

Here, we use the fact that the projection $\Gamma_R(\cdot)$ is a contraction in the first inequality, and triangle inequality in the second inequality. Also, for notational convenience, we denote by $\widehat{\Phi}$, $\Phi$, $\widehat{v}$, and $v$ in (H.1) as follows,

$$\widehat{\Phi} = \Big(\frac{1}{N}\sum_{\ell=1}^{N} \varphi(s_{\ell,1}, a_{\ell,1})\varphi(s_{\ell,1}, a_{\ell,1})^{\top}\Big)^{-1}, \quad \Phi = \big(\mathbb{E}_{\rho_{k+1}}[\varphi(s,a)\varphi(s,a)^{\top}]\big)^{-1},$$

$$\widehat{v} = \frac{1}{N}\sum_{\ell=1}^{N}\big((1-\gamma)r_{\ell,2} + \gamma Q_{\omega_k}(s'_{\ell,2}, a'_{\ell,2})\big) \cdot \varphi(s_{\ell,2}, a_{\ell,2}),$$

$$v = \mathbb{E}_{\rho_{k+1}}\big[\big((1-\gamma)r + \gamma\mathbb{P}^{\pi_{\theta_{k+1}}}Q_{\omega_k}\big)(s,a) \cdot \varphi(s,a)\big].$$

By the fact that $\|\varphi(s,a)\|_2 \le 1$, $|r(s,a)| \le r_{\max}$, and $\|\omega_k\|_2 \le R$ we have

$$\|\Phi\|_2 \le 1/\sigma^*, \qquad \|\widehat{v}\|_2 \le r_{\max} + R. \tag{H.2}$$

Now, following from matrix Bernstein inequality (Tropp, 2015) and Assumption 4.3, we have

$$\mathbb{E}\big[\|\widehat{\Phi} - \Phi\|_2\big] \le \frac{2}{\sqrt{N}(\sigma^*)^2} \cdot \log(N + d), \tag{H.3}$$

where $\sigma^*$ is defined in Assumption 4.3. Similarly, we have

$$\mathbb{E}\big[\|\widehat{v} - v\|_2\big] \le 2(r_{\max} + R)/\sqrt{N} \cdot \log(N + d). \tag{H.4}$$

Now, combining (H.1), (H.2), (H.3), and (H.4), we have

$$\mathbb{E}\big[\|\omega_{k+1} - \bar{\omega}_{k+1}\|_2\big] \le \frac{4(r_{\max} + R)}{\sqrt{N}(\sigma^*)^2} \cdot \log(N + d).$$

Therefore, it holds that

$$\mathbb{E}\big[(Q_{\omega_{k+1}} - Q_{\bar{\omega}_{k+1}})^2\big] \le \frac{16(r_{\max} + R)^2}{N(\sigma^*)^2} \cdot \log^2(N + d). \tag{H.5}$$

Meanwhile, by Assumption 4.2 and the definition of $\bar{\omega}_{k+1}$, we have

$$\widetilde{Q} = Q_{\bar{\omega}_{k+1}}. \tag{H.6}$$

Combining (H.5) and (H.6), we have

$$\mathbb{E}\big[(Q_{\omega_{k+1}} - \widetilde{Q})^2\big] \le \frac{16(r_{\max} + R)^2}{N(\sigma^*)^4} \cdot \log^2(N + d),$$

which concludes the proof of Lemma D.1.

## H.2   Proof of Lemma E.1

Following from the definitions of $\mathbb{P}^{\pi}$ and $\mathbb{P}$ in (2.3), we have

$$A_{1,k}(s,a) = \big[\gamma(\mathbb{P}^{\pi^*} - \mathbb{P}^{\pi_{\theta_{k+1}}})Q_{\omega_k}\big](s,a) = \big[\gamma\mathbb{P}\langle Q_{\omega_k}, \pi^* - \pi_{\theta_{k+1}}\rangle\big](s,a). \tag{H.7}$$

By invoking Lemma F.1 and combining (H.7), it holds for any $(s,a) \in \mathcal{S} \times \mathcal{A}$ that

$$A_{1,k}(s,a) = \big[\gamma(\mathbb{P}^{\pi^*} - \mathbb{P}^{\pi_{\theta_{k+1}}})Q_{\omega_k}\big](s,a) \le \big[\gamma\beta \cdot \mathbb{P}(\vartheta_k + \epsilon_{k+1}^{a})\big](s,a),$$

where $\vartheta_k$ and $\epsilon_{k+1}^{a}$ are defined in (E.4) and (E.5) of Lemma E.1, respectively. We conclude the proof of Lemma E.1.

## H.3 PROOF OF LEMMA E.2

By the definition that $Q^*$ is the action-value function of an optimal policy $\pi^*$, we know that $Q^*(s,a) \geq Q^\pi(s,a)$ for any policy $\pi$ and state-action pair $(s,a) \in \mathcal{S} \times \mathcal{A}$. Therefore, for any $(s,a) \in \mathcal{S} \times \mathcal{A}$, we have

$$A_{2,k}(s,a) = \left[\gamma \mathbb{P}^{\pi^*}(Q^{\pi_{\theta_{k+1}}} - Q_{\omega_k})\right](s,a) \leq \left[\gamma \mathbb{P}^{\pi^*}(Q^* - Q_{\omega_k})\right](s,a). \tag{H.8}$$

In the sequel, we upper bound $Q^*(s,a) - Q_{\omega_k}(s,a)$ for any $(s,a) \in \mathcal{S} \times \mathcal{A}$. We define

$$\widetilde{Q}_{k+1} = (1-\gamma)\cdot r + \gamma \cdot \mathbb{P}^{\pi_{\theta_{k+1}}}Q_{\omega_k}.$$

By its definition, we know that $\widetilde{Q}_{k+1} = \mathbb{T}^{\pi_{\theta_{k+1}}}Q_{\omega_k}$. It holds for any $(s,a) \in \mathcal{S} \times \mathcal{A}$ that

$$
\begin{aligned}
Q^*&(s,a) - Q_{\omega_{k+1}}(s,a) \\
&= Q^*(s,a) - \widetilde{Q}_{k+1}(s,a) + \widetilde{Q}_{k+1}(s,a) - Q_{\omega_{k+1}}(s,a) \\
&= \left[\left((1-\gamma)\cdot r + \gamma\cdot\mathbb{P}^{\pi^*}Q^*\right) - \left((1-\gamma)\cdot r + \gamma\cdot\mathbb{P}^{\pi_{\theta_{k+1}}}Q_{\omega_k}\right)\right](s,a) + \epsilon^{\mathrm{c}}_{k+1}(s,a) \\
&= \gamma\cdot\left[\mathbb{P}^{\pi^*}Q^* - \mathbb{P}^{\pi_{\theta_{k+1}}}Q_{\omega_k}\right](s,a) + \epsilon^{\mathrm{c}}_{k+1}(s,a) \\
&= \gamma\cdot\left[\mathbb{P}^{\pi^*}Q^* - \mathbb{P}^{\pi^*}Q_{\omega_k}\right](s,a) + \gamma\cdot\left[\mathbb{P}^{\pi^*}Q_{\omega_k} - \mathbb{P}^{\pi_{\theta_{k+1}}}Q_{\omega_k}\right](s,a) + \epsilon^{\mathrm{c}}_{k+1}(s,a) \\
&= \gamma\cdot\left[\mathbb{P}^{\pi^*}(Q^* - Q_{\omega_k})\right](s,a) + A_{1,k}(s,a) + \epsilon^{\mathrm{c}}_{k+1}(s,a) \\
&\leq \gamma\cdot\left[\mathbb{P}^{\pi^*}(Q^* - Q_{\omega_k})\right](s,a) + \gamma\beta\cdot\left[\mathbb{P}(\vartheta_k + \epsilon^{\mathrm{a}}_{k+1})\right](s,a) + \epsilon^{\mathrm{c}}_{k+1}(s,a), \tag{H.9}
\end{aligned}
$$

where $\epsilon^{\mathrm{c}}_{k+1}$ and $A_{1,k}$ are defined in (E.6) and (E.3), respectively. Here, we use Lemma E.1 to upper bound $A_{1,k}$ in the last line. We remark that (H.9) upper bounds $Q^* - Q_{\omega_{k+1}}$ using $Q^* - Q_{\omega_k}$. By recursively applying a similar argument as in (H.9), we have

$$
\begin{aligned}
Q^*&(s,a) - Q_{\omega_k}(s,a) \\
&\leq \left[(\gamma\mathbb{P}^{\pi^*})^k(Q^* - Q_{\omega_0})\right](s,a) + \gamma\beta\cdot\sum_{i=0}^{k-1}\left[(\gamma\mathbb{P}^{\pi^*})^{k-i-1}\mathbb{P}(\vartheta_i + \epsilon^{\mathrm{a}}_{i+1})\right](s,a) \tag{H.10} \\
&\quad + \sum_{i=0}^{k-1}\left[(\gamma\mathbb{P}^{\pi^*})^{k-i-1}\epsilon^{\mathrm{c}}_{i+1}\right](s,a).
\end{aligned}
$$

Combining (H.8) and (H.10), it holds for any $(s,a) \in \mathcal{S} \times \mathcal{A}$ that

$$
\begin{aligned}
A_{2,k}(s,a) &\leq \left[\gamma\mathbb{P}^{\pi^*}(Q^* - Q_{\omega_k})\right](s,a) \\
&\leq \left[(\gamma\mathbb{P}^{\pi^*})^{k+1}(Q^* - Q_{\omega_0})\right](s,a) + \gamma\beta\cdot\sum_{i=0}^{k-1}\left[(\gamma\mathbb{P}^{\pi^*})^{k-i}\mathbb{P}(\vartheta_i + \epsilon^{\mathrm{a}}_{i+1})\right](s,a) \\
&\quad + \sum_{i=0}^{k-1}\left[(\gamma\mathbb{P}^{\pi^*})^{k-i}\epsilon^{\mathrm{c}}_{i+1}\right](s,a),
\end{aligned}
$$

where $\vartheta_i$, $\epsilon^{\mathrm{a}}_{i+1}$, and $\epsilon^{\mathrm{c}}_{i+1}$ are defined in (E.4) of Lemma E.1, (E.5) of Lemma E.1, and (E.6) of Lemma E.2, respectively. We conclude the proof of Lemma E.2.

## H.4   Proof of Lemma E.3

Note that for any $(s, a) \in \mathcal{S} \times \mathcal{A}$, we have

$$
\begin{aligned}
A_{3,k}(s, a) &= [\mathbb{T}^{\pi_{\theta_{k+1}}} Q_{\omega_k} - Q^{\pi_{\theta_{k+1}}}](s, a) \\
&= \Big[ \big( (1 - \gamma) \cdot r + \gamma \mathbb{P}^{\pi_{\theta_{k+1}}} Q_{\omega_k} \big) - Q^{\pi_{\theta_{k+1}}} \Big](s, a) \\
&= \Big[ \big( (1 - \gamma) \cdot r + \gamma \mathbb{P}^{\pi_{\theta_{k+1}}} Q_{\omega_k} \big) - \sum_{t=0}^{\infty} (1 - \gamma)(\gamma \mathbb{P}^{\pi_{\theta_{k+1}}})^t r \Big](s, a) \\
&= \Big[ \sum_{t=1}^{\infty} \big( (\gamma \mathbb{P}^{\pi_{\theta_{k+1}}})^t Q_{\omega_k} - (\gamma \mathbb{P}^{\pi_{\theta_{k+1}}})^{t+1} Q_{\omega_k} \big) - \sum_{t=1}^{\infty} (1 - \gamma)(\gamma \mathbb{P}^{\pi_{\theta_{k+1}}})^t r \Big](s, a) \\
&= \sum_{t=1}^{\infty} \Big[ (\gamma \mathbb{P}^{\pi_{\theta_{k+1}}})^t \big( Q_{\omega_k} - \gamma \mathbb{P}^{\pi_{\theta_{k+1}}} Q_{\omega_k} - (1 - \gamma) \cdot r \big) \Big](s, a) \\
&= \sum_{t=1}^{\infty} \Big[ (\gamma \mathbb{P}^{\pi_{\theta_{k+1}}})^t \big( Q_{\omega_k} - \mathbb{T}^{\pi_{\theta_{k+1}}} Q_{\omega_k} \big) \Big](s, a) \\
&= \sum_{t=1}^{\infty} \big[ (\gamma \mathbb{P}^{\pi_{\theta_{k+1}}})^t e_{k+1} \big](s, a) = \big[ \gamma \mathbb{P}^{\pi_{\theta_{k+1}}} (I - \gamma \mathbb{P}^{\pi_{\theta_{k+1}}})^{-1} e_{k+1} \big](s, a),
\end{aligned}
$$

where the term $e_{k+1}$ in the last line is defined in (E.7). We conclude the proof of Lemma E.3.

## H.5   Proof of Lemma E.4

We invoke Lemma F.1 in §F, which gives

$$
\begin{aligned}
\beta^{-1} \cdot &\langle Q_{\omega_k}(s, \cdot), \pi_{\theta_k}(\cdot \,|\, s) - \pi_{\theta_{k+1}}(\cdot \,|\, s) \rangle \\
&\le \big\langle \log(\pi_{\theta_{k+1}}(\cdot \,|\, s)/\pi_{\theta_k}(\cdot \,|\, s)) - \beta^{-1} \cdot Q_{\omega_k}(s, \cdot), \pi_{\theta_k}(\cdot \,|\, s) - \pi_{\theta_{k+1}}(\cdot \,|\, s) \big\rangle \\
&\quad - \mathrm{KL}(\pi_{\theta_k}(\cdot \,|\, s) \,\|\, \pi_{\theta_{k+1}}(\cdot \,|\, s)) \\
&\le \big\langle \log(\pi_{\theta_{k+1}}(\cdot \,|\, s)/\pi_{\theta_k}(\cdot \,|\, s)) - \beta^{-1} \cdot Q_{\omega_k}(s, \cdot), \pi_{\theta_k}(\cdot \,|\, s) - \pi_{\theta_{k+1}}(\cdot \,|\, s) \big\rangle = \epsilon_{k+1}^{\mathrm{b}}(s).
\end{aligned}
\tag{H.11}
$$

Combining (H.11) and the definition of $\mathbb{P}^\pi$ in (2.3), we have

$$
[\mathbb{P}^{\pi_{\theta_k}} Q_{\omega_k} - \mathbb{P}^{\pi_{\theta_{k+1}}} Q_{\omega_k}](s, a) \le \beta [\mathbb{P} \epsilon_{k+1}^{\mathrm{b}}](s).
\tag{H.12}
$$

By the definition of $e_{k+1}$ in (E.7), we have

$$
\begin{aligned}
e_{k+1}(s, a) &= \big[ Q_{\omega_k} - \gamma \cdot \mathbb{P}^{\pi_{\theta_{k+1}}} Q_{\omega_k} - (1 - \gamma) \cdot r \big](s, a) \\
&\le \big[ Q_{\omega_k} - \gamma \cdot \mathbb{P}^{\pi_{\theta_k}} Q_{\omega_k} - (1 - \gamma) \cdot r \big](s, a) + \beta\gamma \cdot [\mathbb{P} \epsilon_{k+1}^{\mathrm{b}}](s, a) \\
&= \big[ \widetilde{Q}_k - \gamma \cdot \mathbb{P}^{\pi_{\theta_k}} \widetilde{Q}_k - (1 - \gamma) \cdot r \big](s, a) + \big[ \beta\gamma \mathbb{P} \epsilon_{k+1}^{\mathrm{b}} - (I - \gamma \mathbb{P}^{\pi_{\theta_k}}) \epsilon_k^{\mathrm{c}} \big](s, a),
\end{aligned}
\tag{H.13}
$$

where we use (H.12) in the first inequality, and

$$
\widetilde{Q}_k = (1 - \gamma) \cdot r + \gamma \cdot \mathbb{P}^{\pi_{\theta_k}} Q_{\omega_{k-1}}.
\tag{H.14}
$$

For the first term on the RHS of (H.13), by (H.14), it holds that

$$
\begin{aligned}
\widetilde{Q}_k - &\gamma \cdot \mathbb{P}^{\pi_{\theta_k}} \widetilde{Q}_k - (1 - \gamma) \cdot r \\
&= (1 - \gamma) \cdot r + \gamma \cdot \mathbb{P}^{\pi_{\theta_k}} Q_{\omega_{k-1}} - \gamma(1 - \gamma) \cdot \mathbb{P}^{\pi_{\theta_k}} r - (\gamma \mathbb{P}^{\pi_{\theta_k}})^2 Q_{\omega_{k-1}} - (1 - \gamma) \cdot r \\
&= \gamma \cdot \mathbb{P}^{\pi_{\theta_k}} \big( Q_{\omega_{k-1}} - \gamma \mathbb{P}^{\pi_{\theta_k}} Q_{\omega_{k-1}} - (1 - \gamma) r \big) = \gamma \cdot \mathbb{P}^{\pi_{\theta_k}} e_k.
\end{aligned}
\tag{H.15}
$$

Combining (H.13) and (H.15), we have for any $(s, a) \in \mathcal{S} \times \mathcal{A}$ that

$$
e_{k+1}(s, a) \le [\gamma \mathbb{P}^{\pi_{\theta_k}} e_k](s, a) + \big[ \beta\gamma \mathbb{P} \epsilon_{k+1}^{\mathrm{b}} - (I - \gamma \mathbb{P}^{\pi_{\theta_k}}) \epsilon_k^{\mathrm{c}} \big](s, a).
\tag{H.16}
$$

By telescoping (H.16), it holds that

$$
e_{k+1}(s, a) \le \Big[ \Big( \prod_{s=1}^{k} \gamma \mathbb{P}^{\pi_{\theta_s}} \Big) e_1 + \sum_{i=1}^{k} \gamma^{k-i} \Big( \prod_{s=i+1}^{k} \mathbb{P}^{\pi_{\theta_s}} \Big) \big( \beta\gamma \mathbb{P} \epsilon_{i+1}^{\mathrm{b}} - (I - \gamma \mathbb{P}^{\pi_{\theta_i}}) \epsilon_i^{\mathrm{c}} \big) \Big](s, a).
$$

This finishes the proof of the lemma.

### H.6  PROOF OF LEMMA E.5

Note that $\|\omega_0\|_2 \le R$ and $|r(s,a)| \le r_{\max}$ for any $(s,a) \in \mathcal{S} \times \mathcal{A}$, which implies that $|Q_{\omega_0}(s,a)| \le R$ and $|Q^*(s,a)| \le r_{\max}$ by their definitions. Thus, for $M_1$, we have

$$
|M_1| \le \mathbb{E}_\rho\Big[(I - \gamma\mathbb{P}^{\pi^*})^{-1} \sum_{k=0}^K (\gamma\mathbb{P}^{\pi^*})^{k+1}|Q^* - Q_{\omega_0}|\Big]
$$

$$
\le 4(1-\gamma)^{-1} \sum_{k=0}^K \gamma^{k+1} \cdot (r_{\max} + R) \le 4(1-\gamma)^{-2} \cdot (r_{\max} + R). \tag{H.17}
$$

For $M_2$, by the definition of $e_1$ in (E.7), $|\omega_k| \le R$, $|\phi(s,a)| \le 1$, and $|r(s,a)| \le r_{\max}$, we have

$$
|e_1(s,a)| = \big|[Q_{\omega_k} - \mathbb{T}^{\pi_{\theta_{k+1}}} Q_{\omega_k}](s,a)\big|
$$
$$
= \big|\omega_k^\top \phi(s,a) - \gamma \cdot \omega_k^\top [\mathbb{P}^{\pi_{\theta_{k+1}}}\phi](s,a) - (1-\gamma) \cdot r(s,a)\big|
$$
$$
\le 2R + r_{\max} \tag{H.18}
$$

for any $(s,a) \in \mathcal{S} \times \mathcal{A}$. Therefore, we have

$$
|M_2| \le (1-\gamma)^{-3} \cdot (2R + r_{\max}). \tag{H.19}
$$

Meanwhile, by the initialization $\tau_0 = \infty$ in Algorithm 1, the initial policy $\pi_{\theta_0}(\cdot \,|\, s)$ is a uniform distribution over $\mathcal{A}$. Therefore, it holds for any $s \in \mathcal{S}$ that

$$
\mathrm{KL}\big(\pi^*(\cdot \,|\, s) \,\|\, \pi_{\theta_0}(\cdot \,|\, s)\big) = \int_\mathcal{A} \pi^*(a \,|\, s) \log \frac{\pi^*(a \,|\, s)}{\pi_{\theta_0}(a \,|\, s)} \mathrm{d}a
$$
$$
= \int_\mathcal{A} \pi^*(a \,|\, s) \log \pi^*(a \,|\, s)\mathrm{d}a - \int_\mathcal{A} \pi^*(a \,|\, s) \log \pi_{\theta_0}(a \,|\, s)\mathrm{d}a
$$
$$
\le - \int_\mathcal{A} \pi^*(a \,|\, s) \log \pi_{\theta_0}(a \,|\, s)\mathrm{d}a
$$
$$
= \int_\mathcal{A} \pi^*(a \,|\, s) \log |\mathcal{A}|\mathrm{d}a = \log |\mathcal{A}|. \tag{H.20}
$$

Therefore, by (H.20), we have

$$
M_3 \le (1-\gamma)^{-2} \cdot \log |\mathcal{A}| \cdot K^{1/2}, \tag{H.21}
$$

where we use $\beta = K^{1/2}$. We see that (H.17), (H.19), and (H.21) upper bound $M_1$, $M_2$, and $M_3$, respectively. We conclude the proof of Lemma E.5.

### H.7  PROOF OF LEMMA E.6

For $M_4$, by changing the index of summation, we have

$$
|M_4| = \Big|\mathbb{E}_\rho\Big[\sum_{k=0}^K \sum_{i=0}^k \sum_{j=0}^\infty (\gamma\mathbb{P}^{\pi^*})^{k-i+j}\epsilon_{i+1}^{\mathrm{c}}\Big]\Big|
$$
$$
= \Big|\mathbb{E}_\rho\Big[\sum_{k=0}^K \sum_{i=0}^k \sum_{t=k-i}^\infty (\gamma\mathbb{P}^{\pi^*})^t\epsilon_{i+1}^{\mathrm{c}}\Big]\Big|
$$
$$
\le \sum_{k=0}^K \sum_{i=0}^k \sum_{t=k-i}^\infty \big|\mathbb{E}_\rho\big[(\gamma\mathbb{P}^{\pi^*})^t\epsilon_{i+1}^{\mathrm{c}}\big]\big|, \tag{H.22}
$$

where we expand $(I - \gamma\mathbb{P}^{\pi^*})^{-1}$ into an infinite sum in the first equality. Further, by changing the measure of the expectation from $\rho$ to $\rho^*$ on the RHS of (H.22), we have

$$
\sum_{k=0}^K \sum_{i=0}^k \sum_{t=k-i}^\infty \big|\mathbb{E}_\rho\big[(\gamma\mathbb{P}^{\pi^*})^t\epsilon_{i+1}^{\mathrm{c}}\big]\big| \le \sum_{k=0}^K \sum_{i=0}^k \sum_{t=k-i}^\infty \gamma^t c(t) \cdot \mathbb{E}_{\rho^*}\big[|\epsilon_{i+1}^{\mathrm{c}}|\big], \tag{H.23}
$$

where $c(t)$ is defined in Assumption 4.1. Further, by changing the index of summation on the RHS of (H.23), combining (H.22), we have

$$
\begin{aligned}
|M_4| &\leq \sum_{k=0}^{K} \sum_{t=0}^{\infty} \sum_{i=\max\{0,k-t\}}^{k} \gamma^t c(t) \cdot \varepsilon_Q \\
&\leq \sum_{k=0}^{K} \sum_{t=0}^{\infty} 2t\gamma^t c(t) \cdot \varepsilon_Q \\
&\leq \gamma \sum_{k=0}^{K} 2C_{\rho,\rho^*} \cdot \varepsilon_Q \leq 3KC_{\rho,\rho^*} \cdot \varepsilon_Q,
\end{aligned}
\tag{H.24}
$$

where $\varepsilon_Q = \max_i \mathbb{E}_{\rho^*}[|\epsilon_{i+1}^c|]$, and $C_{\rho,\rho^*}$ is defined in Assumption 4.1.

Now, for $M_5$, by a similar argument as in the derivation of (H.24), we have

$$
\begin{aligned}
M_5 &\leq \sum_{i=0}^{\infty} \sum_{k=0}^{K} \sum_{j=0}^{\infty} \sum_{\ell=1}^{k} \gamma^{i+j+k-\ell+1} c(i+j+k-\ell+1) \cdot \varepsilon_Q \\
&= \sum_{i=0}^{\infty} \sum_{k=0}^{K} \sum_{j=0}^{\infty} \sum_{t=i+j+1}^{i+j+k} \gamma^t c(t) \cdot \varepsilon_Q \leq \sum_{k=0}^{K} \sum_{t=1}^{\infty} t^2 \gamma^t c(t) \cdot \varepsilon_Q \leq KC_{\rho,\rho^*} \cdot \varepsilon_Q.
\end{aligned}
\tag{H.25}
$$

We see that (H.24) and (H.25) upper bound $M_4$ and $M_5$, respectively. We conclude the proof of Lemma E.6.

## H.8 PROOF OF LEMMA E.7

**Part 1.** We first show that the first inequality holds. Note that

$$
\pi_{\theta_k}(a \mid s) = \exp(\tau_k^{-1} f_{\theta_k}(s,a))/Z_{\theta_k}(s), \qquad \pi_{\theta_{k+1}}(a \mid s) = \exp(\tau_{k+1}^{-1} f_{\theta_{k+1}}(s,a))/Z_{\theta_{k+1}}(s),
$$

Here $Z_{\theta_k}(s), Z_{\theta_{k+1}}(s) \in \mathbb{R}$ are normalization factors, which are defined as

$$
Z_{\theta_k}(s) = \sum_{a' \in \mathcal{A}} \exp(\tau_k^{-1} f_{\theta_k}(s,a')), \qquad Z_{\theta_{k+1}}(s) = \sum_{a' \in \mathcal{A}} \exp(\tau_{k+1}^{-1} f_{\theta_{k+1}}(s,a')).
$$

Thus, we have

$$
\begin{aligned}
&\langle \log(\pi_{\theta_{k+1}}(\cdot \mid s)/\pi_{\theta_k}(\cdot \mid s)) - \beta^{-1} Q_{\omega_k}(s,\cdot), \pi^*(\cdot \mid s) - \pi_{\theta_{k+1}}(\cdot \mid s) \rangle \\
&\quad = \langle \tau_{k+1}^{-1} f_{\theta_{k+1}}(s,\cdot) - (\beta^{-1} Q_{\omega_k}(s,\cdot) + \tau_k^{-1} f_{\theta_k}(s,\cdot)), \pi^*(\cdot \mid s) - \pi_{\theta_k}(\cdot \mid s) \rangle,
\end{aligned}
\tag{H.26}
$$

where we use the fact that

$$
\begin{aligned}
&\langle \log Z_{\theta_{k+1}}(s) - \log Z_{\theta_k}(s), \pi^*(\cdot \mid s) - \pi_{\theta_{k+1}}(\cdot \mid s) \rangle \\
&\quad = (\log Z_{\theta_{k+1}}(s) - \log Z_{\theta_k}(s)) \cdot \sum_{a' \in \mathcal{A}} (\pi^*(a' \mid s) - \pi_{\theta_{k+1}}(a' \mid s)) = 0.
\end{aligned}
$$

Thus, it remains to upper bound the right-hand side of (H.26). We have

$$
\begin{aligned}
&\langle \tau_{k+1}^{-1} f_{\theta_{k+1}}(s,\cdot) - (\beta_k^{-1} Q_{\omega_k}(s,\cdot) + \tau_k^{-1} f_{\theta_k}(s,\cdot)), \pi^*(\cdot \mid s) - \pi_{\theta_{k+1}}(\cdot \mid s) \rangle \\
&\quad = \Big\langle \tau_{k+1}^{-1} f_{\theta_{k+1}}(s,\cdot) - (\beta_k^{-1} Q_{\omega_k}(s,\cdot) + \tau_k^{-1} f_{\theta_k}(s,\cdot)), \pi_{\theta_k}(\cdot \mid s) \cdot \Big( \frac{\pi^*(\cdot \mid s)}{\pi_{\theta_k}(\cdot \mid s)} - \frac{\pi_{\theta_{k+1}}(\cdot \mid s)}{\pi_{\theta_k}(\cdot \mid s)} \Big) \Big\rangle.
\end{aligned}
\tag{H.27}
$$

Taking expectation with respect to $s \sim \nu^*$ on the both sides of (H.27) and using the Cauchy-Schwarz inequality, we obtain

$$
\begin{aligned}
\mathbb{E}_{\nu^*} & \big[ \big| \big\langle \tau_{k+1}^{-1} f_{\theta_{k+1}}(s, \cdot) - (\beta_k^{-1} Q_{\omega_k}(s, \cdot) + \tau_k^{-1} f_{\theta_k}(s, \cdot)), \pi^*(\cdot \mid s) - \pi_{\theta_{k+1}}(\cdot \mid s) \big\rangle \big| \big] \\
& = \int_{\mathcal{S}} \Big| \Big\langle \tau_{k+1}^{-1} f_{\theta_{k+1}}(s, \cdot) - (\beta_k^{-1} Q_{\omega_k}(s, \cdot) + \tau_k^{-1} f_{\theta_k}(s, \cdot)), \\
& \qquad\qquad \pi_{\theta_k}(\cdot \mid s) \cdot \nu_k(s) \cdot \Big( \frac{\pi^*(\cdot \mid s)}{\pi_{\theta_k}(\cdot \mid s)} - \frac{\pi_{\theta_{k+1}}(\cdot \mid s)}{\pi_{\theta_k}(\cdot \mid s)} \Big) \Big\rangle \Big| \cdot \Big| \frac{\nu^*(s)}{\nu_k(s)} \Big| \mathrm{d}s \\
& = \int_{\mathcal{S} \times \mathcal{A}} \big| \tau_{k+1}^{-1} f_{\theta_{k+1}}(s, a) - (\beta_k^{-1} Q_{\omega_k}(s, a) + \tau_k^{-1} f_{\theta_k}(s, a)) \big| \\
& \qquad\qquad \cdot \Big| \frac{\rho^*(a \mid s)}{\rho_k(a \mid s)} - \frac{\pi_{\theta_{k+1}}(a \mid s) \cdot \nu^*(s)}{\rho_k(a \mid s)} \Big| \mathrm{d}\rho_k(s, a) \\
& \le \mathbb{E}_{\rho_k} \big[ \big( \tau_{k+1}^{-1} f_{\theta_{k+1}}(s, a) - (\beta_k^{-1} Q_{\omega_k}(s, a) + \tau_k^{-1} f_{\theta_k}(s, a)) \big)^2 \big]^{1/2} \cdot \mathbb{E}_{\rho_k} \Big[ \Big| \frac{\mathrm{d}\rho^*}{\mathrm{d}\rho_k} - \frac{\mathrm{d}(\pi_{\theta_{k+1}} \nu^*)}{\mathrm{d}\rho_k} \Big|^2 \Big]^{1/2} \\
& \le \sqrt{2} \tau_{k+1}^{-1} \cdot \varepsilon_{k+1, f} \cdot (\phi_k^* + \psi_k^*),
\end{aligned}
$$

where in the last inequality we use the error bound in (E.20) and the definition of $\phi_k^*$ and $\psi_k^*$ in Assumption C.1. This finishes the proof of the first inequality.

**Part 2.** The proof of the second inequality follows from a similar argument as above. We have

$$
\begin{aligned}
\langle \log(\pi_{\theta_{k+1}}(\cdot \mid s) / \pi_{\theta_k}(\cdot \mid s)) & - \beta^{-1} Q_{\omega_k}(s, \cdot), \pi_{\theta_k}(\cdot \mid s) - \pi_{\theta_{k+1}}(\cdot \mid s) \rangle \\
& = \langle \tau_{k+1}^{-1} f_{\theta_{k+1}}(s, \cdot) - (\beta^{-1} Q_{\omega_k}(s, \cdot) + \tau_k^{-1} f_{\theta_k}(s, \cdot)), \pi_{\theta_k}(\cdot \mid s) - \pi_{\theta_{k+1}}(\cdot \mid s) \rangle, \quad \text{(H.28)}
\end{aligned}
$$

where we use the fact that

$$
\begin{aligned}
\langle \log Z_{\theta_{k+1}}(s) - \log Z_{\theta_k}(s), & \pi_{\theta_k}(\cdot \mid s) - \pi_{\theta_{k+1}}(\cdot \mid s) \rangle \\
& = (\log Z_{\theta_{k+1}}(s) - \log Z_{\theta_k}(s)) \cdot \sum_{a' \in \mathcal{A}} (\pi_{\theta_k}(a' \mid s) - \pi_{\theta_{k+1}}(a' \mid s)) = 0.
\end{aligned}
$$

Thus, it remains to upper bound the right-hand side of (H.28). We have

$$
\begin{aligned}
\langle \tau_{k+1}^{-1} f_{\theta_{k+1}}(s, \cdot) & - (\beta_k^{-1} Q_{\omega_k}(s, \cdot) + \tau_k^{-1} f_{\theta_k}(s, \cdot)), \pi_{\theta_k}(\cdot \mid s) - \pi_{\theta_{k+1}}(\cdot \mid s) \rangle \qquad\qquad \text{(H.29)} \\
& = \Big\langle \tau_{k+1}^{-1} f_{\theta_{k+1}}(s, \cdot) - (\beta_k^{-1} Q_{\omega_k}(s, \cdot) + \tau_k^{-1} f_{\theta_k}(s, \cdot)), \pi_{\theta_k}(\cdot \mid s) \cdot \Big( 1 - \frac{\pi_{\theta_{k+1}}(\cdot \mid s)}{\pi_{\theta_k}(\cdot \mid s)} \Big) \Big\rangle.
\end{aligned}
$$

Taking expectation with respect to $s \sim \nu^*$ on the both sides of (H.29) and using the Cauchy-Schwarz inequality, we obtain

$$
\begin{aligned}
\mathbb{E}_{\nu^*} & \big[ \big| \big\langle \tau_{k+1}^{-1} f_{\theta_{k+1}}(s, \cdot) - (\beta_k^{-1} Q_{\omega_k}(s, \cdot) + \tau_k^{-1} f_{\theta_k}(s, \cdot)), \pi_{\theta_k}(\cdot \mid s) - \pi_{\theta_{k+1}}(\cdot \mid s) \big\rangle \big| \big] \\
& = \int_{\mathcal{S}} \Big| \Big\langle \tau_{k+1}^{-1} f_{\theta_{k+1}}(s, \cdot) - (\beta_k^{-1} Q_{\omega_k}(s, \cdot) + \tau_k^{-1} f_{\theta_k}(s, \cdot)), \pi_{\theta_k}(\cdot \mid s) \cdot \nu_k(s) \cdot \Big( 1 - \frac{\pi_{\theta_{k+1}}(\cdot \mid s)}{\pi_{\theta_k}(\cdot \mid s)} \Big) \Big\rangle \Big| \\
& \qquad\qquad \cdot \Big| \frac{\nu^*(s)}{\nu_k(s)} \Big| \mathrm{d}s \\
& = \int_{\mathcal{S} \times \mathcal{A}} \big| \tau_{k+1}^{-1} f_{\theta_{k+1}}(s, a) - (\beta_k^{-1} Q_{\omega_k}(s, a) + \tau_k^{-1} f_{\theta_k}(s, a)) \big| \cdot \Big| 1 - \frac{\pi_{\theta_{k+1}}(a \mid s) \cdot \nu^*(s)}{\rho_k(a \mid s)} \Big| \mathrm{d}\rho_k(s, a) \\
& \le \mathbb{E}_{\rho_k} \big[ \big( \tau_{k+1}^{-1} f_{\theta_{k+1}}(s, a) - (\beta_k^{-1} Q_{\omega_k}(s, a) + \tau_k^{-1} f_{\theta_k}(s, a)) \big)^2 \big]^{1/2} \cdot \mathbb{E}_{\rho_k} \Big[ \Big| 1 - \frac{\mathrm{d}(\pi_{\theta_{k+1}} \nu^*)}{\mathrm{d}\rho_k} \Big|^2 \Big]^{1/2} \\
& \le \sqrt{2} \tau_{k+1}^{-1} \cdot \varepsilon_{k+1, f} \cdot (1 + \psi_k^*),
\end{aligned}
$$

where in the last inequality we use the error bound in (E.20) and the definition of $\psi_k^*$ in Assumption C.1. This finishes the proof of the second inequality.

