# OpenReview forum: "Single-Timescale Actor-Critic Provably Finds Globally Optimal Policy"
_ICLR.cc/2021/Conference — ICLR 2021 Poster_

### Official Review · AnonReviewer4 · 2020-10-26
**First convergence result for single-timescale actor-critic**

**Rating:** 8
**Confidence:** 1

**Review:**

Summary:

This paper proves that the proposed single-timescale actor-critic algorithm with KL regularization converges to a globally optimal policy when appropriate function approximation is used. The authors define a specific form of actor/critic updates, which are PPO update for the actor and the single application of Bellman evaluation operator for the critic. Given such an algorithm, the authors provide an upper bound of the regret for linear and deep neural network function approximation (although there are some strong assumptions, e.g. no approximation error).

Reasons for score:

Overall, I vote for strong acceptance. While I am not capable of fully checking the proofs, I understand the importance of the result. Based on the mathematical rigorousness of the main text and the proof sketch, I assume that the paper is technically correct. Given that the paper is technically correct, it is the first convergence result for a single-timescale actor-critic algorithm as far as I know (and the paper claims so). This work will be very significant as it theoretically grounds the single-timescale actor-critic algorithms, where the state-of-the-art deep RL algorithms are mostly single-timescale actor-critic algorithms.

PROS:
* The paper (at least the main text and the proof sketch) is well written.
* First result on convergence rate and global optimality of single-timescale actor-critic, which is very significant.

CONS:
* No further discussions based on the results for better design of an actor-critic algorithm. It would be helpful if there are more discussions about the algorithm choice, i.e. being off-policy, broken assumption, etc.

Questions:
* I still do not understand how this result can be applied to off-policy results, where we have $\mathbb{P}^{\pi_{\theta_{k+1}}}$ in e.g. equation 3.4 and 3.7. How can the samples be reused if $(s,a)$ are not from $\rho_{k+1}$ but from previous $\rho$s?

Typo:
* Specifically, after K + 1 actor updates, we are interest -> we are interested? : before eq (4.1).

---

> ### Author Response · Authors · 2020-11-18
> **Review Response**
>
> We appreciate the valuable comments from the reviewer. We address the concerns as follows.
> 1. (Off-policy setting.)  In the off-policy setting, all data are drawn according to a behavior policy $\pi_{\text{bhv}}$. We denote the corresponding state-action distribution induced by $\pi_{\text{bhv}}$ as $\rho_{\text{bhv}}$.  Note that $\tilde \omega_{k+1}$ in (3.4) can be re-written as
> \begin{align*}
> \tilde \omega_{k+1} & = \big(\mathbb E_{\rho_{k+1}} [ \phi \phi^\top ] \big)^{-1} \mathbb E_{\rho_{k+1}} \bigl[ \bigl((1-\gamma)r + \gamma \mathbb P^{\pi_{\theta_{k+1}}} Q_{\omega_k}   \bigr) \phi \bigr] \\\\
> & = \big(\mathbb E_{\rho_{k+1}} [ \phi \phi^\top ] \big)^{-1} \cdot \Bigl(  \mathbb E_{\rho_{k+1}} \bigl[ (1-\gamma)r  \phi \bigr]  +   \mathbb E_{\rho_{k+1}, \pi_{\theta_{k+1}}} \bigl[   \gamma Q_{\omega_k}(s',a')  \phi(s,a)  \bigr]     \Bigr) \\\\
> & = \bigg(\mathbb E_{\rho_{\text{bhv}}} \Bigl[ \phi \phi^\top \cdot \frac{\mathrm d \rho_{k+1}}{\mathrm d \rho_{\text{bhv}}} \Bigr] \bigg)^{-1} \cdot \biggl(  \mathbb E_{\rho_{\text{bhv}}} \Bigl[ (1-\gamma)r  \phi \cdot \frac{\mathrm d \rho_{k+1}}{\mathrm d \rho_{\text{bhv}}} \Bigr]  +   \mathbb E_{\text{bhv}} \Bigl[   \gamma Q_{\omega_k}(s',a')  \phi(s,a)   \cdot \frac{\mathrm d (\rho_{k+1} \mathbb P^{\pi_{\theta_{k+1}}})  }{\mathrm d (\rho_{\text{bhv}} \mathbb P^{\pi_{\theta_{\text{bhv}}}})   } \Bigr]  \biggr),
> \end{align*}
> where the last $\mathbb E_{\text{bhv}}$ denotes that the tuple $(s,a,s’,a’)$ follows $\rho_{\text{bhv}}$ and $\pi_{\text{bhv}}$.  By this, we can use off-policy data draw according to the behavior policy.
> 2. (Assumptions.)  Assumption 4.1 is a standard assumption in related literature. It indeed measures the stochastic stability properties of the MDP, and the class of MDPs with such properties is quite large. See (Szepesvari and Munos, 2005; Munos and Szepesvari, 2008; Antos et al., 2008a;b; Scherrer, 2013; Scherrer et al., 2015; Farahmand et al., 2016; Yang et al., 2019b; Geist et al., 2019; Chen and Jiang, 2019) for more discussion and examples.
> Assumption 4.2 imposes a structural assumption of the MDP under the linear setting. Specifically speaking, it assumes that the Bellman operator of each policy maps a linear value function to a linear function. Therefore, the value function associated with each policy (which is the fixed point of the corresponding Bellman operator) lies in the linear function class. We consider an energy-based policy in our work. Since the value functions are linear under the linear setting, the energy-based policy class approximately covers the optimal policy as the temperature parameter goes to zero. In summary, our Assumption 4.2 ensures that the energy-based policy class approximately captures the optimal policy and thus there is no approximation error. When this assumption does not hold, we only need to add an additional linear term in $K$ to the regret upper bound in Theorem 4.4 without much change in the proof.
> Assumption 4.3 essentially states that the linear feature $\varphi(s,a)$ is well-conditioned, in the sense that it ensures the existence and uniqueness of the critic parameter $\omega$.  Similar assumptions are also imposed in related literature with linear function approximation [1,2,3,4].
> 3. (Typo.)  We thank the reviewer for carefully checking paper and pointing out the typo. We will revise accordingly.
>
>
>
> [1] Wu, Y., Zhang, W., Xu, P., & Gu, Q. (2020). A Finite Time Analysis of Two Time-Scale Actor Critic Methods. arXiv preprint arXiv:2005.01350.
>
> [2] Xu, T., Zou, S., & Liang, Y. (2019). Two time-scale off-policy TD learning: Non-asymptotic analysis over Markovian samples. In Advances in Neural Information Processing Systems (pp. 10634-10644).
>
> [3] Zou, S., Xu, T., & Liang, Y. (2019). Finite-sample analysis for SARSA with linear function approximation. In Advances in Neural Information Processing Systems (pp. 8668-8678).
>
> [4] Bhandari, J., Russo, D., & Singal, R. (2018). A finite time analysis of temporal difference learning with linear function approximation. arXiv preprint arXiv:1806.02450.

---

### Official Review · AnonReviewer2 · 2020-10-28
**Meaningful Result for AC algorithm**

**Rating:** 7
**Confidence:** 4

**Review:**

This paper studies the finite time performance of actor-critic algorithm with PPO-type update. Global convergence  is provided by exploring the convex-like property of the objective function in the distributional space. Different from previous studies that focus either on two time-scale update or nested-loop update, this paper studies the single time-scale update scheme. Thus the result in this paper is somehow meaningful.

I have the following questions for the author:

(1) The algorithm proposed and stuided in this paper can be implemented in a "single time-scale" way only at a cost of high computational cost at each iteration, making the motivation of using this algorithm in practice less clear. The AC algorithm in this paper need a near-optimal solution in both policy optimization (actor) and policy evluation (critic) at each iteration, thus requires the critic to perform LSTD instead of TD(0). However LSTD requires computation cost of O(d^2) and TD(0) only requires O(d), thus TD(0)-typer critic is more favored in practice, especially when the dimension d is very large. Since this paper is a pure theoretical paper so let's forget about the practical concern first. Can the author highlight the technique challenging of using LSTD-critic instead of TD(0)-critic compared with previous work Liu et al. (2019)?

Boyi Liu, Qi Cai, Zhuoran Yang, Zhaoran Wang, (2019) "Neural Proximal/Trust Region Policy Optimization Attains Globally Optimal Policy"

(2) The critic stuided in this paper adopts projection to enable the whole proof go through. The projection is a practical issue which has been criticized by many researchers in the RL community. Can the author provide an upper bound for the projection radius so that the algorithm can be at least implemented in practice.

(3) It seems that the neural network parameterization studied in the appendix is still a nested-loop algorithm, not a single-timescale algorithm that highlighted in the title and the story of this paper. I agree that it is meaningful to study the DNN in AC for the first time, but can the author compare the overall sample complexity in this paper with the sample complexity in Liu et al. (2019)? Does the LSTD-critic help to improve the overall sample complexity comparied with the TD(0)-type critic in Liu et at. (2019)?

(4) It seems that the global convergence guarantee is highly depend on the soft-max parameterization, which has limited practical applications. Can the author still establish a similar result in the infinite-action space case by using a similar proof in this paper?

I will keep my score if the author can answer the above questions.

=== post rebuttal ===

I am satisfied with the response and i will keep my score.

---

> ### Author Response · Authors · 2020-11-18
> **Review Response**
>
> We appreciate the valuable comments from the reviewer. We address the concerns as follows.
>
> 1. (LSTD v.s. TD(0).)  [1] employs a bilevel setting, where they need to solve the fixed point problem $\min_{\omega} ||Q_\omega - \mathcal T^{\pi_{\theta_{k}}} Q_{\omega} ||$ approximately in the critic. By stochastic semi-gradient descent, this leads to TD(0) update, which requires a **nested loop**to solve the fixed point problem.  In comparison, our method is single-timescale, and we only need to solve $\min_{\omega} ||Q_\omega - \mathcal T^{\pi_{\theta_{k}}} Q_{\omega_k} ||$ in the critic, which can be solved using only **one**LSTD update under the linear function approximation.
> In terms of the technical challenge, since [1] employs a nested-loop TD(0) update in the critic, which is equivalent to applying the Bellman operator to the value function **infinite times**, the update direction used in the actor update is aligned with the ideal update direction specified by the PPO method. Thus, they can decouple the policy optimization step and the policy evaluation step in the proof, and prove the convergence of policy optimization and policy evaluation steps separately. In comparison, since our setting employs only one LSTD update in the critic, which is equivalent to applying the Bellman operator to the value function **only once**, the update direction used in the actor update is different from the ideal update direction specified by the PPO method. Due to the difference, we cannot decouple the policy optimization and policy evaluation steps, so that we need a more delicate analysis to show the convergence of the single-timescale algorithm, which needs a “double contraction” argument as in our proof.
>
> 2. (Projection.)  Based on our analysis, we only require that the upper bound of $M_3$ dominates $M_1$ and $M_2$. See (E.13), Lemma E.5, and (E.14) for details. Thus, we require that $R = O((1-\gamma)\log |\mathcal A| K^{1/2})$ in the linear function approximation setting. As for the DNN setting, the requirement for $R_{\text{a}}$ and $R_{\text{c}}$ are specified in Theorem C.5.  We remark that our analysis (especially the DNN setting) aims to provide the first theoretical justification of using function approximation in single-timescale AC and thus adopt projection for simplicity. A more delicate analysis that removes projection is left to future work.
>
> 3. (Sample complexity in DNN setting.)  First, [1] studies the AC algorithm under a bilevel setting, where the actor is updated only after the critic solves the policy evaluation sub-problem completely (using a nested-loop TD(0) in their paper), which is equivalent to applying the Bellman evaluation operator to the previous critic for infinite times. In sharp contrast, in our work, the actor is updated using a direction given by the current critic, which is updated using a single step of Bellman update and can be far from the value function associated with the actor (which can be obtained by applying the Bellman evaluation operator to the previous critic for infinite times). Thus, our work requires a more delicate analysis than [1]. Please see point 1 for a detailed comparison with [1].
> In terms of the sample complexity, to simplify our discussion, we omit constant and logarithmic terms here. In our work, By Theorem C.5 of the DNN setting, to obtain an $\varepsilon$-globally optimal policy, it suffices to set $K\asymp \varepsilon^{-2}$ in Algorithm 2. By plugging such a $K$ into $N_{\text{a}} = \Omega(K^2)$ and $N_{\text{c}} = \Omega(K^2)$ (which is required by Theorem C.5), we have $N_{\text{a}} = O(\varepsilon^{-4})$ and $N_{\text{c}} = O(\varepsilon^{-4})$. Thus, to achieve an $\varepsilon$-globally optimal policy, the total sample complexity of Algorithm 2 is $O(\varepsilon^{-6})$.  Further with data reuse, the total sample complexity of Algorithm 2 is $O(\varepsilon^{-4})$ to achieve an $\varepsilon$-globally optimal policy.  In comparison, by Corollary 4.10 in [1], to achieve an $\varepsilon$-globally optimal policy, they need to set $K\asymp \varepsilon^{-2}$ and $T = \Omega(K^3)$, where $T$ is the number of iterations in the nested loop. This results in a total sample complexity of $O(\varepsilon^{-8})$, which is worse than our single-timescale algorithm. Meanwhile, off-policy TD(0) learning is shown to diverge even under linear function approximation [2], thus the method of data reuse cannot be applied to [1] to eliminate the total sample complexity.

---

> > ### Author Response · Authors · 2020-11-18
> > **Review Response (Cont.)**
> >
> > 4. (Infinite action space.) In the case of infinite action space, to sample from the energy-based stochastic policy $\pi(\cdot | s_t)$ in the algorithm, we can adopt SVGD [5] as in [3] or a reparameterization trick as in [4]. We note that both [3] and [4] are practical algorithms. If the Lebesgue measure of the action space is finite, our proof will also work when the action space is infinite. In this case, we only need to replace the cardinality $|\mathcal{A}|$ of the action space by the Lebesgue measure $m(A)$ of the action space in the proof.
> >
> > [1] Boyi Liu, Qi Cai, Zhuoran Yang, Zhaoran Wang, (2019) "Neural Proximal/Trust Region Policy Optimization Attains Globally Optimal Policy"
> >
> > [2] Baird, L. (1995). Residual algorithms: Reinforcement learning with function approximation. In Machine Learning Proceedings 1995 (pp. 30-37). Morgan Kaufmann.
> >
> > [3] Haarnoja, T., Tang, H., Abbeel, P., & Levine, S. (2017). Reinforcement learning with deep energy-based policies. arXiv preprint arXiv:1702.08165.
> >
> > [4] Haarnoja, T., Zhou, A., Abbeel, P., & Levine, S. (2018). Soft actor-critic: Off-policy maximum entropy deep reinforcement learning with a stochastic actor. arXiv preprint arXiv:1801.01290.
> >
> > [5] Liu, Q., & Wang, D. (2016). Stein variational gradient descent: A general purpose bayesian inference algorithm. In Advances in neural information processing systems (pp. 2378-2386).

---

### Official Review · AnonReviewer3 · 2020-10-30
**Theory for an one-timescale AC**

**Rating:** 8
**Confidence:** 3

**Review:**

This is a theoretical paper. The paper studies convergence and optimality of the actor-critic with the function approximation in a single-timescale setting. The proposed single-timescale AC applies PPO for the actor and updates the critic by applying the Bellman operator to the critic one time. Despite the actor-critic coupling, the paper establishes global convergence with sublinear rates for the proposed AC method in both linear and neural network function approximation.

Strong points:

(1) The paper is well-written, and the motivation is easy to follow. The paper also provides detailed literature on related works.

(2) The studied AC is practical in the sense that both the actor and the critic can take function approximation and their updates work in the one-timescale. The proposed scheme could complement the previous study of the AC methods in the two-timescale.

(3) The provided convergence theory could be some new insights into dealing with the coupling of the actor and the critic. The provided intuition makes sense to me although I didn’t get time checking proof details.

Weak points and comments:

(1) Although the paper provides a general setup for the one-timescale AC method, it is worth providing some generic examples to explain the theory, e.g., the energy-based policy with direct parametrization.

(2) The theory seems to be specific to the energy-based policy. Are there any other types of policies that can also be considered? If not, it would be helpful to comment on the importance of the energy-based policy in practice or from the theoretical point of view?

(3) The proposed AC methods rely on the population quantity, e.g., expectation over state-action visitation probability. How practical are they? Or how is the proposed AC related to practical one-timescale AC methods mentioned in the Introduction?

(4) In (2.7), the critic depends on the current policy at k+1. What do you mean by they are updated simultaneously in the Abstract?

(5) It would be helpful to make a table comparing the proposed AC with others in terms of sampling assumptions, policy classes, and convergence.

I believe my concerns can be addressed during reviewing. For me, the development of this paper is new, and I am more inclined to agree with the acceptance.

---

> ### Author Response · Authors · 2020-11-18
> **Review Response**
>
> We appreciate the valuable comments from the reviewer. We address the concerns as follows.
>
> 1. (Examples.)  We will add some examples in the future revision.
> 2. (Energy-based policy.)  Our analysis is based on the energy-based policy, so that we can have the closed-form actor update as shown in Proposition 3.1. Such an energy-based policy can essentially represent any policy, given that the class of the energy function $f(s,a)$ is sufficiently large (e.g., parameterized using DNN). Such a setting with energy-based policy is widely used in both theoretical and empirical works [1,2,3].
> 3. (Sample from visitation measure.)  To sample from the state-action visitation measure $\rho_{\pi}$, one direct way is to run policy $\pi$ for a sufficiently large number of timesteps so that the Markov chain reaches its stationary distribution (given that the Markov chain mixes geometrically under proper conditions, the total sample complexity will only be affected by a logarithmic order). After the Markov chain mixes, the trajectory generated using $\pi$ follows  such a stationary distribution.
> 4. (Simultaneous update.)  By saying that the actor and critic are updated simultaneously, we mean that the actor and critic are updated under the same timescale, which gives the single timescale setting in our work. That is, the actor is updated using a direction given by the current critic, which is updated using a single step of Bellman update and can be far from the value function associated with the actor (which is obtained by finding the fixed point of the Bellman operator). This is in sharp contrast to the two-timescale setting or the bilevel setting, where the critic is updated with a much larger stepsize than the actor, or the critic is solved by a separate inner loop that approximately finds the fixed point of the Bellman operator.
> 5. (Table for comparison.)  We will add a table to compare the proposed AC with other methods in the future revision.
>
> [1] Haarnoja, T., Tang, H., Abbeel, P., & Levine, S. (2017). Reinforcement learning with deep energy-based policies. arXiv preprint arXiv:1702.08165.
>
> [2] Heess, N., Silver, D., & Teh, Y. W. (2013, January). Actor-critic reinforcement learning with energy-based policies. In European Workshop on Reinforcement Learning (pp. 45-58).
>
> [3] Finn, C., Christiano, P., Abbeel, P., & Levine, S. (2016). A connection between generative adversarial networks, inverse reinforcement learning, and energy-based models. arXiv preprint arXiv:1611.03852.

---

### Official Review · AnonReviewer1 · 2020-11-03
**Single Timescale Actor Critic Provably Finds Globally Optimal Policy**

**Rating:** 5
**Confidence:** 4

**Review:**

Summary:

The paper studies global convergence of actor-critic methods in the single-timescale setting, where in each iteration, the critic is update using the Bellman operator only once. The authors consider a setting where both the actor and critic are represented with either a linear model or a deep neural network. For this, the paper shows a sublinear rate of convergence of O(1/\sqrt{K}) to the optimal policy.

Comments:

- I think the analysis in the single timescale setting is an interesting question and probably more challenging than the two-timescale setting where the critic is assumed to be trained to (near) optimality. However, I am not sure how useful the result itself is - it matches the results for two-timescale settings. Why is such a result interesting? The authors don't show any lower bounds. Intuitively, one might expect the single timescale setting to converge faster than a double loop kind of algorithm. In some ways the paper leaves me more confused.

- The authors claim the one advantage of the single timescale setting is the fact that it is amenable to the off-policy setting. I seem to be missing something here. Why can't the same argument be made for the two timescale setting? It might be useful to have a discussion on this in the paper.

- Another confusion I have is that the paper nowhere assumes that the optimal policy is contained in the policy class. How does one get a global optimality result without this? In the function approximation setting, with the class of energy based policies, is there a characterization of the best policy in the policy class and the approximation error it incurs? Intuitively, I was expecting a residual error term in the bound.

- Lastly, and this is one of my main complaints, I tried very hard to follow the proofs in the Appendix and despite my best efforts over a couple of days, I still don't have any intuition why this result holds - specifically, why only a single application of the Bellman operator to update the critic is enough. What do the authors mean by 'double contraction', a term often used in the paper. Clearing out Section D in the Appendix would be helpful - currently there are a bunch of equations, some with proofs in Appendix E etc. I am unable to verify these easily and hence not sure how the authors got the results. The main text only spells out the setting along with a rather long section on related works etc. but gives no intuition.

---

> ### Author Response · Authors · 2020-11-18
> **Review Response**
>
> We appreciate the valuable comments from the reviewer. We address the concerns as follows.
> 1. (Comparison with two-timescale setting and bilevel setting.)  In practice, most actor-critic type algorithms adopt single-timescale updates, where both critic and actor use the **same stepsize**. Thus, the actor is updated before the critic achieves the value function of that actor. This implies that the actor update direction is different from the ideal update direction specified by policy gradient or natural policy gradient. To alleviate this problem, two-timescale updates equip the critic with a larger stepsize,  whereas the bilevel setting uses an inner loop to solve the policy evaluation problem separately. Thus, when focusing on the actor updates, thanks to the two-timescale or the bilevel schemes, the update direction is approximately the ideal policy gradient or natural policy gradient directions. Such schemes bring simplicity to theoretical analysis, however, are different from the practical actor-critic.
> In contrast, our single-timescale algorithm aims to capture the characteristics of practical actor-critic that the actor is updated before the critic approximately solves the associated policy evaluation problem. To this end, for policy update, we only apply the Bellman operator **once**. (Note that if we use an inner loop to apply the Bellman operator for sufficiently many times, we recover a bilevel algorithm.) Hence, the critic is different from the action-value function of the current actor, which leads to a biased estimator of the actor update direction. Such biased actor updates capture the intrinsic difficulty of single-timescale actor-critic, which has not been fully addressed in existing literature.
> Also, in terms of sample complexity for achieving $\varepsilon$ sub-optimality, our single-timescale algorithm achieves $O(\varepsilon^{-2})$ by reusing the data (off-policy sampling). Also see point 2 for a detailed discussion. In comparison, the two-timescale setting only gives a sample complexity of $O(\varepsilon^{-4})$. If we draw new data in each iteration of policy update (on-policy sampling), we indeed recover the same sample complexity as two-timescale actor-critic. In other words, single-timescale actor-critic achieves superior performance with off-policy data, which is advocated by practical implementations where experience replay is shown to be very beneficial.
> 2. (Off-policy evaluation.)  Our LSTD update in the critic can be readily modified for the off-policy setting, which is discussed in many previous works (Antos et al., 2007; Yu, 2010; Liu et al., 2018; Nachum et al., 2019; Xie et al., 2019; Zhang et al., 2020; Uehara and Jiang, 2019; Nachum and Dai, 2020).  However, in a typical two-timescale setting, the critic is updated using TD learning (e.g. TD(0) and TD($\lambda$)), which is shown to diverge even with linear function approximation under off-policy setting [1,2]. One possible way to employ off-policy evaluation under this setting is to use gradient TD [2], which transforms the optimization problem of policy evaluation into a minimax optimization problem and such an algorithm is only known to converge under the linear function approximation setting. When nonlinear function approximators are used, the only convergent gradient TD algorithm is proposed by [3], which requires the Hessian of the value function and thus cannot be applied to neural networks. Meanwhile, gradient TD in [3] itself is a two-timescale algorithm and its convergence guarantee is only asymptotic.
> 3. (Policy class and approximation error.) Under the linear setting, the structural assumption of the MDP is given in Assumption 4.2. Specifically, it postulates that the Bellman operator of each policy maps a linear value function to a linear function. Thus, the value function associated with each policy, being the fixed point of the Bellman operator, lies in the linear function class. For the linear setting, the policy class we consider is the energy-based policy where the energy function is linear. Because we know that the value functions are linear, such a policy class approximately covers the optimal policy by letting the temperature go to zero. In summary, our Assumption 4.2 ensures that the energy-based policy class approximately captures the optimal policy and thus there is no approximation error. When this assumption does not hold, we only need to add an additional linear term in $K$ to the regret upper bound in Theorem 4.4 without much change in the proof.
> Similarly, Assumption C.2 also ensures that energy-based policies with overparameterized neural networks capture the optimal policy approximately. When such an assumption fails to hold, we also only need to add a linear term in $K$ to the regret upper bound in Theorem C.5 without much change in the proof.

---

> > ### Author Response · Authors · 2020-11-18
> > **Review Response (Cont.)**
> >
> > 4. (Proof of main theorem.)  We thank the reviewer diving into the proof of our theorem. In the proof of our main theorem, we first decompose the total regret into three terms: $A_{1,k}$, $A_{2,k}$, and $A_{3,k}$. Roughly speaking, $A_{1,k}$ represents the convergence of $\pi_{\theta_k}$ towards $\pi^*$, $A_{2,k}$ represents the difference between $\pi_{\theta_k}$ and $\pi_{\theta_{k+1}}$, while $A_{3,k}$ represents the difference between $Q^{\pi_{\theta_k}}$ and $Q^{\pi_{\theta_{k+1}}}$.
> > Among these three terms, $A_{3,k}$ plays a key role in the proof of the convergence, since its contraction not only implies the contraction of the critic, but also implies the contraction of the actor through the difference between $\pi_{\theta_k}$ and $\pi_{\theta_{k+1}}$. This is the “double contraction” phenomenon we mentioned in our work. Meanwhile, by unrolling the term $A_{3,k}$, we know that $A_{3,k}$ is fully characterized by $e_{k+1}$ defined in (D.6). Please see Figure 1 on page 18 for the relationship among these quantities. Also, by recursion, roughly we have $e_{k+1} = O(\gamma^k)$, which is a contraction. By further assembling the components mentioned above, we obtain the convergence of our single-timescale AC algorithm.
> >
> >
> > [1] Baird, Leemon. "Residual algorithms: Reinforcement learning with function approximation." Machine Learning Proceedings 1995. Morgan Kaufmann, 1995. 30-37.
> >
> > [2] Sutton, Richard S., Csaba Szepesvári, and Hamid Reza Maei. "A convergent O (n) algorithm for off-policy temporal-difference learning with linear function approximation." Advances in neural information processing systems 21.21 (2008): 1609-1616.
> >
> > [3] Bhatnagar, S., Precup, D., Silver, D., Sutton, R. S., Maei, H., & Szepesvári, C. (2009). Convergent temporal-difference learning with arbitrary smooth function approximation. Advances in neural information processing systems, 22, 1204-1212.

---

### Author Response · Authors · 2020-11-23
**General response**

We thank the reviewers for the valuable comments and suggestions!  The main contribution of our work is that we establish the rate of convergence and global optimality of **single-timescale**actor-critic with function approximation for the first time. Also, with off-policy evaluation, we show that our single-timescale actor-critic achieves a better sample complexity compared to bilevel and two-timescale settings.

We would like to address some common concerns as follows.
1. (Comparison with bilevel and two-timescale setting.)  In practice, most actor-critic algorithms adopt single-timescale updates, where both actor and critic use the **same stepsize**. Thus, the actor is updated before the critic achieves the value function of that actor. This implies that the actor update direction is different from the ideal update direction specified by (natural) policy gradient. To alleviate this problem, two-timescale updates equip the critic with a **larger stepsize**, whereas the bilevel setting uses an **inner loop**to solve the policy evaluation problem separately. Thus, when focusing on the actor updates, thanks to the two-timescale or the bilevel schemes, the update direction is approximately the ideal (natural) policy gradient. Such schemes bring simplicity to theoretical analysis, however, are different from the practical actor-critic.
In contrast, our single-timescale algorithm aims to capture the characteristic of practical actor-critic that the actor is updated **before**the critic approximately solves the associated policy evaluation problem. To this end, for policy update, we only apply the Bellman operator **once**. (Note that if we use an inner loop to apply the Bellman operator for **sufficiently many times**, we recover a bilevel algorithm.) Hence, the critic is different from the action-value function of the current actor (i.e., the action-value function $Q^\pi$ is associated with the current actor $\pi$), which leads to a biased estimator of the actor update direction. Such biased actor updates capture the intrinsic difficulty of single-timescale actor-critic, which has not been fully addressed in existing literature.
2. (LSTD and TD under off-policy setting.)  Our LSTD update in the critic can be readily modified for the off-policy setting, which is discussed in many previous works (Antos et al., 2007; Yu, 2010; Liu et al., 2018; Nachum et al., 2019; Xie et al., 2019; Zhang et al., 2020; Uehara and Jiang, 2019; Nachum and Dai, 2020). However, in the two-timescale and bilevel settings, the critic is updated using TD learning (e.g. TD(0) and TD($\lambda$)), which is shown to diverge even with linear function approximation under off-policy setting [1,2]. One possible way to employ off-policy evaluation under this setting is to use gradient TD [2], which transforms the optimization problem of policy evaluation into a minimax optimization problem and such an algorithm is only known to converge under the linear function approximation setting. When nonlinear function approximators are used, the only convergent gradient TD algorithm is proposed by [3], which requires the Hessian of the value function and thus cannot be applied to neural networks. Meanwhile, gradient TD in [3] itself is a two-timescale algorithm and its convergence guarantee is only asymptotic.
3. (Sample Complexities.)  In terms of sample complexity, under linear function approximation, to achieve an $\varepsilon$-globally optimal policy, the total sample complexity of our single-timescale algorithm is $O(\varepsilon^{-4})$, while that of two-timescale algorithm (Xu et al., 2020; Hong et al., 2020) is also $O(\varepsilon^{-4})$. In addition, with data reuse (off-policy evaluation), the total sample complexity of our single-timescale algorithm reduces to $O(\varepsilon^{-2})$, while the idea of data reuse cannot be applied to two-timescale setting (see point 2 for details).
Under deep neural network approximation, to achieve an $\varepsilon$-globally optimal policy, the total sample complexity of our single-timescale algorithm is $O(\varepsilon^{-6})$, while that of bilevel algorithm (Liu et al., 2018) is $O(\varepsilon^{-8})$. In addition, with data reuse, the total sample complexity of our single-timescale algorithm reduces to $O(\varepsilon^{-4})$, while the idea of data reuse cannot be applied to bilevel setting (see point 2 for details).

---

> ### Author Response · Authors · 2020-11-23
> **General response (Cont.)**
>
>
> As per the suggestions of the reviewers, we would also like to summarize major changes in our revision as follows.
> 1. We add more discussion under Assumption 4.2 to illustrate that under this assumption, the optimal policy lies in the policy class that we consider, and thus there is no approximation error.
> 2. We add more discussion under Theorem 4.4 regarding the off-policy setting. Specifically speaking, we cannot use off-policy evaluation in the two-timescale setting, since TD learning is employed in the critic update, which is shown to diverge under off-policy setting.
> 3. Under Theorem C.5 in the appendix, we compare our sample complexity with that of (Liu et al., 2018), and show that our single-timescale algorithm achieves better sample complexity compared with the bilevel algorithm in (Liu et al., 2018).
>
>
> In the meantime, we are happy to answer any questions or concerns that remain. We hope to hear back from the reviewers!
>
> [1] Baird, Leemon. "Residual algorithms: Reinforcement learning with function approximation." Machine Learning Proceedings 1995. Morgan Kaufmann, 1995. 30-37.
>
> [2] Sutton, Richard S., Csaba Szepesvári, and Hamid Reza Maei. "A convergent O (n) algorithm for off-policy temporal-difference learning with linear function approximation." Advances in neural information processing systems 21.21 (2008): 1609-1616.
>
> [3] Bhatnagar, S., Precup, D., Silver, D., Sutton, R. S., Maei, H., & Szepesvári, C. (2009). Convergent temporal-difference learning with arbitrary smooth function approximation. Advances in neural information processing systems, 22, 1204-1212.

---

### Decision · Program_Chairs · 2021-01-07
**Final Decision**

**Decision:**

Accept (Poster)

**Comment:**

Most reviewers agree that the paper makes valuable contribution in analyzing single-timescale actor-critic algorithms. There were some doubts on the theoretical advantage over two-timescale algorithms and the realizability assumptions, but the authors made satisfactory clarifications.

Therefore, acceptance is recommended, though I strongly suggest the authors to explicitly state key assumptions required to ensure global optimality in the abstract and introduction to avoid confusion.